# Family Matters: A Systematic Study of Spatial vs. Frequency Masking for Continual Test-Time Adaptation

**Chandler Timm C. Doloriel**[1], **Yunbei Zhang**[2], **Yeonguk Yu**[3], **Taki Hasan Rafi**[4],
**Muhammad Salman Siddiqui**[1], **Tor Kristian Stevik**[1], **Fadi Al Machot**[1],
**Kristian Hovde Liland**[1], **Habib Ullah**[1]

[1] *Faculty of Science and Technology (REALTEK), Norwegian University of Life Sciences (NMBU)*
[2] *Tulane University*
[3] *Gwangju Institute of Science and Technology*
[4] *Hanyang University*
*chandler.timm.cagmat.doloriel@nmbu.no, habib.ullah@nmbu.no*

**Reviewed on OpenReview:** *https://openreview.net/forum?id=pBI64qNXHp*

## Abstract

Recent continual test-time adaptation (CTTA) methods adopt masked image modeling to stabilize learning under distribution shift, yet each treats its masking family $\mathcal{F}$ as a fixed design choice and innovates exclusively along the selection strategy $\mathcal{S}$, leaving the family axis underexplored. We present a systematic empirical study that isolates this axis. Using a controlled CTTA instantiation, Mask to Adapt (M2A), that fixes $\mathcal{S}=random$ and standard losses, we vary only $\mathcal{F}$ across spatial (patch, pixel) and frequency (all-band, low-band, high-band) families while keeping every other component identical. The study's contributions are the design guidance it extracts for the CTTA settings we evaluated: (1) *the masking family determines whether adaptation compounds useful structure or compounds errors*, on patch-tokenized architectures, spatial masking accumulates stable representations over long streams while frequency masking collapses catastrophically. We characterize this instability through a *structural-preservation* account, where spatial coherence maintains the broad-spectrum redundancy needed to avoid terminally overlapping with a corruption's spectral signature; (2) *the optimal family depends on architecture-task alignment*, on CNNs, whose overlapping receptive fields dilute patch occlusion, the family gap vanishes, whereas on fine-grained tasks with global cues and large-capacity ViTs, frequency masking becomes competitive. In confounded system-level comparisons, where baselines also differ in losses and auxiliary components, M2A's random selection performs comparably to heuristic strategies, though we treat this observation as suggestive context rather than a controlled quantification of $\mathcal{S}$'s relative importance.

## 1 Introduction

Distribution shifts at test time can severely degrade the performance of vision models. Test-Time Adaptation (TTA) addresses this by updating a pre-trained model on unlabeled test data, and a growing family of methods, spanning entropy minimization (Wang et al., 2021; Niu et al., 2023), weight averaging (Wang et al., 2022), prompt and adapter tuning (Gan et al., 2023; Liu et al., 2024b), teacher–student regularization (Brahma & Rai, 2023), and self-supervised feature stabilization (Sun et al., 2020; You et al., 2025; Zhang et al., 2025a; Ambekar et al., 2025), has been proposed. While effective on single or slowly varying domains, these mechanisms can still accumulate errors or overfit recent domains under long, strongly corrupted streams.

Recent CTTA methods (Maharana et al., 2026) increasingly rely on *masking* to stabilize adaptation, but each commits to a single masking family and invests its innovation in the selection strategy, Continual-MAE (Liu et al., 2024a) pairs patch masking with uncertainty scoring, and REM (Han et al., 2025) pairs patch

masking with attention ranking, without ever comparing across families. Related TTA methods follow the same pattern: SPA (Niu et al., 2025) optimizes active weak-to-strong consistency ($\mathcal{S}$=rules/policy) via low-frequency amplitude masking and noise injection, and TCA (Wang et al., 2025b) uses domain-aware attention ($\mathcal{S}$=attention) to prune and merge tokens. Any masking-based CTTA method makes two orthogonal choices: a *masking family* $\mathcal{F} \in$ {patch, pixel, all-freq, low-freq, high-freq} and a *selection strategy* $\mathcal{S}$ (Figure 1). Prior work treats $\mathcal{F}$ as a fixed design choice and innovates exclusively along $\mathcal{S}$; this paper inverts the emphasis by fixing $\mathcal{S}$=random and varying $\mathcal{F}$, systematically quantifying how much the masking family matters relative to the selection strategy.

To test this, we build a controlled CTTA instantiation, Mask to Adapt (M2A), that fixes $\mathcal{S}$=random and varies $\mathcal{F}$ while keeping all other components, standard consistency and entropy losses (Wang et al., 2022; Liu et al., 2024a; Han et al., 2025), a masking schedule, and a single gradient step per batch, identical across conditions, so that any performance difference is attributable to $\mathcal{F}$ alone. We compare spatial masking (Bao et al., 2022; He et al., 2021; Xie et al., 2021; Wei et al., 2021) (patch, pixel) against frequency masking (Xie et al., 2023; Wang et al., 2023; Monsefi et al., 2025) (all-band, low-band, high-band) across standard corruption benchmarks, multiple streaming protocols, backbone architectures, and task domains (Section 4; Appendix).

Our primary contribution is a systematic empirical study of the underexplored family axis, yielding actionable design guidance organized as two findings: ❶ **The masking family determines whether adaptation compounds useful structure or compounds errors.** On patch-tokenized architectures, spatial masking accumulates stable representations over long streams while frequency masking collapses catastrophically. This instability can be explained by *structural preservation*: spatial coherence maintains broad-spectrum structural redundancy, whereas frequency-domain zeroing can terminally overlap with the environment's noise profile, such as blur concentrating power centrally while attenuating the periphery. We use structural preservation as a conceptual explanation for these family-level trends (Sections 4, 5). ❷ **The optimal family depends on architecture-task alignment.** On CNNs, whose overlapping receptive fields dilute patch occlusion, the family gap vanishes and the choice is less consequential. On fine-grained tasks where discriminative cues are global rather than spatially localized, frequency masking becomes competitive or preferable, but only on large-capacity ViTs, whereas on smaller backbones, the perturbation overwhelms the adaptation signal. The guidance is thus condition-specific: prefer patch masking on ViTs with spatially localized cues, whereas on global-cue tasks with large ViTs, frequency masking is a viable alternative. As a side note, in confounded system-level comparisons, where baselines differ from M2A in losses and auxiliary components, random selection performs comparably to heuristic strategies. Because these comparisons cannot isolate $\mathcal{S}$ from other methodological variations, we treat this observation as suggestive context. Ablations confirm broad robustness to hyperparameter choices (Section 4; Appendix).

## 2 Related Work

### 2.1 Masked Image Modelling

Masked image modelling (MIM) has emerged as a strong pre-training paradigm in both spatial and frequency domains. Spatially, BEiT (Bao et al., 2022) and MAE (He et al., 2021) showed that masking large fractions of patches yields useful representations, and SimMIM (Xie et al., 2021) and MaskFeat (Wei et al., 2021) further simplified the pipeline. In the frequency domain, MFM (Xie et al., 2023), FreMIM (Wang et al., 2023), and FOLK (Monsefi et al., 2025) mask spectral coefficients and reconstruct the missing spectrum. Follow-up work varies masking ratios, sampling policies, and reconstruction targets and deploys MIM features across classification, detection, and segmentation, but almost always under offline, stationary evaluation. Crucially, the two families differ structurally: spatial masking removes localized content while leaving remaining pixels intact, whereas frequency masking suppresses globally distributed coefficients, altering every pixel. This distinction matters little for reconstruction, where a decoder compensates for either perturbation, but may matter for prediction-centric adaptation, where no decoder is available and the loss operates directly on classifier outputs (see conclusion, Section 5).

Recent work confirms that masking can stabilize CTTA, but transferring MIM is not straightforward: CTTA replaces reconstruction with prediction-centric objectives (entropy, consistency) on a non-stationary, single-

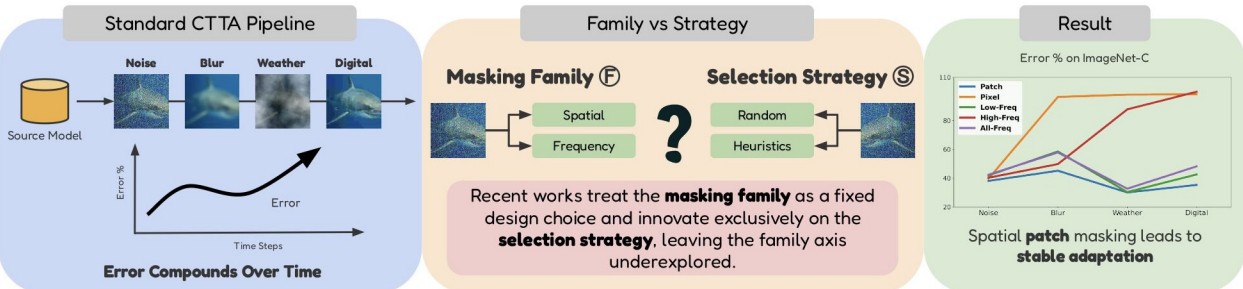

Figure 1: **Motivation: The masking family axis is underexplored in CTTA.** Recent masking-based CTTA methods couple a specific masking family $\mathcal{F}$ with a specific selection strategy $\mathcal{S}$, innovating exclusively along the strategy axis while treating the family as a fixed design choice. **Left:** Standard CTTA pipeline showing error accumulation over time under distribution shift. **Center:** The two orthogonal design axes—masking family (spatial vs. frequency) and selection strategy (random vs. heuristics)—are typically confounded in prior work. **Right:** Our systematic study isolates the family axis by fixing $\mathcal{S}$=random and varying only $\mathcal{F}$, revealing that spatial patch masking leads to stable adaptation through structural preservation: maintaining spatially coherent views preserves broad-spectrum redundancy, avoiding terminal overlap with corruption spectral signatures that cause frequency masking to collapse.

pass stream where updates must avoid catastrophic forgetting. Existing masking-based systems therefore inherit MIM's intuition about information removal but re-purpose it for robustness rather than representation learning, and it remains unclear a priori whether the spatial–frequency distinction above still governs stability once reconstruction is removed from the loop.

## 2.2 Test-Time Adaptation

Test-Time Adaptation (TTA) (Wang et al., 2021; Liang et al., 2024; Sun et al., 2020; Xiao & Snoek, 2024; Zhang et al., 2025a; Ambekar et al., 2025; Zhang et al., 2026) updates a model on unlabeled test data. Early methods, Pseudo-Labeling (Lee, 2013), corruption benchmarks like ImageNet-C (Hendrycks & Dietterich, 2018), and Tent (Wang et al., 2021), established entropy minimization as the dominant paradigm. Subsequent work added self-supervised objectives (Sun et al., 2020), feature-based alignment (You et al., 2025), and parameter-efficient transformer schemes (Zhang et al., 2025a; Ambekar et al., 2025). These approaches span entropy reduction, consistency enforcement, and feature stabilization, and differ in how aggressively they update the model (from single-pass to episodic adaptation), but all assume roughly stationary batches, leaving them prone to error accumulation under long, corrupted streams.

**Continual Test-Time Adaptation** (CTTA) (Maharana et al., 2026; Wang et al., 2022; Liu et al., 2024b; Zhang et al., 2025b; Ni et al., 2025) addresses evolving streams via weight averaging (Wang et al., 2022), prompt/adapter tuning (Gan et al., 2023; Liu et al., 2024b; Zhang et al., 2025b), or gradient filtering and teacher–student regularization (Niu et al., 2023; Brahma & Rai, 2023). These methods differ in how they control drift, for example, by anchoring to source weights, constraining updates to adapters, or filtering gradients, but share the goal of maintaining performance over long, non-stationary streams. Recent masking-based CTTA baselines each couple a specific $\mathcal{F}$ with a specific $\mathcal{S}$: Continual-MAE (Liu et al., 2024a) pairs $\mathcal{F}$=*patch* with $\mathcal{S}$=*uncertainty*, and REM (Han et al., 2025) pairs $\mathcal{F}$=*patch* with $\mathcal{S}$=*attention*. Related TTA methods follow the same pattern: SPA (Niu et al., 2025) optimizes active weak-to-strong consistency ($\mathcal{S}$=*rules/policy*) via low-frequency amplitude masking and noise injection, and TCA (Wang et al., 2025b) uses domain-aware attention ($\mathcal{S}$=*attention*) to prune and merge tokens. In effect, the field has invested in $\mathcal{S}$ while taking $\mathcal{F}$ for granted, the dominant methods default to patch masking without testing alternatives, and most results are reported on a narrow set of backbones and protocols, leaving the family axis underexplored.

Our work addresses this gap with a systematic empirical study that isolates the masking-family axis in CTTA. We introduce a simple masking-based adapter (M2A) that holds the selection strategy and objectives fixed

while varying the masking family, and we show that this axis both governs long-term stability and interacts strongly with architecture–task alignment (see Sections 1 and 4).

## 3 Methodology

**Preliminaries.** Let $f_\theta : [0,1]^{C \times H \times W} \to \mathbb{R}^K$ be a source-trained classifier producing logits $z = f_\theta(x) \in \mathbb{R}^K$ and probabilities $p = \mathrm{softmax}(z) \in \Delta^{K-1}$. At test time, we receive an unlabeled stream of batches $\{B_s\}_{s \geq 1}$ from potentially different corruption domains. For each batch, M2A performs a single gradient step on $\mathcal{L}_{\mathrm{CTTA}}$ (Section 3.3) and carries the updated $\theta$ forward without reset, matching the standard online CTTA protocol of CoTTA (Wang et al., 2022), Continual-MAE (Liu et al., 2024a), and REM (Han et al., 2025). All technical ingredients below are shared across conditions to isolate the masking family $\mathcal{F}$ as the sole experimental variable. We organize these families through a *structural-preservation principle*: masking families that preserve spatially contiguous redundancy so masked views remain broadly informative, and avoid masking bands that already coincide with the corruption's damage zone, are hypothesized to be more stable under online adaptation. We use this principle as our qualitative explanation for the family-level trends. It has two ingredients: *spatial coherence*, meaning that masking should preserve contiguous structure rather than fragment it, and *spectral overlap*, meaning that masking should avoid deleting the same frequency bands that the corruption has already degraded. Within this organizing hypothesis, spatial families differ mainly through coherence, whereas frequency families differ mainly through overlap. The experiments are designed to test whether the observed family-level patterns are consistent with this interpretation, while Appendix A.3 gives a conceptual formalization of the same lens.

The masking schedule is shared across all families to avoid confounding $\mathcal{F}$ comparisons. Given a positive integer $n$ (number of masking views) and a mask step $\alpha \in (0,1)$, we set $m_t = t\,\alpha$ for $t \in \{0,\ldots,n-1\}$ with $m_0 = 0$, so that $x^{(0)}$ is the unmasked input and each subsequent view is progressively more heavily masked: for example, with the default $n{=}3$ and $\alpha{=}0.1$ used in our experiments, the three views have masked-area fractions 0%, 10%, and 20%, producing an easy–medium–hard progression that balances stability (the unmasked anchor preserves source knowledge) and plasticity (harder views push the model toward domain-robust features). For each $m_t$, we construct a masked view $x^{(t)}$ of $x$ using one of the spatial or frequency mechanisms described below, forward it to obtain $z^{(t)} = f_\theta(x^{(t)})$ and $p^{(t)} = \mathrm{softmax}(z^{(t)})$, and optimize the combined loss $\mathcal{L}_{\mathrm{CTTA}}$ with a single gradient step per batch.

We restrict to zero-out masking families (spatial or spectral), following MIM literature (Hondru et al., 2024). We also only focus on input-based masking, not feature-based. We do not use reconstruction; augmentations like blur or noise are not masking in the MIM sense, hence we do not implement or include them in the analysis.

### 3.1 Random Spatial Masking

Spatial masking instantiates structural preservation by removing local content while maintaining global layout in the remaining pixels. For each difficulty level $t$ with target masked fraction $m_t$, we sample a binary mask $M^{(t)} \in \{0,1\}^{H \times W}$ and zero locations across channels: $x^{(t)} = x \odot (\mathbf{1} - \tilde{M}^{(t)})$, where $\tilde{M}^{(t)} := \mathrm{broadcast}_C(M^{(t)}) \in \{0,1\}^{C \times H \times W}$ and $\odot$ is elementwise multiplication. We instantiate $M^{(t)}$ via two mechanisms.

**Patch-based masking.** In all reported experiments, $\mathcal{F}{=}$patch denotes a single contiguous *Block-Patch* aligned to the token grid, not a scattered subset of token-aligned patches. We reserve the term *Free-Patch* for that fragmented variant, which is evaluated only in Appendix A.5. We mask one contiguous square region to force reliance on global context. In our ViT experiments the block side length matches the model's input tokenization grid (e.g., 16×16 for ViT-B/16), so the masked region is aligned with the token lattice; this alignment is deliberate and may strengthen the masking signal on patch-tokenized architectures, a possibility examined in the architecture study in Section 4. Formally, for each view $t$ we sample one top-left corner $(y_0^{(t)}, x_0^{(t)})$ and define $\mathcal{A}^{(t)} := \{(i,j) : y_0^{(t)} \leq i < y_0^{(t)}{+}s_t,\ x_0^{(t)} \leq j < x_0^{(t)}{+}s_t\}$ with $|\mathcal{A}^{(t)}| \approx m_t HW$, then set $M^{(t)}(i,j) = \mathbf{1}[(i,j) \in \mathcal{A}^{(t)}]$.

**Pixel-wise masking.** As a finer-grained alternative, we draw $k_t = \lfloor m_t HW \rceil$ pixel positions uniformly at random without replacement, i.e. $\mathcal{A}^{(t)} \sim \mathrm{Unif}(\{\mathcal{S} \subseteq \{0,\ldots,H{-}1\} \times \{0,\ldots,W{-}1\} : |\mathcal{S}| = k_t\})$, and

set $M^{(t)}(i,j) = \mathbf{1}[(i,j) \in \mathcal{A}^{(t)}]$. This sparse occlusion perturbs low-level details while largely preserving global structure. Extended implementation details (patch placement, rejection sampling) are provided in Appendix A.4.1.

## 3.2 Random Frequency Masking

Frequency masking probes structural preservation by zeroing spectral components globally. This mechanism tests stability when perturbations target specific scales that may terminally overlap with the domain's spectral signature. The pipeline proceeds in three steps. (i) We apply a 2D orthonormal Fourier transform to each channel, $X_c = \mathrm{DFT}_{\mathrm{ortho}}(x_c)$. (ii) We select $k_t = \lceil m_t HW \rceil$ frequency bins to zero from the chosen band (see below), always masking bins in *conjugate pairs*, if $(u,v)$ is zeroed, so is $(-u \bmod H, -v \bmod W)$, to preserve the Hermitian symmetry of the spectrum. Self-conjugate bins (DC, and Nyquist bins when $H$ or $W$ is even) are excluded from masking, guaranteeing that the DC component and the real-valuedness of the signal are maintained. The masked spectrum is $\tilde{X}_c^{(t)}[u,v] = M^{(t)}[u,v]$
$, X_c[u,v]$. (iii) We reconstruct via the orthonormal inverse transform and take the real part: $x_c^{(t)} = \mathrm{Re}\left(\mathrm{DFT}_{\mathrm{ortho}}^{-1}(\tilde{X}_c^{(t)})\right)$. Because conjugate symmetry is preserved, the imaginary residual is numerically negligible ($\lesssim 10^{-7}$). Zeroing bins can introduce mild Gibbs-like ringing, but at the moderate ratios used ($m_t \leq 20\%$) these artifacts are small and subsumed by the corruption already present in the target stream.

Band membership is defined by the normalised radial distance $r(u,v) \in [0,1]$ from DC, computed in centered-frequency coordinates, with a cutoff at $r_c = 0.5$.

**All-frequency masking.** $\Omega_{\mathrm{all}}$: all non-self-conjugate pairs, uniform removal across scales.

**Low-frequency masking.** $\Omega_{\mathrm{low}}$: pairs with $r \leq r_c$, perturbs coarse structure and illumination. Noise corruptions distribute energy across all bands, while blur acts as a low-pass filter that attenuates the spectral periphery; removing the surviving low-frequency structure strips the remaining intact signal.

**High-frequency masking.** $\Omega_{\mathrm{high}}$: pairs with $r > r_c$, affects textures and edges. Blur and weather corruptions act as low-pass filters that attenuate the spectral periphery; masking this band compounds the loss of informative high-frequency detail. Extended implementation details (self-conjugate bin enumeration, conjugate-pair selection, orthonormal DFT formulas) are provided in Appendix A.4.2.

## 3.3 Loss Function

Both loss components below, cross-view consistency and entropy minimization, are standard objectives used by CoTTA, Continual-MAE, and REM, adopted unchanged so that any performance difference across families is attributable to the masking mechanism rather than the objective. We write $H(p)$ for the Shannon entropy and $H(p,q)$ for the cross-entropy between distributions $p,q \in \Delta^{K-1}$. The operator $\mathrm{sg}(\cdot)$ denotes stop-gradient.

We model the unlabeled stream as draws from a family of target domains $D \in \mathcal{D}$, each with distribution $P_D$ over inputs $x$. Random masking augments every $x$ with views $x^{(0)}, \ldots, x^{(n-1)}$ using the spatial and frequency mechanisms above; writing $\mathrm{Mask}(x,M)$ for the masking operator and $M$ for the random mask (encoding mask type and level $m_t$), the ideal masked risk on domain $D$ is $\overline{\mathcal{R}}_D(\theta) = \mathbb{E}_{(x,y) \sim P_D} \mathbb{E}_M[\ell(f_\theta(\mathrm{Mask}(x,M)), y)]$. Since CTTA never observes labels $y$, we optimize the self-supervised surrogate $\mathcal{L}_{\mathrm{CTTA}} = \mathcal{L}_{\mathrm{CL}} + \lambda \mathcal{L}_{\mathrm{EML}}$: the consistency loss in Equation 1 makes predictions $p^{(t)}$ approximately mask-invariant across views of the same $x$, while the entropy loss in Equation 2 encourages confident, cluster-aligned predictions along the evolving domain sequence $(D_s)_{s \geq 1}$.

**Consistency Loss**. We enforce consistency by aligning masked-view predictions to earlier views (including the unmasked anchor), with the targets detached:

$$\mathcal{L}_{\mathrm{CL}} = \sum_{t=1}^{n-1} \sum_{r=0}^{t-1} H\big(p^{(t)}, \mathrm{sg}(p^{(r)})\big). \tag{1}$$

**Entropy Loss**. We encourage confident predictions on all masking levels by minimizing the average entropy across views:

$$\mathcal{L}_{\text{EL}} \;=\; \frac{1}{n} \sum_{t=0}^{n-1} H\big(p^{(t)}\big), \qquad H(p) = - \sum_{k=1}^{K} p_k \log p_k. \tag{2}$$

### 3.3.1 Total Loss

The combined objective $\mathcal{L}_{\text{CTTA}} = \mathcal{L}_{\text{CL}} + \lambda\mathcal{L}_{\text{EL}}$ promotes mask-invariant predictions within each domain (via $\mathcal{L}_{\text{CL}}$) and low-entropy, cluster-aligned predictions across the CTTA stream (via $\mathcal{L}_{\text{EL}}$). The two are complementary: $\mathcal{L}_{\text{CL}}$ encourages features invariant to the specific perturbation introduced by the masking family; $\mathcal{L}_{\text{EL}}$ sharpens predictions toward confident class assignments, preventing the model from settling into a uniform-prediction regime. The weight $\lambda$ controls the balance: ablations (Section 4) show that removing $\mathcal{L}_{\text{EL}}$ ($\lambda$=0) causes catastrophic spikes, while any positive $\lambda$ yields stable performance, indicating that the entropy term is essential but its precise magnitude is not critical. Because both losses operate on predictions rather than reconstructed pixels, their usefulness still depends on whether the masking family leaves enough structure for cross-view agreement. The per-batch procedure is detailed in Appendix A.2.

## 4 Experiments

### 4.1 Dataset

We evaluate on CIFAR-10/100-C and ImageNet-C (Hendrycks & Dietterich, 2018), the standard corruption benchmarks derived from CIFAR (Krizhevsky et al., 2009) and ImageNet (Deng et al., 2009). Each contains 15 corruption types at 5 severity levels; following prior work (Wang et al., 2022; Liu et al., 2024b;a; Han et al., 2025), we use the highest severity level (5) for all 15 corruptions and report online classification error after adaptation for each target domain. We also evaluate on a real-world aquaculture dataset, MRSFFIA-C (Du et al., 2024). Further detailes are provided in Appendix.

Corruption abbreviations used in figures and tables: GN=Gaussian noise, SN=Shot noise, IN=Impulse noise, DB=Defocus blur, GB=Glass blur, MB=Motion blur, ZB=Zoom blur, S=Snow, Fr=Frost, F=Fog, B=Brightness, C=Contrast, ET=Elastic transform, P=Pixelate, JC=JPEG compression.

### 4.2 Implementation Details

All experiments use a ViT-B/16 backbone (Dosovitskiy et al., 2021) trained on the source data. Baselines include Pseudo-label (Lee, 2013), Tent (Wang et al., 2021), VDP (Gan et al., 2023), SAR (Niu et al., 2023), CoTTA (Wang et al., 2022), PETAL (Brahma & Rai, 2023), ViDA (Liu et al., 2024b), RoTTA (Yuan et al., 2023), DeYO (Lee et al., 2024), ROID (Marsden et al., 2024), Continual-MAE (Liu et al., 2024a) ($\mathcal{F}$=patch, $\mathcal{S}$=uncertainty), and REM (Han et al., 2025) ($\mathcal{F}$=patch, $\mathcal{S}$=attention). All reported results use seeds 1, 2, and 3, and we report the mean and standard deviation across these runs.

**Default hyperparameters.** Unless stated otherwise: one gradient step per batch; Adam optimizer with learning rate $10^{-3}$; weight decay 0.0; mask step $\alpha$=0.1; $n$=3 masking views; batch size 64 for ImageNet-C and 20 for CIFAR10-C/100-C. We follow REM (Han et al., 2025) and update only a set of the layernorm parameters. All runs use a single AMD Instinct MI200 GPU. Unless stated otherwise, $\mathcal{F}$=patch in all tables and figures refers to the single contiguous *Block-Patch* mask from Section 3. The scattered *Free-Patch* and checkerboard *Grid-Patch* variants are evaluated only in the appendix ablations on mask sampling methods. In the discussion below, we use the structural-preservation principle mainly as a qualitative lens for the central family contrast and the scoped exception cases, while keeping the remaining comparisons descriptive.

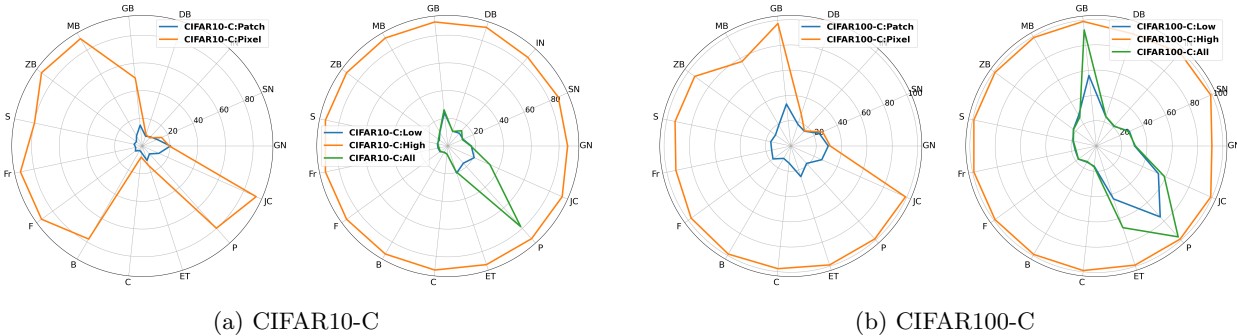

(a) CIFAR10-C                                    (b) CIFAR100-C

Figure 2: Masking types in classification error rate (%) under CTTA using ViT-B/16 backbone. Each subfigure shows random spatial masking (Patch/Pixel) and random frequency masking (All/Low/High). Corruption starts at Gaussian noise (GN) and progresses through 15 types of corruption in counter-clockwise order without model reset.

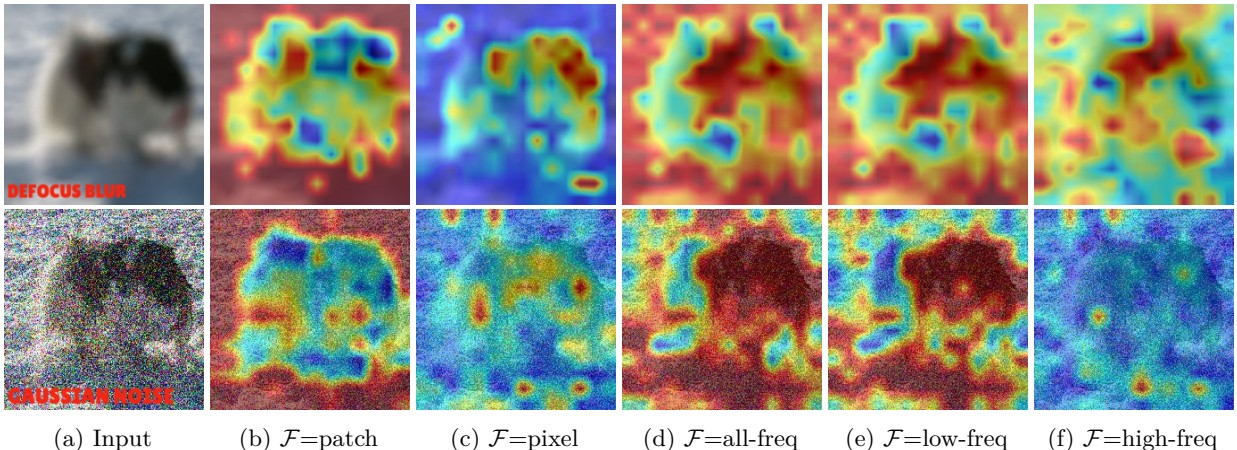

(a) Input    (b) $\mathcal{F}$=patch    (c) $\mathcal{F}$=pixel    (d) $\mathcal{F}$=all-freq    (e) $\mathcal{F}$=low-freq    (f) $\mathcal{F}$=high-freq

Figure 3: Class activation maps using GradCAM on ImageNet-C with severity 5 (highest). $\mathcal{F}$ indicates the masking family.

## 4.3 Family Comparison

### 4.3.1 Per-Corruption Family Profiles

Figure 2 compares all five $\mathcal{F}$ on CIFAR-10-C and CIFAR-100-C with $\mathcal{S}$=random held fixed. $\mathcal{F}$=patch achieves the lowest error on nearly every corruption. Here $\mathcal{F}$=patch is the default single contiguous Block-Patch. This is the main family comparison where the structural-preservation principle is most informative: contiguous block masks are more stable than fragmented spatial masks or narrowband frequency deletions. $\mathcal{F}$=high-freq sits near the random-chance ceiling across almost all corruptions: blur-type corruptions (defocus, glass, motion, zoom) act as low-pass filters that concentrate energy at the spectral center while attenuating the periphery (high frequencies), so removing the surviving high-frequency content leaves uninformative views. $\mathcal{F}$=pixel yields much higher error than patch, with pronounced spikes on blur, weather, and digital corruptions where scattering pixels injects noise the consistency loss cannot reconcile. $\mathcal{F}$=low-freq and $\mathcal{F}$=all-freq stay closer to patch on several corruptions but show sharp peaks on pixelate, corruption that preserve edge structure at the spectral periphery while degrading the low-frequency manifold, so masking low frequencies removes the remaining informative band.

Table 1: Classification error rate (%) under CTTA using ViT-B/16 backbone on three standard corruption benchmarks. Mean is the average across 15 corrupted domains. Gain is the relative improvement over the source model.

(a) CIFAR-10 → CIFAR-10-C

| Method | GN | SN | IN | DB | GB | MB | ZB | S | Fr | F | B | C | ET | P | JC | Mean↓ | Gain |
|---|---|---|---|---|---|---|---|---|---|---|---|---|---|---|---|---|---|
| Source (Dosovitskiy et al., 2021) | 60.1 | 53.2 | 38.3 | 19.9 | 35.5 | 22.6 | 18.6 | 12.1 | 12.7 | 22.8 | 5.3 | 49.7 | 23.6 | 24.7 | 23.1 | 28.2 | 0.0 |
| Pseudo-label (Lee, 2013) | 59.8 | 52.5 | 37.2 | 19.8 | 35.2 | 21.8 | 17.6 | 11.6 | 12.3 | 20.7 | 5.0 | 41.7 | 21.5 | 25.2 | 22.1 | 26.9 | +1.3 |
| Tent (Wang et al., 2021) | 57.7 | 56.3 | 29.4 | 16.2 | 35.3 | 16.2 | 12.4 | 11.0 | 11.6 | 14.9 | 4.7 | 22.5 | 15.9 | 29.1 | 19.5 | 23.5 | +4.7 |
| CoTTA (Wang et al., 2022) | 58.7 | 51.3 | 33.0 | 20.1 | 34.8 | 20 | 15.2 | 11.1 | 11.3 | 18.5 | 4.0 | 34.7 | 18.8 | 19.0 | 17.9 | 24.6 | +3.6 |
| VDP (Gan et al., 2023) | 57.5 | 49.5 | 31.7 | 21.3 | 35.1 | 19.6 | 15.1 | 10.8 | 10.3 | 18.1 | 4.0 | 27.5 | 18.4 | 22.5 | 19.9 | 24.1 | +4.1 |
| SAR (Niu et al., 2023) | 54.1 | 47.6 | 38.0 | 19.9 | 34.8 | 22.6 | 18.6 | 12.1 | 12.7 | 22.8 | 5.3 | 39.9 | 23.6 | 24.7 | 23.1 | 26.6 | +1.6 |
| PETAL (Brahma & Rai, 2023) | 59.9 | 52.3 | 36.1 | 20.1 | 34.7 | 19.4 | 14.8 | 11.5 | 11.2 | 17.8 | 4.4 | 29.6 | 17.6 | 19.2 | 17.3 | 24.4 | +3.8 |
| RoTTA (Yuan et al., 2023) | 60.3 | 53.5 | 34.7 | 16.5 | 32.1 | 15.0 | 10.8 | 9.5 | 10.0 | 10.5 | **3.5** | 12.5 | 12.9 | 27.3 | 18.4 | 21.8 | +6.4 |
| DeYO (Lee et al., 2024) | 41.5 | 23.5 | 20.2 | 15.1 | 31.0 | 15.0 | 10.1 | 9.1 | 8.1 | 12.8 | 4.6 | 15.2 | 15.9 | 18.9 | 17.5 | 17.2 | +10.9 |
| ROID (Marsden et al., 2024) | 26.6 | 20.7 | 15.4 | 10.6 | 28.0 | 11.8 | 10.1 | 9.3 | 9.5 | 11.2 | 4.5 | 11.9 | 15.7 | 15.8 | 18.1 | 14.6 | +13.6 |
| ViDA (Liu et al., 2024b) | 52.9 | 47.9 | 19.4 | 11.4 | 31.3 | 13.3 | 7.6 | 7.6 | 9.9 | 12.5 | 3.8 | 26.3 | 14.4 | 33.9 | 18.2 | 20.7 | +7.5 |
| Continual-MAE (Liu et al., 2024a) | 30.6 | 18.9 | 11.5 | 10.4 | 22.5 | 13.9 | 9.8 | 6.6 | 6.5 | 8.8 | 4.0 | 8.5 | 12.7 | 9.2 | 14.4 | 12.6 | +15.6 |
| SPA (Niu et al., 2025) | 56.8 | 45.7 | 29.9 | 16.7 | 33.1 | 17.9 | 13.5 | 11.5 | 11.0 | 17.6 | 4.6 | 23.9 | 19.7 | 30.0 | 20.8 | 23.5 | +4.7 |
| DPCore (Zhang et al., 2025b) | 22.0 | 18.2 | 14.9 | 14.3 | 24.4 | 13.9 | 12.0 | 11.6 | 10.7 | 15.0 | 5.7 | 21.8 | 15.6 | 12.7 | 18.0 | 15.4 | +12.8 |
| PAID (Wang et al., 2025a) | 22.9 | **11.8** | 9.9 | 9.1 | 16.7 | 10.8 | 7.4 | 7.4 | 6.6 | 11.4 | 4.5 | 9.3 | 12.8 | 9.4 | 14.5 | 11.0 | +17.1 |
| REM (Han et al., 2025) | 17.3 | 12.5 | 10.3 | 8.4 | 17.7 | 8.4 | 5.5 | 6.6 | 5.6 | 7.2 | 3.7 | 6.4 | 11.0 | **7.3** | 13.0 | 9.4 | +18.8 |
| M2A ($\mathcal{F}$=low-freq) | 16.8±0.2 | 11.8±0.2 | 13.0±0.6 | 11.2±0.4 | 24.5±0.2 | 11.5±0.4 | 8.0±0.4 | 7.7±0.1 | 6.7±0.4 | 6.7±0.2 | 5.2±0.1 | 5.8±0.1 | 20.1±0.4 | 15.1±1.4 | 20.6±0.1 | 12.3 | +15.9 |
| M2A ($\mathcal{F}$=patch) | 19.9±0.5 | 11.9±0.2 | 8.6±0.2 | 7.6±0.2 | 15.0±0.4 | 8.2±0.3 | 5.4±0.1 | 6.0±0.1 | 5.0±0.1 | 6.2±0.1 | 3.7±0.1 | 4.9±0.1 | 10.7±0.3 | 7.5±0.2 | 12.7±0.4 | 8.9 | +19.3 |

(b) CIFAR-100 → CIFAR-100-C

| Method | GN | SN | IN | DB | GB | MB | ZB | S | Fr | F | B | C | ET | P | JC | Mean↓ | Gain |
|---|---|---|---|---|---|---|---|---|---|---|---|---|---|---|---|---|---|
| Source (Dosovitskiy et al., 2021) | 55.0 | 51.5 | 26.9 | 24.0 | 60.5 | 29.0 | 21.4 | 21.1 | 25.0 | 35.2 | 11.8 | 34.8 | 43.2 | 56.0 | 35.9 | 35.4 | 0.0 |
| Pseudo-label (Lee, 2013) | 53.8 | 48.9 | 25.4 | 23.0 | 58.7 | 27.3 | 19.6 | 20.6 | 23.4 | 31.3 | 11.8 | 28.4 | 39.6 | 52.3 | 33.9 | 33.2 | +2.2 |
| Tent (Wang et al., 2021) | 53.0 | 47.0 | 24.6 | 22.3 | 58.5 | 26.5 | 19.0 | 21.0 | 23.0 | 30.1 | 11.8 | 25.2 | 39.0 | 47.1 | 33.3 | 32.1 | +3.3 |
| CoTTA (Wang et al., 2022) | 55.0 | 51.3 | 25.8 | 24.1 | 59.2 | 28.9 | 21.4 | 21.0 | 24.7 | 34.9 | 11.7 | 31.7 | 40.4 | 55.7 | 35.6 | 34.8 | +0.6 |
| VDP (Gan et al., 2023) | 54.8 | 51.2 | 25.6 | 24.2 | 59.1 | 28.8 | 21.2 | 20.5 | 23.3 | 33.8 | **7.5** | **11.7** | 32.0 | 51.7 | 35.2 | 32.0 | +3.4 |
| SAR (Niu et al., 2023) | 39.4 | 31.0 | 19.8 | 20.9 | 43.9 | 22.6 | 19.1 | 20.3 | 20.2 | 24.3 | 11.8 | 22.3 | 35.2 | 32.1 | 30.1 | 26.2 | +9.2 |
| PETAL (Brahma & Rai, 2023) | 49.2 | 38.7 | 24.1 | 26.3 | 38.2 | 25.4 | 19.4 | 21.0 | 19.3 | 26.6 | 15.4 | 31.8 | 28.3 | 26.6 | 29.5 | 28.0 | +7.4 |
| RoTTA (Yuan et al., 2023) | 54.1 | 48.4 | 23.6 | 21.3 | 51.2 | 23.5 | 16.5 | 18.5 | 19.5 | 24.1 | 11.4 | 20.4 | 33.8 | 31.7 | 31.2 | 28.6 | +6.8 |
| DeYO (Lee et al., 2024) | 33.1 | 26.9 | 17.4 | 19.1 | 35.4 | 20.1 | 15.8 | 18.0 | 15.9 | 19.5 | 12.2 | 14.9 | 30.8 | 22.8 | 29.3 | 22.1 | +13.3 |
| ROID (Marsden et al., 2024) | 31.7 | 28.6 | 17.4 | 19.7 | 34.4 | 20.2 | 16.5 | 18.0 | 17.2 | 20.6 | 11.0 | 17.9 | 28.4 | 23.0 | 27.6 | 22.1 | +13.3 |
| ViDA (Liu et al., 2024b) | 50.1 | 40.7 | 22.0 | 21.2 | 45.2 | 21.6 | 16.5 | 17.9 | 16.6 | 25.6 | 11.5 | 29.0 | 29.6 | 34.7 | 27.1 | 27.3 | +8.1 |
| Continual-MAE (Liu et al., 2024a) | 48.6 | 30.7 | 18.5 | 21.3 | 38.4 | 22.2 | 17.5 | 19.3 | 18.0 | 24.8 | 13.1 | 27.8 | 31.4 | 35.5 | 29.5 | 26.4 | +9.0 |
| SPA (Niu et al., 2025) | 42.1 | 34.7 | 19.3 | 22.1 | 50.9 | 23.6 | 18.8 | 22.4 | 19.8 | 26.0 | 12.8 | 18.0 | 42.1 | 43.5 | 38.2 | 29.0 | +6.4 |
| DPCore (Zhang et al., 2025b) | 48.2 | 40.2 | 21.3 | 20.2 | 44.1 | 21.1 | 16.2 | 18.1 | **15.2** | 22.3 | 9.4 | 13.2 | 28.6 | 32.8 | **25.5** | 25.1 | +10.3 |
| PAID (Wang et al., 2025a) | 40.7 | 31.9 | 20.4 | 19.8 | 35.9 | 23.0 | 16.3 | 20.5 | 18.2 | 25.3 | 12.6 | 19.8 | 29.4 | 28.2 | 31.3 | 24.9 | +10.5 |
| REM (Han et al., 2025) | **29.2** | 25.5 | 17.0 | 19.1 | 35.2 | 21.2 | 18.3 | 19.5 | 18.7 | 22.8 | 15.5 | 17.6 | 31.6 | 26.2 | 33.0 | 23.4 | +12.0 |
| M2A ($\mathcal{F}$=low-freq) | 30.6±0.3 | 28.0±0.6 | 20.8±0.1 | 24.6±0.3 | 64.5±9.6 | 28.0±1.6 | 22.0±0.5 | 18.4±0.5 | 16.8±0.2 | 17.7±0.4 | 14.6±0.1 | 16.2±0.2 | 48.6±9.6 | 62.2±21.4 | 47.0±5.9 | 30.7 | +4.7 |
| M2A ($\mathcal{F}$=patch) | 29.7±0.8 | 24.6±1.0 | 15.7±0.4 | 18.1±0.2 | 30.6±0.5 | 19.0±0.2 | 15.3±0.1 | 16.2±0.2 | 15.2±0.3 | 16.8±0.1 | 11.7±0.2 | 14.0±0.3 | 25.7±0.6 | 18.9±0.1 | 26.1±0.3 | 19.8 | +15.6 |

(c) ImageNet → ImageNet-C

| Method | GN | SN | IN | DB | GB | MB | ZB | S | Fr | F | B | C | ET | P | JC | Mean↓ | Gain |
|---|---|---|---|---|---|---|---|---|---|---|---|---|---|---|---|---|---|
| Source (Dosovitskiy et al., 2021) | 53.0 | 51.8 | 52.1 | 68.5 | 78.8 | 58.5 | 63.3 | 49.9 | 54.2 | 57.7 | 26.4 | 91.4 | 57.5 | 38.0 | 36.2 | 55.8 | 0.0 |
| Pseudo-label (Lee, 2013) | 45.2 | 40.4 | 41.6 | 51.3 | 53.9 | 45.6 | 47.7 | 40.4 | 45.7 | 93.8 | 98.5 | 99.9 | 99.9 | 98.9 | 99.6 | 61.2 | -5.4 |
| Tent (Wang et al., 2021) | 52.2 | 48.9 | 49.2 | 65.8 | 73.0 | 54.5 | 58.4 | 44.0 | 47.7 | 50.3 | 23.9 | 72.8 | 55.7 | 34.4 | 33.9 | 51.0 | +4.8 |
| CoTTA (Wang et al., 2022) | 52.9 | 51.6 | 51.4 | 68.3 | 78.1 | 57.1 | 62.0 | 48.2 | 52.7 | 55.3 | 25.9 | 90.0 | 56.4 | 36.4 | 35.2 | 54.8 | +1.0 |
| VDP (Gan et al., 2023) | 52.7 | 51.6 | 50.1 | 58.1 | 70.2 | 56.1 | 58.1 | 42.1 | 46.1 | 45.8 | 23.6 | 70.4 | 54.5 | 36.4 | 36.1 | 50.0 | +5.8 |
| SAR (Niu et al., 2023) | 49.3 | 43.8 | 44.9 | 58.2 | 60.9 | 46.1 | 51.8 | 41.3 | 44.1 | 41.8 | 23.8 | 57.2 | 49.9 | 32.9 | 32.7 | 45.2 | +10.6 |
| PETAL (Brahma & Rai, 2023) | 52.1 | 48.2 | 47.5 | 66.8 | 74.0 | 56.7 | 59.7 | 46.8 | 47.2 | 52.7 | 26.4 | 91.3 | 50.7 | 32.0 | 32.0 | 52.3 | +3.5 |
| RoTTA (Yuan et al., 2023) | 49.8 | 47.6 | 47.6 | 67.0 | 68.8 | 53.3 | 58.3 | 43.4 | 47.0 | 51.9 | 24.1 | 82.7 | 51.3 | 35.7 | 33.6 | 50.8 | +5.0 |
| DeYO (Lee et al., 2024) | 47.2 | 39.4 | 40.2 | 54.7 | 53.8 | 42.5 | 49.1 | 39.3 | 40.0 | 39.8 | 22.8 | 48.9 | 44.9 | 32.3 | 30.5 | 41.7 | +14.1 |
| ROID (Marsden et al., 2024) | 43.7 | 37.7 | 39.3 | 49.0 | 49.4 | 40.8 | 45.2 | 36.1 | 37.8 | 36.0 | **21.3** | 49.3 | 41.2 | 30.5 | 30.0 | 39.2 | +16.6 |
| ViDA (Liu et al., 2024b) | 47.7 | 42.5 | 42.9 | 52.2 | 56.9 | 45.5 | 48.9 | 38.9 | 42.7 | 40.7 | 24.3 | 52.8 | 49.1 | 33.5 | 33.1 | 43.4 | +12.4 |
| Continual-MAE (Liu et al., 2024a) | 46.3 | 41.9 | 42.5 | 51.4 | 54.9 | 43.3 | **40.7** | **34.2** | 36.3 | 64.3 | 23.4 | 60.3 | **37.5** | 29.2 | 31.4 | 43.3 | +13.3 |
| SPA (Niu et al., 2025) | 43.9 | 37.4 | 38.8 | 54.9 | **43.8** | 40.7 | 45.2 | 36.5 | 34.8 | 36.6 | 22.5 | 59.5 | 45.3 | **27.8** | 28.8 | 39.8 | +16.0 |
| DPCore (Zhang et al., 2025b) | 42.2 | 38.7 | 39.3 | **47.2** | 51.4 | 47.7 | 46.9 | 39.3 | 36.9 | 37.4 | 22.0 | 44.4 | 45.1 | 30.9 | 29.6 | 39.9 | +15.9 |
| PAID (Wang et al., 2025a) | 48.8 | 43.7 | 44.4 | 49.4 | 49.6 | 47.3 | 44.2 | 37.5 | 39.4 | 42.1 | 25.2 | 50.0 | 39.8 | 33.5 | 36.5 | 42.2 | +13.6 |
| REM (Han et al., 2025) | 43.5 | 38.1 | 39.2 | 53.2 | 49.0 | 43.5 | 42.8 | 37.5 | 35.2 | 35.4 | 23.2 | 46.8 | 41.6 | 28.9 | 30.2 | 39.2 | +16.6 |
| M2A ($\mathcal{F}$=low-freq) | 44.1±0.5 | 39.7±0.0 | 41.4±0.3 | 62.7±0.4 | 51.8±0.6 | 48.7±0.0 | 66.0±5.0 | 37.9±1.7 | 35.9±0.6 | 30.7±0.1 | 23.5±0.1 | 51.6±1.0 | 44.7±1.2 | 35.9±0.5 | 36.3±0.6 | 43.4 | +12.4 |
| M2A ($\mathcal{F}$=patch) | 41.9±0.3 | 35.9±0.1 | 36.6±0.1 | 52.2±0.3 | 45.4±0.4 | 40.2±0.2 | 42.5±0.2 | 35.5±0.2 | 34.6±0.5 | 34.2±0.3 | 21.4±0.1 | 44.2±0.2 | 39.0±0.8 | 28.4±0.3 | 29.3±0.0 | 37.4 | +18.4 |

### 4.3.2 Class Activation Map

Figure 3 visualizes GradCAM activations under defocus blur and Gaussian noise. $\mathcal{F}$=patch produces tightly focused activations with clear object–background separation even under severe corruption. $\mathcal{F}$=pixel yields broader coverage but weaker localization, with scattered hot spots drifting onto the background. Among frequency families, $\mathcal{F}$=low-freq and $\mathcal{F}$=all-freq retain a rough object silhouette, particularly under Gaussian noise, but leak substantial activation into the background. $\mathcal{F}$=high-freq is the most diffuse: under Gaussian noise the CAM becomes nearly indistinguishable from noise, as removing high-frequency detail on top of noise corruption leaves almost no discriminative structure. These visualizations broadly mirror the family-level ranking in Figure 2.

Table 2: Domain generalization (forward transfer) under CTTA using ViT-B/16 backbone. Models adapt on 10 seen corruptions and are then evaluated without further adaptation on 5 unseen corruptions. Fine-grained per-corruption results with mean ± standard deviation over seeds are provided in Appendix A.8.4 (Table 11).

(a) CIFAR10-C

| Method | Directly test on unseen domains | | | | | Mean↓ | Gain |
|---|---|---|---|---|---|---|---|
| | B | C | ET | P | JC | | |
| Source | 5.3 | 49.7 | 23.6 | **24.7** | 23.1 | 25.2 | 0.0 |
| Tent | 5.1 | 34.1 | 22.1 | 27.7 | 22.4 | 22.2 | +3.0 |
| CoTTA | 20.4 | 83.5 | 52.1 | 35.3 | 44.0 | 47.0 | −21.8 |
| SAR | 5.3 | 49.2 | 23.5 | 24.8 | 23.1 | 25.1 | +0.1 |
| RoTTA | **3.9** | 8.0 | 14.3 | 32.3 | 21.4 | 15.9 | +9.3 |
| DeYO | 4.7 | 18.6 | 16.0 | 26.1 | **18.4** | 16.7 | +8.5 |
| ROID | 4.7 | 16.1 | 18.2 | 35.3 | 21.9 | 19.2 | +6.0 |
| REM | 5.5 | 13.7 | **13.8** | 24.9 | 22.4 | 16.0 | +9.2 |
| M2A($\mathcal{F}$=low-freq) | 6.4 | 12.0 | 18.9 | 36.3 | 25.2 | 19.8 | +5.4 |
| M2A($\mathcal{F}$=high-freq) | 89.9 | 90.0 | 90.0 | 90.0 | 90.0 | 90.0 | −64.8 |
| M2A($\mathcal{F}$=all-freq) | 6.5 | 11.7 | 18.8 | 34.5 | 24.7 | 19.2 | +6.0 |
| M2A($\mathcal{F}$=pixel) | 33.9 | 36.6 | 41.6 | 50.8 | 46.7 | 41.9 | -16.7 |
| M2A($\mathcal{F}$=patch) | 5.1 | **7.5** | **13.8** | 31.0 | 20.7 | **15.6** | +9.6 |

(b) ImageNet-C

| Method | Directly test on unseen domains | | | | | Mean↓ | Gain |
|---|---|---|---|---|---|---|---|
| | B | C | ET | P | JC | | |
| Source | 24.8 | 91.1 | 55.7 | 39.1 | 36.7 | 49.4 | 0.0 |
| Tent | 23.4 | 88.5 | 51.7 | 35.4 | 35.8 | 46.9 | +2.5 |
| CoTTA | 24.5 | 90.2 | 54.5 | 37.5 | 34.8 | 48.3 | +1.2 |
| SAR | 23.4 | 79.7 | 48.4 | **35.2** | 33.8 | 44.1 | +5.4 |
| RoTTA | 24.4 | 64.6 | 49.3 | 36.6 | **33.4** | 41.7 | +7.8 |
| DeYO | 23.6 | 74.8 | **48.3** | 37.0 | 35.3 | 43.8 | +5.7 |
| ROID | **22.3** | 75.6 | 49.7 | 37.4 | 34.8 | 43.9 | +5.5 |
| REM | 23.6 | 60.2 | 51.6 | 37.2 | 37.3 | 41.9 | +7.5 |
| M2A($\mathcal{F}$=low-freq) | 25.4 | 65.0 | 56.2 | 43.5 | 39.9 | 46.0 | +3.5 |
| M2A($\mathcal{F}$=high-freq) | 91.3 | 99.6 | 96.6 | 95.0 | 93.1 | 95.1 | -45.7 |
| M2A($\mathcal{F}$=all-freq) | 26.1 | 65.4 | 59.2 | 44.5 | 40.4 | 47.1 | +2.4 |
| M2A($\mathcal{F}$=pixel) | 98.6 | 99.8 | 99.8 | 99.4 | 98.9 | 99.3 | -49.8 |
| M2A($\mathcal{F}$=patch) | 22.8 | **56.3** | 51.5 | 38.4 | **37.2** | **41.3** | +8.2 |

Table 3: Lifelong adaptation on ImageNet-C under CTTA. Each pass consists of 15 corruptions evaluated sequentially without resetting the model; entries report mean classification error rate (%) per pass.

| Method | Pass 1 | Pass 2 | Pass 3 | Pass 4 | Pass 5 | Pass 6 | Pass 7 | Pass 8 | Pass 9 | Pass 10 | Mean↓ | Gain |
|---|---|---|---|---|---|---|---|---|---|---|---|---|
| Source | 56.2 | 56.2 | 56.2 | 56.2 | 56.2 | 56.2 | 56.2 | 56.2 | 56.2 | 56.2 | 56.2 | 0.0 |
| Tent | 51.2 | 47.1 | 45.7 | 45.2 | 44.8 | 44.7 | 44.5 | 44.3 | 44.3 | 44.2 | 45.6 | +10.6 |
| CoTTA | 55.6 | 54.7 | 54.1 | 53.7 | 53.8 | 54.2 | 54.2 | 54.5 | 54.4 | 54.5 | 54.4 | +1.8 |
| SAR | 43.4 | 40.4 | 40.1 | 39.7 | 39.5 | 39.3 | 39.3 | 39.2 | 39.2 | 39.1 | 39.9 | +16.3 |
| RoTTA | 50.5 | 45.9 | 44.1 | 43.2 | 42.9 | 42.6 | 42.4 | 42.3 | 42.2 | 42.2 | 43.9 | +12.3 |
| DeYO | 41.7 | 39.7 | 39.3 | 39.0 | 38.8 | 38.8 | 38.7 | 38.7 | 38.7 | 38.6 | 39.2 | +17.0 |
| ROID | 39.0 | 39.6 | 39.5 | 39.6 | 38.9 | 39.2 | 39.1 | 39.0 | 39.1 | 38.8 | 39.2 | +17.0 |
| REM | 39.2 | 38.1 | 37.5 | 37.1 | 36.9 | 36.9 | 36.9 | 36.9 | 36.9 | 36.9 | 37.3 | +18.9 |
| M2A($\mathcal{F}$=low-freq) | 43.6±0.4 | 42.7±2.7 | 55.8±25.3 | 60.9±33.8 | 61.9±33.0 | 63.5±31.6 | 66.1±29.6 | 76.1±23.2 | 84.7±18.9 | 91.7±14.0 | 64.7 | -8.5 |
| M2A($\mathcal{F}$=patch) | **37.4±0.1** | **35.3±0.1** | **34.6±0.1** | **34.2±0.0** | **33.9±0.1** | **33.7±0.0** | **33.5±0.0** | **33.5±0.1** | **33.4±0.1** | **33.3±0.1** | **34.3** | +21.9 |

### 4.3.3 Standard Benchmarks

Table 1 quantifies the family gap at scale across three benchmarks of increasing difficulty, establishing the baseline design recommendation: $\mathcal{F}$=patch achieves the lowest mean error on all three, matching or surpassing REM ($\mathcal{S}$=attention) and Continual-MAE ($\mathcal{S}$=uncertainty), notably, these baselines also differ from M2A in loss functions and auxiliary components, so this comparison provides suggestive context rather than a controlled quantification of $\mathcal{S}$. The benchmark scaling also reveals when frequency masking becomes unreliable: $\mathcal{F}$=low-freq is competitive on the simpler benchmark (CIFAR10-C) but degrades sharply on the harder ones, exhibiting severe spikes on blur (low-pass filters) and digital corruptions (JPEG compression attenuates high frequencies, pixelate preserves edges).

### 4.3.4 Domain Generalization

Table 2 asks a different question from the preceding benchmarks: do adapted representations transfer to unseen corruptions? Models adapt on ten seen corruptions and are tested without further updates on five held-out ones. $\mathcal{F}$=patch achieves the lowest mean error on both CIFAR-10-C and ImageNet-C, slightly surpassing REM and other baselines on held-out domains. The transferability gap across families is informative: $\mathcal{F}$=high-freq and $\mathcal{F}$=pixel collapse to near-chance accuracy on every unseen corruption, indicating that their adapted representations carry almost no transferable structure, $\mathcal{F}$=high-freq strips the edges and textures that define object boundaries, while $\mathcal{F}$=pixel scatters perturbations too finely to force reliance on global semantics. $\mathcal{F}$=low-freq and $\mathcal{F}$=all-freq avoid complete collapse and improve over the unadapted source but remain well above $\mathcal{F}$=patch, suggesting that removing low-frequency content provides some regularization without building features that generalize as reliably to novel corruption types.

### 4.4 Lifelong Adaptation

Table 3 asks whether frequency masking's failures are self-correcting or irreversible: ten sequential passes through all ImageNet-C corruptions with no parameter reset. $\mathcal{F}$=patch achieves the lowest mean error and improves monotonically from the first pass to the tenth, accumulating useful structure rather than forgetting it. Most baselines plateau earlier; REM converges several points above patch masking. For M2A

Table 4: Continual Dynamic Change (CDC) on ImageNet-C under CTTA. Entries report classification error rate (%) per corruption type.

| Method | GN | SN | IN | DB | GB | MB | ZB | S | Fr | F | B | C | ET | P | JC | Mean↓ | Gain |
|---|---|---|---|---|---|---|---|---|---|---|---|---|---|---|---|---|---|
| Source | 54.3 | 53.3 | 53.6 | 68.5 | 78.6 | 58.3 | 63.0 | 50.5 | 54.5 | 57.8 | 26.7 | 91.3 | 57.5 | 38.2 | 36.4 | 56.2 | 0.0 |
| Tent | 51.9 | 50.7 | 49.4 | 60.2 | 71.7 | 53.8 | 57.0 | 46.0 | 49.5 | 49.5 | 24.1 | 82.4 | 55.6 | 36.2 | 34.8 | 51.5 | +4.7 |
| Cotta | 49.0 | 48.1 | 47.5 | 66.9 | 71.8 | 54.7 | 60.5 | 46.7 | 51.4 | 55.6 | 24.3 | 91.9 | 54.7 | 38.4 | 35.7 | 53.1 | +3.1 |
| SAR | 45.1 | 42.8 | 43.1 | 48.9 | 54.5 | 45.9 | 51.0 | 39.8 | 42.5 | 38.9 | 22.5 | 62.8 | 49.3 | 33.8 | 31.7 | 43.5 | +12.7 |
| Rotta | 47.5 | 46.7 | 45.2 | 62.3 | 64.6 | 51.2 | 54.9 | 43.8 | 48.4 | 50.9 | 24.0 | 87.6 | 52.0 | 37.3 | 34.5 | 50.1 | +6.1 |
| DeYO | 43.8 | 41.5 | 42.7 | 49.2 | 51.6 | 44.8 | 50.6 | 38.6 | 41.1 | 38.9 | 22.5 | 58.4 | 47.0 | 32.9 | 31.0 | 42.3 | +13.9 |
| ROID | 43.6 | 40.6 | 43.4 | 49.0 | 52.4 | 45.2 | 47.4 | 37.8 | 40.5 | 34.8 | **21.7** | 56.2 | 43.9 | 32.9 | 31.1 | 41.4 | +14.8 |
| DPCore | 42.7 | 40.4 | 42.2 | 57.4 | 61.0 | 51.1 | 52.4 | **35.8** | 41.0 | 38.8 | 22.1 | 55.4 | 48.3 | 31.1 | **28.1** | 43.2 | +13.0 |
| REM | 44.1 | 41.4 | 43.1 | 49.4 | 49.1 | 46.5 | 45.3 | 38.1 | 37.0 | 35.9 | 23.4 | 53.0 | 42.7 | **30.2** | 30.9 | 40.7 | +15.5 |
| M2A($\mathcal{F}$=low-freq) | 43.6 | 42.0 | 43.3 | 53.9 | 49.5 | 51.6 | 61.9 | **35.8** | **36.7** | 30.3 | 23.5 | 55.9 | 42.1 | 33.3 | 35.1 | 42.6 | +13.6 |
| M2A($\mathcal{F}$=patch) | **40.9** | **38.9** | **39.3** | **47.2** | **45.5** | **43.3** | **43.1** | 36.7 | **36.7** | 33.0 | 22.0 | **50.0** | 40.4 | **30.2** | 29.5 | **38.5** | +17.7 |

with $\mathcal{F}$=low-freq, error remains moderate on the first two passes but then increases sharply from the third onward, with later passes climbing into the 80–90% range, well above the source. The family gap widens with each pass, providing direct evidence that frequency masking's instability compounds over time rather than attenuating.

## 4.5 Continual Dynamic Change

Table 4 reveals *why* the compounding occurs, by examining per-corruption behavior under non-stationarity on ImageNet-C: corruption types change within the stream without resets. $\mathcal{F}$=patch maintains stable performance across all fifteen corruption types. $\mathcal{F}$=low-freq spikes severely on zoom blur, a low-pass filter that attenuates the spectral periphery, so additionally removing low-frequency structure leaves the model with nearly content-free views. This corruption-dependent spiking validates the structural-preservation principle: because each corruption has a distinct signature (blur concentrates power centrally, noise distributes energy across the full spectrum, digital artifacts introduce structured aliases), a fixed frequency band will inevitably overlap with some damage zone.

## 4.6 Architecture Scoping

Table 5: Experiment on other model agnostic methods with other architectures in ImageNet-C under CTTA. Entries report mean classification error (%) across 15 corruptions.

| Method | ResNext-50 | WideResNet-50 | ResNet-50 | ConvNext-B | ConvNext-L | ViT-B/16 | ViT-L/16 | Mean↓ | Gain |
|---|---|---|---|---|---|---|---|---|---|
| Source | 78.4 | 78.5 | 81.9 | 48.3 | 43.8 | 55.8 | 44.6 | 61.6 | — |
| Tent (Wang et al., 2021) | 58.4 | 57.8 | 62.5 | 38.6 | 45.1 | 51.0 | 40.1 | 50.5 | +11.1 |
| CoTTA (Wang et al., 2022) | 59.5 | 58.0 | 62.7 | 53.7 | 62.5 | 54.8 | 44.3 | 56.5 | +5.1 |
| SAR (Niu et al., 2023) | 58.0 | 57.9 | 61.9 | 39.1 | 36.3 | 45.2 | 41.0 | 48.5 | +13.1 |
| RoTTA (Yuan et al., 2023) | 64.2 | 62.7 | 67.1 | 46.4 | 50.6 | 50.8 | 40.9 | 54.7 | +6.9 |
| DeYO (Lee et al., 2024) | 56.1 | 54.4 | 59.8 | 37.3 | 41.8 | 41.7 | 41.8 | 47.6 | +14.0 |
| ROID (Marsden et al., 2024) | **51.0** | **50.7** | **54.4** | 58.1 | 39.4 | 39.2 | 41.6 | 47.8 | +13.8 |
| M2A($\mathcal{F}$=low-freq) | 58.7 | 58.6 | 62.2 | 87.6 | 85.3 | 44.0 | 35.3 | 61.7 | −0.1 |
| M2A($\mathcal{F}$=patch) | 61.7 | 61.5 | 64.9 | **34.2** | **30.1** | **37.4** | **32.1** | **46.0** | +15.6 |

Table 5 tests whether the family ranking generalizes across architectures, revealing architecture-dependent design recommendations. We evaluate seven architectures: traditional CNNs (ResNet-50, ResNeXt-50, WideResNet-50), modern ConvNets (ConvNeXt-B/L), and vision transformers (ViT-B/16, ViT-L/16). On patch-tokenized architectures, $\mathcal{F}$=patch achieves the lowest error by a substantial margin. On traditional CNNs, patch masking underperforms because overlapping convolutions partially "see through" patch boundaries, making the family choice less consequential on these architectures—a result consistent with the structural-preservation principle (Section 5). $\mathcal{F}$=low-freq achieves the strongest result on ViT-L/16, suggesting that larger transformer capacity can absorb frequency masking's global perturbation, a positive condition for frequency masking, yet collapses on ConvNeXt, where the patch stem amplifies spectral distortion. This is one of the main boundary cases for the principle: the advantage of patch masking is strongest on patch-tokenized ViTs, whereas low-frequency masking can become competitive when task structure and model capacity are more tolerant of global perturbations. The practical guidance is architecture-specific: prefer patch masking on ViTs; on CNNs the family gap vanishes and the choice is less critical.

Table 6: Model efficiency under CTTA using ViT-B/16 backbone (86M parameters). Error is averaged over 15 corruptions and total time is computed after adapting to 15 domain corruptions. We use a single AMD Instinct MI200 GPU.

(a) CIFAR10-C

| Method | Params (%) | Time (min) | Pass(es) | Error ↓ |
|---|---|---|---|---|
| Source | 0 | 12.6 | 1 | 28.2 |
| Tent | 0.045 | 30.4 | 1 | 23.5 |
| CoTTA | 100.0 | 379 | 3–35 | 24.6 |
| SAR | 0.032 | 60.0 | 2 | 26.6 |
| RoTTA | 0.045 | 57.6 | 1–3 | 21.8 |
| DeYO | 0.032 | 42.7 | 2 | 17.2 |
| ROID | 0.045 | 47.9 | 2 | 14.6 |
| Continual-MAE | 100.0 | - | 12 | 12.6 |
| REM | 0.032 | 83.0 | 3 | 9.4 |
| M2A($\mathcal{F}$=low-freq) | 0.032 | 90.0 | 3 | 12.3 |
| M2A($\mathcal{F}$=patch) | 0.032 | 90.8 | 3 | 8.9 |

(b) ImageNet-C

| Method | Params (%) | Time (min) | Pass(es) | Error ↓ |
|---|---|---|---|---|
| Source | 0 | 2.3 | 1 | 55.8 |
| Tent | 0.045 | 4.7 | 1 | 51.0 |
| CoTTA | 100.0 | 28.4 | 3–35 | 54.8 |
| SAR | 0.032 | 8.7 | 2 | 45.2 |
| RoTTA | 0.045 | 11.7 | 1–3 | 50.8 |
| DeYO | 0.032 | 8.1 | 2 | 41.7 |
| ROID | 0.045 | 8.7 | 2 | 39.2 |
| Continual-MAE | 100.0 | - | 12 | 42.5 |
| REM | 0.032 | 11.5 | 3 | 39.2 |
| M2A($\mathcal{F}$=low-freq) | 0.032 | 13.7 | 3 | 43.4 |
| M2A($\mathcal{F}$=patch) | 0.032 | 13.6 | 3 | 37.4 |

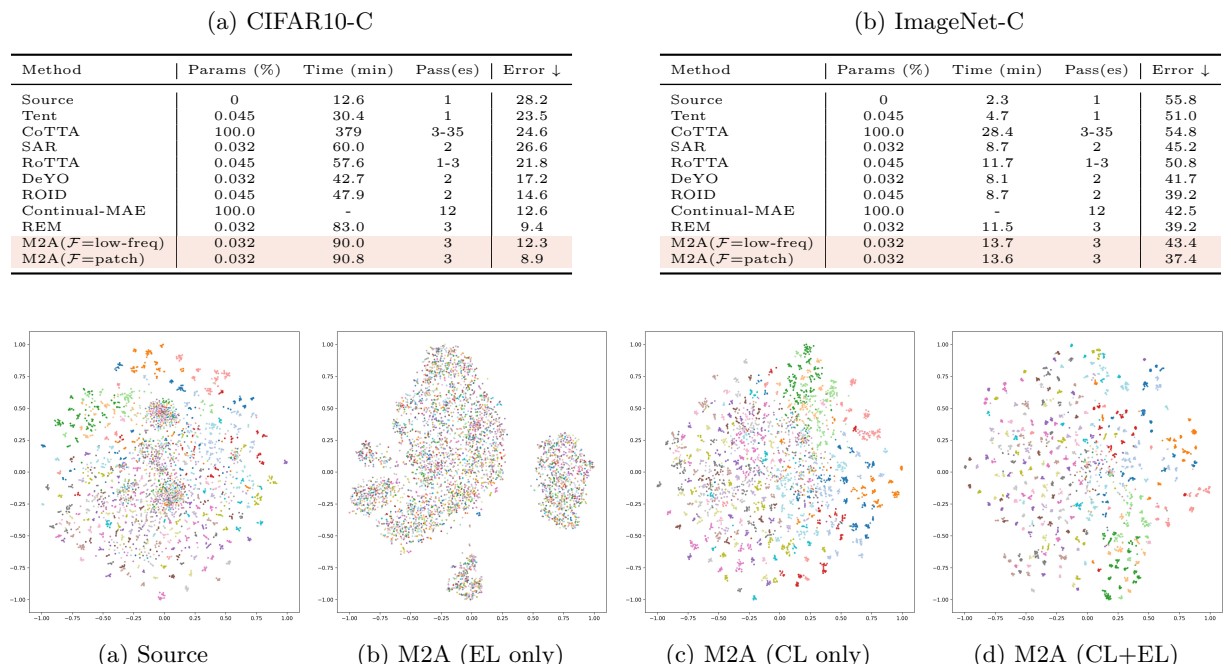

(a) Source          (b) M2A (EL only)          (c) M2A (CL only)          (d) M2A (CL+EL)

Figure 4: t-SNE visualizations of forward features from the adapted ViT-B/16 backbone on ImageNet-C under CTTA using M2A($\mathcal{F}$=patch).

### 4.6.1 Efficiency

Table 6 reports the computational cost of each method. M2A updates fewer than one tenth of a percent of the model's parameters and requires only three passes per batch, the same count as REM ($\mathcal{S}$=attention) but far fewer than Continual-MAE ($\mathcal{S}$=uncertainty, twelve passes) or CoTTA (up to thirty-five). Fixing $\mathcal{S}$=random eliminates the extra forward passes that heuristic scoring demands, yet M2A with $\mathcal{F}$=patch still achieves the best mean error on both benchmarks, though this system-level comparison is suggestive context given the confounding differences beyond $\mathcal{S}$. Wall-clock time is slightly higher than REM's, reflecting mask-generation overhead, but modest compared to CoTTA's large time budget. $\mathcal{F}$=low-freq consumes identical compute (same parameter count, passes, and wall-clock time) yet returns substantially worse accuracy, confirming that the performance gap tracks $\mathcal{F}$, not computational budget.

### 4.6.2 Feature Space

Figure 4 shows t-SNE projections of adapted ViT-B/16 features on ImageNet-C under $\mathcal{F}$=patch, comparing the source model, entropy loss only, consistency loss only, and their combination. The source model produces heavily intermixed class clusters with poor separation under corruption. Entropy minimization alone makes the picture worse: it collapses features into mixed-class blobs, indicating that confident predictions without cross-view agreement push representations toward degenerate modes rather than class-discriminative structure. Consistency loss alone improves local per-class coherence at the periphery but does not produce globally separated clusters. Combining both yields the clearest class separation, with distinct per-class groups spread across the embedding space, confirming complementary roles: consistency enforces mask-invariant structure, entropy sharpens class boundaries. Some classes remain partially overlapping, indicating that $\mathcal{F}$=patch does not fully resolve fine-grained confusion on the hardest corruptions.

Table 7: Mean classification error (%) of MRSFFIA-C across 15 corruptions under CTTA with severities 1, 3, and 5. Model is reset back to source weights for every corruption severity. The source model achieved accuracies of 90.2% on ViT-B/16 and 91.0% on ViT-L/16 on the clean test set. Dataset and implementation details are provided in the appendix.

(a) ViT-B/16

| Method | Sev. 1 | Sev. 3 | Sev. 5 | Mean | Gain |
|---|---|---|---|---|---|
| Source | 23.8 | 36.3 | 50.9 | 37.0 | 0.0 |
| Tent | 23.6 | 36.2 | 51.1 | 36.9 | +0.1 |
| CoTTA | 23.8 | 36.3 | 50.9 | 37.0 | 0.0 |
| SAR | 23.6 | 36.4 | 50.8 | 36.9 | +0.1 |
| DeYO | 23.4 | 36.7 | 50.9 | 37.0 | 0.0 |
| ROID | 24.1 | 37.9 | 52.7 | 38.2 | −1.2 |
| REM | 28.1 | 39.4 | 60.5 | 42.7 | −5.7 |
| M2A($\mathcal{F}$=low-freq) | 24.4 | 40.6 | 54.7 | 39.9 | −2.9 |
| M2A($\mathcal{F}$=patch) | **20.9** | **33.7** | **47.7** | **34.1** | **+2.9** |

(b) ViT-L/16

| Method | Sev. 1 | Sev. 3 | Sev. 5 | Mean | Gain |
|---|---|---|---|---|---|
| Source | 17.2 | 31.1 | 48.5 | 32.2 | 0.0 |
| Tent | 15.9 | 30.7 | 49.2 | 31.9 | +0.3 |
| CoTTA | 16.6 | 30.9 | 48.9 | 32.1 | +0.1 |
| SAR | 16.6 | 30.8 | 49.9 | 32.4 | −0.2 |
| DeYO | 15.2 | 30.6 | 48.3 | 31.4 | +0.8 |
| ROID | 15.4 | 29.9 | **47.6** | 30.9 | +1.3 |
| REM | 29.1 | 43.4 | 60.3 | 44.3 | −12.1 |
| M2A($\mathcal{F}$=low-freq) | **13.4** | **24.1** | 48.1 | **28.5** | **+3.7** |
| M2A($\mathcal{F}$=patch) | 13.5 | 26.0 | 48.9 | 29.5 | +2.7 |

(a) Lambda $\lambda$    (b) Mask views    (c) Mask steps    (d) Gradient steps    (e) Batch size

Figure 5: M2A($\mathcal{F}$=patch) ablations on ImageNet-C under CTTA. Each panel shows the effect of a single hyperparameter: (a) Lambda $\lambda$, (b) mask views, (c) mask steps, (d) gradient steps, and (e) batch size.

### 4.6.3 Use Case: Aquaculture

Table 7 applies M2A to MRSFFIA-C (Du et al., 2024), an aquaculture fish feeding behavior dataset with four feeding-intensity classes under 15 corruptions at severities one, three, and five. This experiment identifies a concrete condition under which frequency masking is preferable: when discriminative cues are global rather than spatially localized. $\mathcal{F}$=patch attains the lowest mean error on ViT-B/16, but on ViT-L/16 $\mathcal{F}$=low-freq achieves the strongest result and essentially matches $\mathcal{F}$=patch, while on ViT-B/16 it degrades below the source. Class distinctions in MRSFFIA-C depend on global appearance cues (feeding posture, turbidity), so $\mathcal{F}$=low-freq perturbations that alter coarse structure while preserving edges are informative when backbone capacity is sufficient; on smaller backbones the perturbation overwhelms the adaptation signal, but the ViT-L/16 result demonstrates that frequency masking can be the better choice when task structure and backbone capacity align. This is the clearest task-level boundary case in our study: when cues are global and capacity is sufficient, low-frequency masking can be competitive rather than pathological. REM ($\mathcal{S}$=attention) performs *worse* than the source on both backbones, providing suggestive evidence that family choice may dominate selection strategy in these settings. Improvements over the source are modest at the strongest severity, indicating that no family fully resolves the hardest fine-grained cases. The guidance for aquaculture is task-specific: when cues are global and capacity sufficient, frequency masking is a viable or preferable alternative to patch masking. Further details are in the Appendix.

### 4.6.4 Ablation Study

Figure 5 ablates five hyperparameters of M2A($\mathcal{F}$=patch) on ImageNet-C. The entropy loss weight $\lambda$ is the most critical: removing it ($\lambda$=0) causes catastrophic spikes, but any positive value yields stable performance, confirming that entropy minimization is essential while its precise magnitude is not. The number of views has a sweet spot at three: a single view is slightly worse, while five views destabilizes adaptation on individual corruptions, suggesting too many hard views push the model past a stability threshold. The masking-ratio schedule is robust, doubling or tripling $\alpha$ produces nearly identical profiles, indicating that a graded easy-to-hard progression matters more than any particular ratio. In particular, we explicitly varied both the masking ratios (0–10–20%, 0–20–40%, 0–30–60%) and the number of views (1, 3, 5), showing that our conclusions hold across a broad range of masking strengths. Additional gradient steps reduce error with diminishing returns beyond two; batch sizes from 4 to 128 keep performance in a similar range, with small variations consistent

with the trade-off between gradient noise at low data and slower adaptation at very large batches. Overall, $\mathcal{F}$=patch with $\mathcal{S}$=random is stable across a wide hyperparameter range, requiring only a nonzero entropy loss and a batch size large enough for meaningful consistency estimates. This robustness is intrinsic to the family, not an artifact of tuning, frequency masking's instability persists across the full hyperparameter range.

### 4.7 Limitations and Scope

While M2A holds losses, selection strategy, and masking schedule fixed to isolate the family axis, different masking families still perturb the input in structurally different ways. Equal masked-area ratios do not imply equal information removal: spatial masks occlude localized regions while leaving the remaining pixels unchanged, whereas frequency masks redistribute spectral energy globally and can remove more task-relevant structure even at the same nominal ratio. Our experiments should therefore be read as comparing families under a shared masking curriculum rather than as a perfectly controlled causal isolation of $\mathcal{F}$; accordingly, we use the structural-preservation principle as a qualitative lens for organizing the observed patterns, not as a calibrated causal model. Appendix A.10 discusses these limitations in more detail and explains how they constrain the scope of our claims.

## 5 Conclusion and Future Work

Within the controlled M2A protocol used throughout this study, fixed losses, random selection strategy, shared masking schedule, and single-step update, we find that the masking family ($\mathcal{F}$) is a major determinant of adaptation stability. Our systematic evaluation reveals that while the selection strategy ($\mathcal{S}$) may possibly offer refinement, the choice of $\mathcal{F}$ determines whether adaptation builds robust representations or compounds errors. We use a *structural-preservation principle* as a conceptual explanation for these trends. In summary, the main empirical pattern is that contiguous patch masking is the most reliable default, while the main exceptions arise in regimes where architecture or task structure makes frequency masking more competitive. We use the principle primarily as a compact lens for those central findings, through the two qualitative ingredients of spatial coherence and spectral overlap. Practical design guidance therefore follows two axes: (i) *architecture alignment*, where patch-tokenized ViTs usually demand spatial occlusion to avoid collapse, and (ii) *task alignment*, where global-cue tasks with high-capacity models can make frequency masking competitive.

We emphasize that the conclusions are established under the specific M2A framework and the benchmarks studied here. Whether the same family ordering holds under substantially different losses, update rules, or adaptation protocols remains an open empirical question. The broader design guidance, that practitioners should attend to masking-family choice and its interaction with architecture and task structure, is a plausible implication of our results rather than a fully general statement about all masking-based CTTA methods. Future work should extend this principle to cross-modal adaptation and investigate the formal relationship between gradient signal quality and structural preservation.

## 6 Acknowledgements

The first author is supported by a PhD scholarship from the Faculty of Science and Technology (REALTEK), Norwegian University of Life Sciences (NMBU). We thank the editors and reviewers for their valuable feedback and comments. Computational support was provided by Sigma2—the National Infrastructure for High-Performance Computing and Data Storage in Norway—including access to the LUMI supercomputer (project NN11074K). LUMI is owned by the EuroHPC Joint Undertaking and hosted by CSC (Finland) and the LUMI consortium through Sigma2—Norway. Additional computational support was provided by the Orion High Performance Computing Center (OHPCC) at NMBU.

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

# A  Appendix

## Contents

## A.1  Masking-Based CTTA Baselines

This section contextualizes M2A's findings relative to four recent masking-based baselines: Continual-MAE (Liu et al., 2024a), REM (Han et al., 2025), SPA (Niu et al., 2025), and TCA (Wang et al., 2025b). Each baseline couples a specific masking family $\mathcal{F}$ with a specific selection strategy $\mathcal{S}$. Our study allows us to reinterpret their results through the lens of Finding 1 (stability is family-dependent) and Finding 2 (optimality is architecture-task dependent).

- **Continual-MAE** pairs $\mathcal{F}$=patch with a Distribution-aware Masking (DaM) mechanism ($\mathcal{S}$=uncertainty) that uses Monte Carlo dropout to estimate token-wise distribution shifts. It adds a linear decoder to reconstruct hand-crafted Histograms of Oriented Gradients (HOG) for masked tokens. **Similarities:** Its stability over long streams supports Finding 1, confirming that spatial masking avoids the failures predicted by the structural-preservation principle. **Differences in Findings:** Continual-MAE concludes that explicitly reconstructing invariant features (HOG) is essential to mitigate distribution shift accumulation. M2A's findings suggest a simpler alternative: when the masking family is well-aligned with the architecture (Finding 2), prediction-centric consistency losses alone are sufficient to achieve state-of-the-art stability. The minimal performance gap between M2A ($\mathcal{S}$=random) and Continual-MAE suggests that $\mathcal{F}$ is the primary driver of stability, while reconstruction provides secondary regularization rather than a fundamental necessity.

- **REM** pairs $\mathcal{F}$=patch with $\mathcal{S}$=attention (cross-head ranking) and enforces a Ranked Entropy Minimization loss across a "mask chain" of increasing ratios. **Similarities:** REM's stability reinforces Finding 1. Its observation that accuracy tracks the masking ratio aligns with the structural-preservation principle: spatial masking remains "spectrally safe" across varying intensities. **Differences in Findings:** REM finds that an explicit entropy ranking relationship is necessary to prevent model collapse. M2A demonstrates that collapse is family-dependent: while frequency masking collapses due to the structural-preservation principle (Finding 1), patch masking is inherently stable enough to avoid collapse under standard losses without requiring an explicit hierarchy or ranking loss.

- **SPA** pairs $\mathcal{F}$=low-freq with an active self-bootstrapping scheme ($\mathcal{S}$=rules/policy) that only adapts when strong-view confidence exceeds weak-view confidence. It uses random low-frequency amplitude masking and high-frequency Gaussian noise injection. **Similarities:** SPA's analysis of Radially Averaged Power Spectral Density (RAPSD) mirrors our own frequency analysis: both identify that low-frequency components carry the highest information power. **Differences in Findings:** SPA finds that low-frequency masking is a versatile and effective TTA signal because it creates large RAPSD

differences for bootstrapping. M2A's Finding 1 identifies the opposite for CTTA: frequency-domain occlusion is inherently unstable on ViTs according to the structural-preservation principle, as its spectral footprint overlaps with common corruptions, causing consistency losses to reinforce errors. This divergence likely stems from two factors: (i) protocol: SPA's TTA protocol (with resets) masks the cumulative collapse identified in M2A; (ii) stabilization: SPA's noise injection supplies learning signals that may act as a spectral anchor, whereas M2A's controlled experiments isolate the instability of the frequency band itself.

- **TCA** introduces a dual-threshold, cross-head ranking system ($\mathcal{S}$=Attention) to classify CLIP tokens into prune, merge, and retain groups, using historical domain anchor tokens to refine attention. **Similarities:** TCA's emphasis on robust ranking (averaging relative positions rather than raw scores) aligns with our use of random selection to avoid outlier signals. **Differences in Findings:** TCA finds that merging ambiguous tokens (condensing them via coresets) is superior to pruning them, as it preserves more diversity. M2A's Finding 2 suggests that this sophisticated semantic selection is most beneficial when the masking family does not naturally align with the task. While M2A shows random patch masking is sufficient for localized-cue tasks on supervised backbones, TCA's success on CLIP suggests that for robust, semantically structured feature spaces, the selection strategy $\mathcal{S}$ can transition from grid-based masking to fine-grained semantic condensation, effectively extending the architecture-alignment principle of Finding 2.

## A.2 M2A Algorithm

Algorithm 1 consolidates the per-batch procedure described in Section 3. Mask positions are sampled uniformly from $\mathcal{F}$ rather than scored by uncertainty (Continual-MAE) or attention (REM). M2A always uses $\mathcal{S}$=random; comparisons with heuristic baselines are therefore system-level rather than controlled $\mathcal{S}$ swaps.

---

**Algorithm 1** M2A per-batch procedure. Only the masking family $\mathcal{F}$ varies across experimental conditions.

---

**Require:** Source-trained classifier $f_\theta$; masking family $\mathcal{F} \in \{$patch, pixel, all-freq, low-freq, high-freq$\}$; number of views $n$; mask step $\alpha$; loss weight $\lambda$; learning rate $\eta$
**Ensure:** Continually adapted parameters $\theta$
1: **for** each incoming test batch $B_s$ in the target stream **do**
2:     **for** each image $x \in B_s$ **do**
3:         Set $m_0 = 0$ {view $x^{(0)} = x$ is unmasked}
4:         **for** $t = 1, \ldots, n-1$ **do**
5:             $m_t \leftarrow t \cdot \alpha$ {linear masking-ratio curriculum}
6:             Sample mask $M^{(t)}$ from family $\mathcal{F}$ using strategy $\mathcal{S}$ with masked fraction $m_t$ {$\mathcal{S}$=random in M2A; cf. uncertainty/attention in baselines}
7:             $x^{(t)} \leftarrow \text{Mask}(x, M^{(t)})$ {Sec. 3: spatial or frequency}
8:         **end for**
9:         Forward all views: $p^{(t)} = \text{softmax}\big(f_\theta(x^{(t)})\big)$, $t = 0, \ldots, n-1$
10:     **end for**
11:     Compute $\mathcal{L}_{\text{CL}}$ (Equation 1) and $\mathcal{L}_{\text{EL}}$ (Equation 2) over $B_s$
12:     $\mathcal{L}_{\text{CTTA}} \leftarrow \mathcal{L}_{\text{CL}} + \lambda \, \mathcal{L}_{\text{EL}}$
13:     $\theta \leftarrow \theta - \eta \, \nabla_\theta \mathcal{L}_{\text{CTTA}}$ {single gradient step; no reset}
14: **end for**

---

## A.3 Structural-Preservation Hypothesis

We use structural preservation as a compact qualitative lens built around two ingredients, spatial coherence and spectral overlap. This subsection provides a *conceptual formalization* of that lens in the online CTTA setting, where the model receives only the observed target input $x \in [0,1]^{C \times H \times W}$, which may already be corrupted. The purpose of this formulation is interpretive rather than operational: it is not a calibrated predictor used in our empirical analysis, but a stylized way to organize our preferred qualitative explanation for why some masking families are more stable than others. The experimental evidence is supportive of this

interpretation, but it does not by itself establish structural preservation as the sole or definitive mechanism behind the observed family-dependent behavior. For masking family $\mathcal{F}$, difficulty level $t$, and binary mask $M^{(t)}$, let the corresponding masked view be

$$x_{\mathcal{F}}^{(t)} = \text{Mask}_{\mathcal{F}}(x, M^{(t)}). \tag{3}$$

To connect this conceptual formalization to the actual masked view, we introduce a retained-energy ratio, $\rho_{\mathcal{F}}^{(t)}(x) = \frac{\|x_{\mathcal{F}}^{(t)}\|_2^2}{\|x\|_2^2}$. Its role is only to exclude degenerate cases in which the mask removes so much of the observed input that the remaining view is nearly blank, regardless of coherence or overlap. Here $\rho_{\mathcal{F}}^{(t)}(x) \in [0, 1]$ is the retained-energy ratio: values near 1 indicate that the masked view still contains most of the available signal, while small values indicate aggressive information removal.

For spatial masking, the key variable is fragmentation of the mask, not only its total area. Let $\mathcal{A}^{(t)} = \{(i, j) : M^{(t)}(i, j) = 1\}$ denote the masked region, and let $\partial \mathcal{A}^{(t)}$ denote a schematic boundary set capturing locations where the mask meets its complement. We summarize spatial coherence by

$$\mathcal{C}_{\mathcal{F}}(M^{(t)}) = \begin{cases} \exp\left(-\beta \frac{|\partial \mathcal{A}^{(t)}|}{HW}\right), & \mathcal{F} \in \{\text{patch}, \text{pixel}\}, \\ 1, & \mathcal{F} \in \{\text{all-freq}, \text{low-freq}, \text{high-freq}\}, \end{cases} \tag{4}$$

Here $|\partial \mathcal{A}^{(t)}|$ should be read only as a proxy for fragmentation: contiguous masks have smaller boundary than scattered masks at the same masked ratio, and therefore larger $\mathcal{C}_{\mathcal{F}}(M^{(t)})$. The scalar $\beta > 0$ is a user-independent penalty coefficient controlling how strongly fragmentation is penalized.

For frequency masking, the dominant issue is overlap with the image's spectral damage zone. We denote by $\mathcal{D}(x) \subseteq \Omega_{\text{all}}$ the set of frequencies that are already strongly degraded in the observed corrupted image $x$; this is a theoretical latent set, not an additional input to the algorithm. If $\mathcal{F} \in \{\text{all-freq}, \text{low-freq}, \text{high-freq}\}$, let $\Omega_{\mathcal{F}} \in \{\Omega_{\text{all}}, \Omega_{\text{low}}, \Omega_{\text{high}}\}$ be the band masked by that family, using the notation of Section A.4.2. We then define the masked-band overlap by

$$\chi(\mathcal{F}, x) = \begin{cases} 0, & \mathcal{F} \in \{\text{patch}, \text{pixel}\}, \\ \dfrac{|\Omega_{\mathcal{F}} \cap \mathcal{D}(x)|}{|\Omega_{\mathcal{F}}|}, & \mathcal{F} \in \{\text{all-freq}, \text{low-freq}, \text{high-freq}\}. \end{cases} \tag{5}$$

The quantity $\chi(\mathcal{F}, x) \in [0, 1]$ measures how much the masked band overlaps with frequencies that are already damaged in the current target image. Large overlap means that the mask deletes signal from exactly the region of the spectrum where the image is already weak, making unstable views more likely.

$$\mathcal{I}_{\text{pres}}(\mathcal{F}; x, t) = \rho_{\mathcal{F}}^{(t)}(x) \, \mathcal{C}_{\mathcal{F}}(M^{(t)}) \left(1 - \chi(\mathcal{F}, x)\right). \tag{6}$$

$$\mathcal{I}_{\text{pres}}(\mathcal{F}; x, t) \geq \tau \implies \text{more likely stable}, \qquad \mathcal{I}_{\text{pres}}(\mathcal{F}; x, t) < \tau \implies \text{more likely unstable}, \tag{7}$$

Here $\mathcal{I}_{\text{pres}}(\mathcal{F}; x, t)$ is a schematic structural-preservation index and $\tau \in (0, 1)$ is a conceptual task-dependent stability threshold. They are introduced only to express the appendix intuition in compact form: the index is high when the masked view is non-degenerate, preserves spatial coherence when spatial masking is used, and avoids masking frequencies that lie in the damaged part of the spectrum. We do *not* estimate $\mathcal{I}_{\text{pres}}$, fit $\tau$, or identify $\mathcal{D}(x)$ in the experiments. Thus the formalization should be read as a stylized bridge from the two-ingredient narrative to the observed family-level trends. It is intended as an organizing hypothesis that is consistent with the empirical patterns, not as definitive evidence that structural preservation is the main mechanism behind every family-dependent effect. Architecture, task structure, and corruption-specific behavior may also contribute to the observed differences. The two conditions below are therefore meant only as heuristic, interpretive sufficient conditions, not as guarantees.

**Heuristic condition 1 (label-information preservation).** If $\mathcal{I}_{\text{pres}}(\mathcal{F}; x, t)$ is relatively large, then the masked view $x_{\mathcal{F}}^{(t)}$ is unlikely to substantially alter the class-relevant conditional label information present in $x$. In other words, structurally safe masking should preserve the dominant predictive content well enough that the class posterior is only mildly perturbed, rather than exactly invariant.

**Heuristic condition 2 (preservation along the continual stream).** For masking families that are empirically stable, the preceding condition should tend to hold for most inputs and mask draws encountered along the CTTA stream at the masking levels used by M2A. Conversely, collapse-prone families should violate it more often on some domain–level pairs $(D_s, t)$, so that unstable views are encountered frequently enough to degrade adaptation over time. This distinction is intended only as a qualitative way to separate stable from unstable families, not as a proved high-probability guarantee.

### A.4 Masking Implementation Details

This section provides the full algorithmic details of the spatial and frequency masking mechanisms summarised in Section 3. All notation follows this: $x \in [0,1]^{C \times H \times W}$ is the input image, $m_t = t\alpha$ is the masked-area fraction for view $t$, and $M^{(t)} \in \{0,1\}^{H \times W}$ is the binary mask broadcast across channels as $\tilde{M}^{(t)} = \text{broadcast}_C(M^{(t)})$.

#### A.4.1 Spatial Masking

**Patch-based masking.** The default patch sampler used in all experiments is *Block-Patch*: a single contiguous masked block aligned to the ViT token grid, not a scattered subset of token-aligned patches. Let $p$ denote the token-grid patch size (e.g. $p$=16 for ViT-B/16), and define

$$P_h = \left\lfloor \frac{H}{p} \right\rfloor, \qquad P_w = \left\lfloor \frac{W}{p} \right\rfloor, \qquad N_{\text{patch}} = P_h P_w. \tag{8}$$

For target masked fraction $m_t$, we choose the number of masked grid cells as

$$k_t = \lfloor m_t N_{\text{patch}} \rceil. \tag{9}$$

We then select block dimensions $h_t, w_t \in \mathbb{N}$ such that $h_t w_t \approx k_t$, sample one top-left grid index $(u_0^{(t)}, v_0^{(t)})$ uniformly from $\{0, \ldots, P_h - h_t\} \times \{0, \ldots, P_w - w_t\}$, and define the contiguous masked patch set

$$\mathcal{S}_{\text{block}}^{(t)} = \{(u,v) : u_0^{(t)} \le u < u_0^{(t)} + h_t, \; v_0^{(t)} \le v < v_0^{(t)} + w_t\}. \tag{10}$$

If $\mathcal{R}_{u,v}$ denotes the image region covered by grid cell $(u,v)$, then the masked image support is $\mathcal{A}^{(t)} = \bigcup_{(u,v) \in \mathcal{S}_{\text{block}}^{(t)}} \mathcal{R}_{u,v}$ and $M^{(t)}(i,j) = \mathbf{1}[(i,j) \in \mathcal{A}^{(t)}]$. The masked view is $x^{(t)} = x \odot (\mathbf{1} - \tilde{M}^{(t)})$. The appendix-only *Free-Patch* variant differs only in this sampling step: it removes the contiguity constraint and samples token-aligned grid cells independently, as detailed in Section A.5.

**Pixel-wise masking.** We draw $k_t = \lfloor m_t HW \rceil$ pixel positions uniformly without replacement:

$$\mathcal{A}^{(t)} \sim \text{Unif}\Big(\{\mathcal{S} \subseteq \{0, \ldots, H-1\} \times \{0, \ldots, W-1\} : |\mathcal{S}| = k_t\}\Big). \tag{11}$$

The mask and masked view are constructed identically to the patch case: $M^{(t)}(i,j) = \mathbf{1}[(i,j) \in \mathcal{A}^{(t)}]$ and $x^{(t)} = x \odot (\mathbf{1} - \tilde{M}^{(t)})$. In practice, the $k_t$ indices are obtained via a random permutation of $\{0, \ldots, HW-1\}$, selecting the first $k_t$ entries.

#### A.4.2 Frequency Masking

**Forward transform.** Each channel $x_c \in \mathbb{R}^{H \times W}$ is transformed independently via the orthonormal 2-D DFT:

$$X_c[u,v] = \frac{1}{\sqrt{HW}} \sum_{i=0}^{H-1} \sum_{j=0}^{W-1} x_c[i,j] \, e^{-2\pi \mathrm{i}(ui/H + vj/W)}, \qquad u \in \{0, \ldots, H-1\}, \; v \in \{0, \ldots, W-1\}. \tag{12}$$

**Self-conjugate bin identification.** Because $x_c$ is real-valued, its DFT satisfies Hermitian symmetry: $X_c[u,v] = \overline{X_c[(-u \bmod H, -v \bmod W)]}$. A bin $(u,v)$ is *self-conjugate* when $(-u \bmod H, -v \bmod W) = (u,v)$, i.e. it is its own conjugate partner. The self-conjugate set is

$$\mathcal{C}_{\text{sc}} = \{(0,0)\} \; \cup \; \mathbf{1}[H \text{ even}]\{(\tfrac{H}{2}, 0)\} \; \cup \; \mathbf{1}[W \text{ even}]\{(0, \tfrac{W}{2})\} \; \cup \; \mathbf{1}[H, W \text{ even}]\{(\tfrac{H}{2}, \tfrac{W}{2})\}. \tag{13}$$

These bins are *always excluded* from masking. Excluding DC preserves the mean intensity; excluding Nyquist bins guarantees that the inverse transform remains real-valued without requiring explicit imaginary-part correction.

**Conjugate-pair enumeration.** All remaining bins are grouped into conjugate pairs $\{(u,v), (-u \bmod H, -v \bmod W)\}$. We enumerate the unique pairs by retaining only the representative with the smaller row-major linear index:

$$\mathcal{P} = \big\{ \big((u,v), (u',v')\big) : u' = -u \bmod H, \, v' = -v \bmod W, \, uW+v < u'W+v', \, (u,v) \notin \mathcal{C}_{\text{sc}} \big\}. \tag{14}$$

The total number of unique pairs is $|\mathcal{P}| = (HW - |\mathcal{C}_{\text{sc}}|)/2$.

**Band membership.** Each bin's normalised radial distance from DC is computed in centered-frequency coordinates:

$$\tilde{u} = \begin{cases} u & u \leq \lfloor H/2 \rfloor \\ u - H & \text{otherwise} \end{cases}, \qquad \tilde{v} = \begin{cases} v & v \leq \lfloor W/2 \rfloor \\ v - W & \text{otherwise} \end{cases}, \qquad r(u,v) = \frac{\sqrt{\tilde{u}^2 + \tilde{v}^2}}{\sqrt{\lfloor H/2 \rfloor^2 + \lfloor W/2 \rfloor^2}}. \tag{15}$$

A pair is assigned to a band via the representative bin's radial distance and a fixed cutoff $r_c = 0.5$:

$$\Omega_{\text{low}} = \{p \in \mathcal{P} : r(p) \leq r_c\}, \qquad \Omega_{\text{high}} = \{p \in \mathcal{P} : r(p) > r_c\}, \qquad \Omega_{\text{all}} = \mathcal{P}. \tag{16}$$

**Pair selection and masking.** For a chosen band $\Omega \in \{\Omega_{\text{low}}, \Omega_{\text{high}}, \Omega_{\text{all}}\}$ and target fraction $m_t$, the number of pairs to zero is $k = \lceil m_t |\Omega| \rceil$. For each sample in the batch independently, $k$ pairs are drawn uniformly without replacement from $\Omega$ via a random permutation. Both bins in every selected pair are zeroed simultaneously, producing a frequency-domain mask $M_{\text{freq}}^{(t)} \in \{0,1\}^{H \times W}$ that is symmetric under conjugation:

$$\tilde{X}_c^{(t)}[u,v] = M_{\text{freq}}^{(t)}[u,v] \, X_c[u,v]. \tag{17}$$

Because both members of every conjugate pair are zeroed together and all self-conjugate bins are preserved, $\tilde{X}_c^{(t)}$ retains Hermitian symmetry.

**Inverse transform.** The masked view is reconstructed via the orthonormal inverse DFT and the real part is taken:

$$x_c^{(t)} = \text{Re}\left( \frac{1}{\sqrt{HW}} \sum_{u=0}^{H-1} \sum_{v=0}^{W-1} \tilde{X}_c^{(t)}[u,v] \, e^{+2\pi i (ui/H + vj/W)} \right). \tag{18}$$

The imaginary residual is numerically negligible ($\lesssim 10^{-7}$) due to the preserved conjugate symmetry. No clamping is applied; at the moderate ratios used ($m_t \leq 20\%$), Gibbs-like ringing artifacts are small and subsumed by the corruption already present in the target stream.

## A.5 Mask Sampling Methods

Table 8: Different mask sampling types. Entries report classification error rate (%) per corruption type on ImageNet-C.

| Sampling | Mask Steps | GN | SN | IN | DB | GB | MB | ZB | S | Fr | F | B | C | ET | P | JC | Mean↓ |
|---|---|---|---|---|---|---|---|---|---|---|---|---|---|---|---|---|---|
| Free-Patch | 0%-10%-20% | 42.22 | 36.84 | 37.84 | 62.20 | 97.16 | 99.24 | 99.76 | 43.18 | 36.96 | 37.04 | 23.04 | 47.06 | 41.12 | 30.88 | 30.42 | 51.00 |
| Free-Patch | 0%-20%-40% | 42.22 | 36.54 | 37.32 | 53.78 | 46.36 | 43.30 | 49.30 | 35.66 | 34.46 | 38.14 | 21.94 | 43.80 | 40.42 | 29.08 | 29.90 | 38.81 |
| Free-Patch | 0%-30%-60% | 42.70 | 36.56 | 37.70 | 53.88 | 45.52 | 41.20 | 45.30 | 35.42 | 34.96 | 92.78 | 23.26 | 46.72 | 39.08 | 28.50 | 30.12 | 42.25 |
| Grid-Patch | 0%-10%-20% | 42.12 | 36.98 | 38.12 | 71.38 | 99.86 | 98.66 | 97.92 | 40.18 | 36.44 | 37.62 | 22.68 | 45.82 | 42.46 | 30.12 | 30.18 | 51.37 |
| Grid-Patch | 0%-20%-40% | 42.92 | 37.20 | 38.64 | 63.82 | 59.44 | 89.76 | 98.52 | 82.30 | 86.00 | 99.82 | 81.98 | 99.92 | 99.86 | 99.66 | 97.80 | 78.51 |
| Grid-Patch | 0%-30%-60% | 43.46 | 37.64 | 38.70 | 56.94 | 45.90 | 43.78 | 48.74 | 36.54 | 37.02 | 92.02 | 23.84 | 49.28 | 42.18 | 28.88 | 30.90 | 43.72 |
| Block-Patch | 0%-10%-20% | 41.58 | 35.84 | 36.56 | 52.38 | 45.38 | 40.36 | 42.28 | 35.56 | 34.44 | 34.16 | 21.34 | 44.06 | 39.50 | 28.08 | 29.30 | 37.39 |
| Block-Patch | 0%-20%-40% | 41.72 | 35.64 | 36.10 | 52.82 | 43.44 | 39.46 | 41.22 | 35.26 | 33.74 | 32.90 | 21.10 | 43.70 | 37.76 | 28.00 | 28.46 | 36.75 |
| Block-Patch | 0%-30%-60% | 42.10 | 35.62 | 36.56 | 54.60 | 44.46 | 39.22 | 41.96 | 34.96 | 33.70 | 35.56 | 20.78 | 43.94 | 38.52 | 28.12 | 28.54 | 37.24 |

### A.5.1 Free-Patch

Free-Patch keeps the same $16 \times 16$ patch grid but removes the contiguity constraint. For a target masked fraction $m_t$, the implementation chooses $k_t = \lfloor m_t N_{\text{patch}} \rfloor$ and samples a subset of distinct patch indices uniformly without replacement,

$$\mathcal{S}_{\text{free}}^{(t)} \sim \text{Unif}\Big(\{\mathcal{S} \subseteq \{0, \ldots, P_h{-}1\} \times \{0, \ldots, P_w{-}1\} : |\mathcal{S}| = k_t\}\Big). \tag{19}$$

The resulting mask is

$$M^{(t)}(i,j) = \mathbf{1}\big[\exists (u,v) \in \mathcal{S}_{\text{free}}^{(t)} : (i,j) \in \mathcal{R}_{u,v}\big]. \tag{20}$$

Thus, the same total masked area is distributed over multiple spatially separated token-aligned patches.

### A.5.2 Grid-Patch

Grid-Patch also uses the same patch grid, but imposes a checkerboard parity structure. Let $B = \{(u,v) : (u+v) \bmod 2 = 0\}$ and $W = \{(u,v) : (u+v) \bmod 2 = 1\}$ denote the two parity classes. The target number of masked patches is again $k_t = \lfloor m_t N_{\text{patch}} \rfloor$. If $k_t \leq |B|$, the sampler draws only from one parity class,

$$\mathcal{S}_{\text{grid}}^{(t)} \sim \text{Unif}\big(\{\mathcal{S} \subseteq B : |\mathcal{S}| = k_t\}\big), \tag{21}$$

whereas if $k_t > |B|$, it first masks all cells in $B$ and then samples the remaining indices from $W$,

$$\mathcal{S}_{\text{grid}}^{(t)} = B \cup \mathcal{S}_W^{(t)}, \qquad \mathcal{S}_W^{(t)} \sim \text{Unif}\big(\{\mathcal{S} \subseteq W : |\mathcal{S}| = k_t - |B|\}\big). \tag{22}$$

The mask is then constructed from $\mathcal{S}_{\text{grid}}^{(t)}$ in the same way as for Free-Patch. This creates a highly regular, spatially dispersed occlusion pattern.

### A.5.3 Block-Patch

Block-Patch is exactly the default contiguous patch-based masking described in Section 3 and Appendix A.4.1: it selects a spatially contiguous masked region aligned with this patch grid, so we do not restate the same equations here.

**Empirical analysis.** Table 8 shows that, beyond the default Block-Patch design, we evaluate Free-Patch and Grid-Patch under three masking-ratio schedules. Block-Patch is the most stable across all masking-ratio steps and achieves the lowest mean error in all three schedules (37.39, 36.75, and 37.24), whereas Free-Patch and especially Grid-Patch show much larger error spikes. This indicates that CTTA behavior tracks the masking family and, more specifically here, the spatial sampling structure: concentrating the masked area into one contiguous block is substantially more stable than fragmenting it across scattered or checkerboard patches.

## A.6 Spectral Analysis of Corruptions

This section provides the empirical grounding for the *structural-preservation principle* (Finding 1) by visualizing the average power spectra of the 15 ImageNet-C corruption types. **Implementation details.** The 2D power spectra are generated by averaging the log-magnitude of the Fourier transform across 5,000 randomly sampled images per corruption type at severity 5. The resulting visualizations in Figure 6 represent the characteristic "spectral footprint" of each corruption.

The spectra confirm the mechanistic claims in Section 4. Blur-type corruptions (Defocus, Glass, Motion, Zoom) concentrate energy in the center (low frequencies) and severely attenuate the periphery (high frequencies), acting as low-pass filters. Consequently, *high-frequency masking* on these corruptions removes the only remaining discriminative signal, leading to the collapse observed in Table 1c.

Conversely, "noise" corruptions (Gaussian, Shot, Impulse) distribute energy across all frequency bands, creating a broad damage zone. Digital artifacts like *Pixelate* and *JPEG Compression* create structured

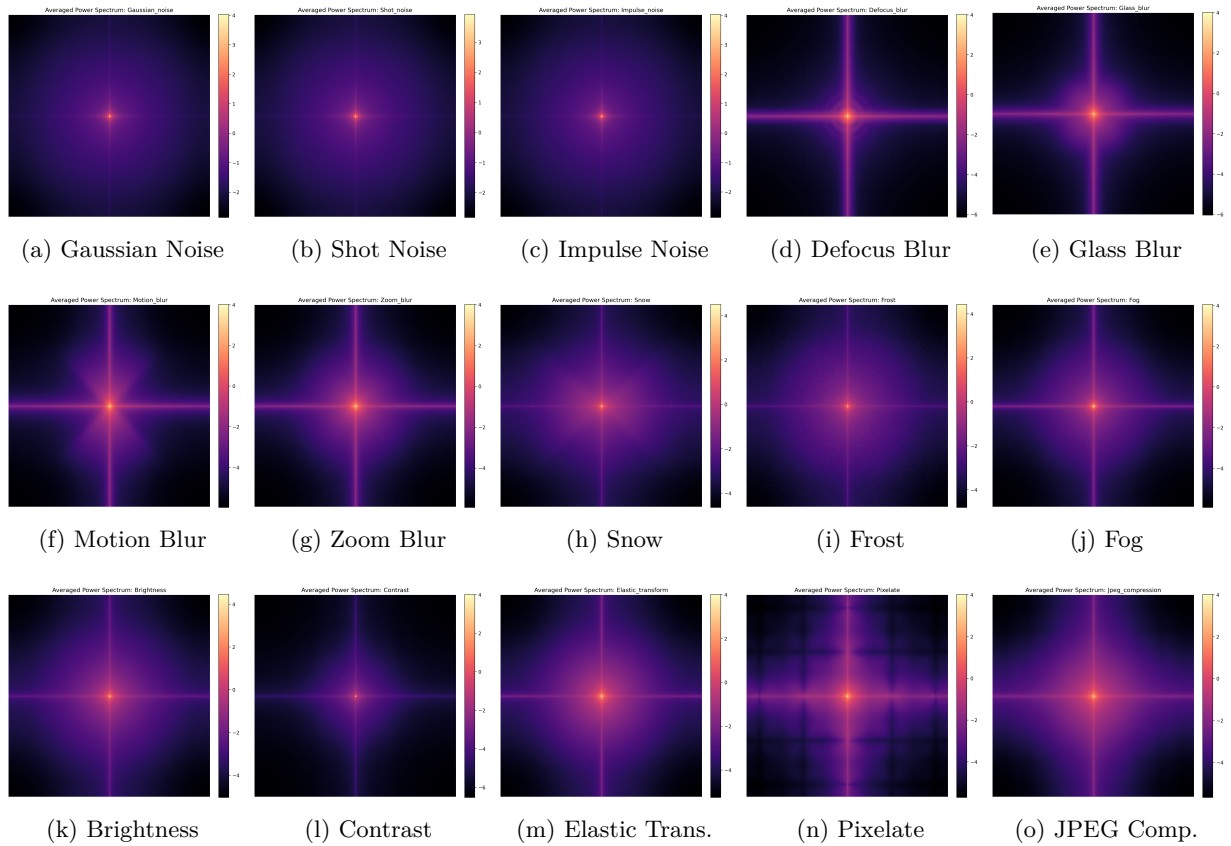

Figure 6: Average power spectra for 15 ImageNet-C corruption types (severity 5). Each plot shows the log-magnitude 2D Fourier spectrum averaged over 5,000 examples. The center of each plot represents low frequencies, while the periphery represents high frequencies.

spectral residues: Pixelate preserves sharp edges (high-frequency spikes) while degrading the natural low-frequency manifold. The *structural-preservation principle* provides a predictive account for these failures: whenever the chosen masking family $\mathcal{F}$ (Section 3) targets a frequency band that is already attenuated or terminally corrupted by the environment, the model is forced to adapt on uninformative views, causing the catastrophic drift.

### A.6.1 Reordered Streams

This subsection evaluates randomized corruption ordering with inter-pass resets on CIFAR-10-C and ImageNet-C (Table 9). On CIFAR-10-C, $\mathcal{F}$=patch achieves near-perfect stability; on ImageNet-C, it remains the leading family despite mild sensitivity to specific orderings. $\mathcal{F}$=all-freq exhibits high instability, oscillating between baseline performance and large error spikes. Under resets, $\mathcal{F}$=all-freq occasionally recovers, whereas the lifelong protocol (Table 3) shows irreversible collapse. This confirms that frequency-masking instability is a structural property that intensifies when drift accumulates without resets.

### A.7 Architecture and Task Scoping

### A.7.1 CarlaTTA: Dataset and Experimental Setup

The CarlaTTA benchmark (Döbler et al., 2024) provides synthetic driving scenes rendered in the CARLA simulator across continuously changing environmental conditions (day, night, fog, rain). Frames are organized into scenario-specific sequences (e.g. `clear2fog`, `clear2rain`, `day2night`, and `dynamic`), each containing

Table 9: Domain shuffling in CIFAR10-C and ImageNet-C under CTTA. In each pass, the model is reset back to source weights. Entries report mean classification error rate (%) per pass.

(a) CIFAR10-C

| Method | Pass 1 | Pass 2 | Pass 3 | Pass 4 | Pass 5 | Pass 6 | Pass 7 | Pass 8 | Pass 9 | Pass 10 | Mean↓ | Gain |
|---|---|---|---|---|---|---|---|---|---|---|---|---|
| Source | 28.2 | 28.2 | 28.2 | 28.2 | 28.2 | 28.2 | 28.2 | 28.2 | 28.2 | 28.2 | 28.2 | 0.0 |
| Tent | 22.9 | 21.6 | 22.1 | 23.6 | 26.3 | 22.2 | 21.7 | 23.9 | 22.4 | 21.8 | 22.8 | +5.4 |
| CoTTA | 25.8 | 26.1 | 26.6 | 26.8 | 26.6 | 26.5 | 26.8 | 27.0 | 26.1 | 26.9 | 26.5 | +1.7 |
| SAR | 26.2 | 26.2 | 26.2 | 26.2 | 26.2 | 26.2 | 26.2 | 26.2 | 26.2 | 26.2 | 26.2 | +2.0 |
| Rotta | 20.7 | 20.8 | 20.5 | 20.5 | 20.6 | 20.7 | 20.5 | 20.6 | 20.7 | 20.5 | 20.6 | +7.6 |
| Deyo | 16.9 | 16.7 | 16.7 | 16.9 | 16.9 | 16.7 | 17.1 | 16.5 | 16.6 | 16.8 | 16.8 | +11.4 |
| ROID | 14.7 | 14.8 | 14.8 | 14.7 | 14.8 | 14.6 | 14.8 | 14.7 | 14.7 | 14.6 | 14.7 | +13.5 |
| REM | 9.4 | 9.0 | 9.9 | 9.1 | 9.1 | 9.2 | 9.2 | 14.6 | 9.1 | 9.9 | 9.9 | +18.3 |
| M2A($\mathcal{F}$=low-freq) | 12.3 | 12.2 | 12.7 | 12.3 | 12.7 | 12.1 | 12.5 | 12.1 | 12.1 | 12.2 | 12.3 | +15.9 |
| M2A($\mathcal{F}$=patch) | 8.3 | 8.4 | 8.4 | 8.4 | 8.7 | 8.4 | 8.4 | 8.5 | 8.9 | 8.9 | 8.5 | +19.7 |

(b) ImageNet-C

| Method | Pass 1 | Pass 2 | Pass 3 | Pass 4 | Pass 5 | Pass 6 | Pass 7 | Pass 8 | Pass 9 | Pass 10 | Mean↓ | Gain |
|---|---|---|---|---|---|---|---|---|---|---|---|---|
| Source | 56.2 | 56.2 | 56.2 | 56.2 | 56.2 | 56.2 | 56.2 | 56.2 | 56.2 | 56.2 | 56.2 | 0.0 |
| Tent | 51.2 | 51.2 | 51.5 | 51.5 | 51.8 | 51.6 | 51.9 | 51.5 | 51.4 | 51.6 | 51.5 | +4.7 |
| CoTTA | 53.1 | 53.1 | 53.2 | 53.1 | 53.2 | 53.2 | 53.1 | 53.1 | 53.1 | 53.0 | 53.2 | +3.0 |
| SAR | 43.2 | 49.7 | 43.3 | 43.4 | 43.5 | 43.1 | 43.2 | 43.8 | 43.4 | 43.9 | 44.0 | +12.2 |
| ROTTA | 50.4 | 50.3 | 50.3 | 50.4 | 50.4 | 50.3 | 50.3 | 50.3 | 50.2 | 50.3 | 50.3 | +5.9 |
| DEYO | 42.0 | 41.9 | 41.8 | 41.8 | 41.9 | 42.0 | 42.2 | 41.8 | 41.9 | 41.9 | 41.9 | +14.3 |
| ROID | 40.4 | 42.7 | 42.5 | 40.6 | 41.5 | 40.1 | 39.9 | 40.0 | 40.0 | 39.7 | 40.7 | +15.5 |
| REM | 39.4 | 39.5 | 39.6 | 39.5 | 39.5 | 39.8 | 39.8 | 39.4 | 39.6 | 39.9 | 39.6 | +16.6 |
| M2A($\mathcal{F}$=low-freq) | 41.7 | 41.8 | 41.7 | 41.7 | 41.7 | 41.7 | 41.8 | 41.9 | 41.9 | 41.6 | 41.7 | +14.5 |
| M2A($\mathcal{F}$=patch) | 37.5 | 41.4 | 37.7 | 41.8 | 37.6 | 37.7 | 37.8 | 41.1 | 37.8 | 41.4 | 39.2 | +17.0 |

paired RGB images and pixel-wise semantic labels across 14 classes. Raw CARLA label IDs are mapped to contiguous training IDs; all unmapped pixels receive an ignore index of 255.

**Source-model training.** DeepLabV2 backbone (`MODEL.NUM_CLASSES` = 14) is trained on the clear-weather split. Source-domain augmentation includes random scale–crop (scale range [0.75, 2.0], base size 512, crop 1024×512) and horizontal flip ($p$=0.5). At test time, images are resized so the shorter side is 1 024 pixels and normalized to [0, 1].

**Test-time adaptation protocol.** All TTA methods share the same evaluation pipeline: the source checkpoint is loaded, wrapped with the chosen adaptation method, and evaluated on the full target sequence reporting mean IoU. Gradient-based methods use a common optimizer factory supporting SGD and Adam; defaults are SGD with learning rate $1\times10^{-4}$, weight decay 0.0, and batch size 1. Adaptation is online (updates accumulate along the target stream; no episodic reset), matching the CTTA protocol used on the classification benchmarks.

### A.7.2 CNN and Dense Prediction

We evaluate M2A on semantic segmentation using the DeepLabV2 backbone across CarlaTTA scenarios (Table 10, Figure 7). $\mathcal{F}$=patch achieves the highest mIoU on most scenarios. The family gap is narrower than on ViT-based classification, consistent with overlapping convolutional receptive fields not aligning with patch boundaries as naturally as tokenization does. However, on "clear2fog," $\mathcal{F}$=patch leads by 2.8 mIoU, indicating that spectral-overlap failures persist on CNNs when corruptions (like fog) heavily attenuate informative bands.

### A.7.3 Per-Corruption Architecture Profiles

Figure 8 decomposes aggregate errors into per-architecture profiles. Under $\mathcal{F}$=patch, CNNs form a high-error band while ViTs and ConvNeXts achieve lower error without catastrophic spikes. Under $\mathcal{F}$=low-freq, ConvNeXt backbones collapse starting at defocus blur, while ViT-L/16 remains stable, confirming that large transformer capacity can absorb global spectral perturbations. These profiles demonstrate that the ConvNeXt

Table 10: Segmentation results on CarlaTTA under CTTA. Entries report per-class IoU (%) and mean IoU. We use DeepLabV2 as the segmentation model.

(a) day2night

| Method | Road | Sidewalk | Building | Wall | Fence | Pole | Traffic Light | Traffic Sign | Vegetation | Terrain | Sky | Person | Vehicle | Roadline | Mean↑ |
|---|---|---|---|---|---|---|---|---|---|---|---|---|---|---|---|
| Source | 91.74 | 72.40 | 83.07 | 47.91 | 21.05 | 44.81 | 71.31 | 52.76 | 71.38 | 16.08 | 33.29 | 69.79 | 78.90 | 62.60 | 58.3 |
| Tent | 96.00 | 83.79 | 84.24 | 55.78 | 13.72 | 45.54 | 69.21 | 55.39 | 74.74 | 13.66 | 33.44 | 68.86 | 88.50 | 75.80 | 61.3 |
| CoTTA | 96.21 | 83.62 | 84.29 | 55.86 | 12.52 | 45.78 | 69.31 | 55.22 | 75.15 | 13.43 | 33.37 | 68.66 | 88.48 | 75.58 | 61.2 |
| SAR | 96.10 | 83.67 | 84.27 | 57.10 | 15.31 | 45.90 | 69.82 | 55.89 | 75.19 | 14.77 | 33.53 | 68.72 | 88.26 | 75.86 | 61.7 |
| DeYO | 96.11 | 83.61 | 84.28 | 57.57 | 15.89 | 45.93 | 69.98 | 55.98 | 75.37 | 14.78 | 33.55 | 68.57 | 87.84 | 75.86 | 61.8 |
| ROID | 96.52 | 82.83 | 83.24 | 49.29 | 9.22 | 42.99 | 66.51 | 53.31 | 69.62 | 14.01 | 33.48 | 68.73 | 91.55 | 74.59 | 59.7 |
| M2A($\mathcal{F}$=low-freq) | 96.62 | 83.23 | 84.49 | 58.95 | 16.57 | 45.86 | 70.15 | 56.10 | 75.88 | 13.88 | 34.14 | 69.00 | 89.98 | 75.86 | 62.2 |
| M2A($\mathcal{F}$=patch) | 96.49 | 82.61 | 84.58 | 55.92 | 15.61 | 46.32 | 69.87 | 55.99 | 75.47 | 14.96 | 34.77 | 69.34 | 90.79 | 75.88 | 62.0 |

(b) clear2fog

| Method | Road | Sidewalk | Building | Wall | Fence | Pole | Traffic Light | Traffic Sign | Vegetation | Terrain | Sky | Person | Vehicle | Roadline | Mean↑ |
|---|---|---|---|---|---|---|---|---|---|---|---|---|---|---|---|
| Source | 85.50 | 62.74 | 52.71 | 46.19 | 24.33 | 40.77 | 55.86 | 51.93 | 51.54 | 24.13 | 15.66 | 66.85 | 81.66 | 78.64 | 52.7 |
| Tent | 84.24 | 77.01 | 71.69 | 43.71 | 18.50 | 44.06 | 65.71 | 56.26 | 58.00 | 22.61 | 39.51 | 67.16 | 59.60 | 70.77 | 55.6 |
| CoTTA | 87.83 | 77.36 | 72.78 | 44.67 | 16.63 | 45.54 | 65.70 | 56.80 | 58.50 | 22.00 | 39.19 | 67.45 | 64.78 | 72.13 | 56.5 |
| SAR | 84.75 | 77.08 | 71.80 | 43.95 | 19.50 | 44.59 | 66.38 | 56.85 | 58.32 | 23.88 | 39.91 | 67.66 | 60.13 | 71.10 | 56.1 |
| DeYO | 86.52 | 77.18 | 71.38 | 43.80 | 20.08 | 45.00 | 66.63 | 57.10 | 58.72 | 23.91 | 38.57 | 67.87 | 62.46 | 71.70 | 56.4 |
| ROID | 80.47 | 75.66 | 73.01 | 43.87 | 15.28 | 43.29 | 63.34 | 54.44 | 55.78 | 21.84 | 36.07 | 65.72 | 52.87 | 66.63 | 53.4 |
| M2A($\mathcal{F}$=low-freq) | 84.90 | 76.86 | 69.74 | 44.31 | 21.01 | 45.07 | 66.50 | 56.78 | 58.86 | 23.70 | 38.41 | 67.42 | 53.66 | 70.85 | 55.6 |
| M2A($\mathcal{F}$=patch) | 91.46 | 77.48 | 71.63 | 46.10 | 19.53 | 45.79 | 66.51 | 57.50 | 59.41 | 24.40 | 34.97 | 68.38 | 82.16 | 73.01 | 58.4 |

(c) clear2rain

| Method | Road | Sidewalk | Building | Wall | Fence | Pole | Traffic Light | Traffic Sign | Vegetation | Terrain | Sky | Person | Vehicle | Roadline | Mean↑ |
|---|---|---|---|---|---|---|---|---|---|---|---|---|---|---|---|
| Source | 94.90 | 85.52 | 88.84 | 72.60 | 33.66 | 56.70 | 83.50 | 67.96 | 81.64 | 16.66 | 71.58 | 78.18 | 92.68 | 79.85 | 71.7 |
| Tent | 95.56 | 87.24 | 90.00 | 72.72 | 26.05 | 54.61 | 81.27 | 65.73 | 80.72 | 22.34 | 68.47 | 75.04 | 89.80 | 80.60 | 70.7 |
| CoTTA | 95.76 | 87.18 | 90.08 | 72.03 | 22.45 | 54.82 | 81.28 | 65.56 | 81.16 | 21.72 | 69.03 | 74.87 | 90.58 | 80.50 | 70.5 |
| SAR | 95.74 | 87.20 | 90.21 | 73.25 | 26.98 | 55.11 | 81.61 | 66.17 | 80.92 | 23.38 | 69.72 | 75.20 | 90.25 | 80.69 | 71.1 |
| DeYO | 95.76 | 87.16 | 90.25 | 73.51 | 27.20 | 55.17 | 81.65 | 66.21 | 80.95 | 23.60 | 70.01 | 75.18 | 90.19 | 80.69 | 71.2 |
| ROID | 96.36 | 86.45 | 89.41 | 69.44 | 22.44 | 53.03 | 80.25 | 64.58 | 79.24 | 22.53 | 67.18 | 74.70 | 92.59 | 79.71 | 69.8 |
| M2A($\mathcal{F}$=low-freq) | 96.44 | 87.14 | 90.84 | 74.61 | 28.33 | 55.44 | 81.87 | 66.48 | 81.46 | 22.39 | 72.81 | 75.45 | 91.13 | 80.79 | 71.8 |
| M2A($\mathcal{F}$=patch) | 96.61 | 86.32 | 90.92 | 73.45 | 27.44 | 56.33 | 81.81 | 66.48 | 81.30 | 24.75 | 74.84 | 75.34 | 93.40 | 80.70 | 72.1 |

(d) dynamic

| Method | Road | Sidewalk | Building | Wall | Fence | Pole | Traffic Light | Traffic Sign | Vegetation | Terrain | Sky | Person | Vehicle | Roadline | Mean↑ |
|---|---|---|---|---|---|---|---|---|---|---|---|---|---|---|---|
| Source | 75.0 | 55.3 | 68.9 | 35.2 | 19.4 | 38.3 | 62.8 | 47.6 | 55.5 | 5.0 | 20.1 | 63.8 | 55.6 | 50.2 | 46.6 |
| Tent | 78.0 | 65.8 | 72.5 | 38.1 | 13.2 | 39.7 | 63.7 | 51.1 | 57.9 | 4.1 | 29.7 | 60.0 | 65.2 | 61.4 | 50.0 |
| CoTTA | 78.6 | 63.7 | 69.0 | 27.0 | 8.0 | 35.1 | 58.6 | 45.8 | 46.4 | 3.4 | 29.9 | 49.9 | 71.1 | 58.4 | 46.0 |
| SAR | 81.6 | 65.9 | 73.0 | 42.8 | 16.7 | 41.7 | 66.3 | 53.8 | 59.9 | 5.7 | 30.7 | 61.3 | 68.9 | 62.3 | 52.2 |
| DeYO | 81.8 | 65.8 | 73.1 | 43.0 | 17.0 | 41.8 | 66.4 | 53.9 | 60.1 | 5.7 | 30.9 | 61.0 | 68.9 | 62.4 | 52.3 |
| ROID | 80.4 | 64.0 | 69.5 | 32.1 | 9.7 | 34.6 | 59.4 | 48.6 | 48.7 | 4.8 | 29.8 | 58.0 | 75.4 | 59.4 | 48.2 |
| M2A($\mathcal{F}$=low-freq) | 85.2 | 66.4 | 74.0 | 43.3 | 16.3 | 41.9 | 66.6 | 54.0 | 60.6 | 5.2 | 31.4 | 60.6 | 70.5 | 62.9 | 52.8 |
| M2A($\mathcal{F}$=patch) | 84.7 | 65.6 | 73.9 | 40.9 | 15.3 | 41.8 | 65.5 | 53.7 | 59.0 | 5.9 | 31.9 | 61.7 | 80.1 | 62.4 | 53.0 |

failure is a sharp transition triggered by specific spectral overlaps, while the ViT advantage is consistent across the full corruption spectrum.

### A.7.4    MRSFFIA-C: Dataset and Experimental Setup

The clean MRSFFIA dataset (Du et al., 2024) consists of top-down video frames of *Punctatus Oplegnathus* cultured in a recirculating tank (diameter 3 m, depth 0.75 m) in Yantai City, Shandong Province, China. The task, feeding-intensity recognition, targets a core aquaculture monitoring problem: estimating feeding activity from overhead cameras to support operational decisions (e.g., feeding control) without manual observation. Accurate, automated feeding-intensity estimates directly affect feed efficiency, water quality, and labor costs

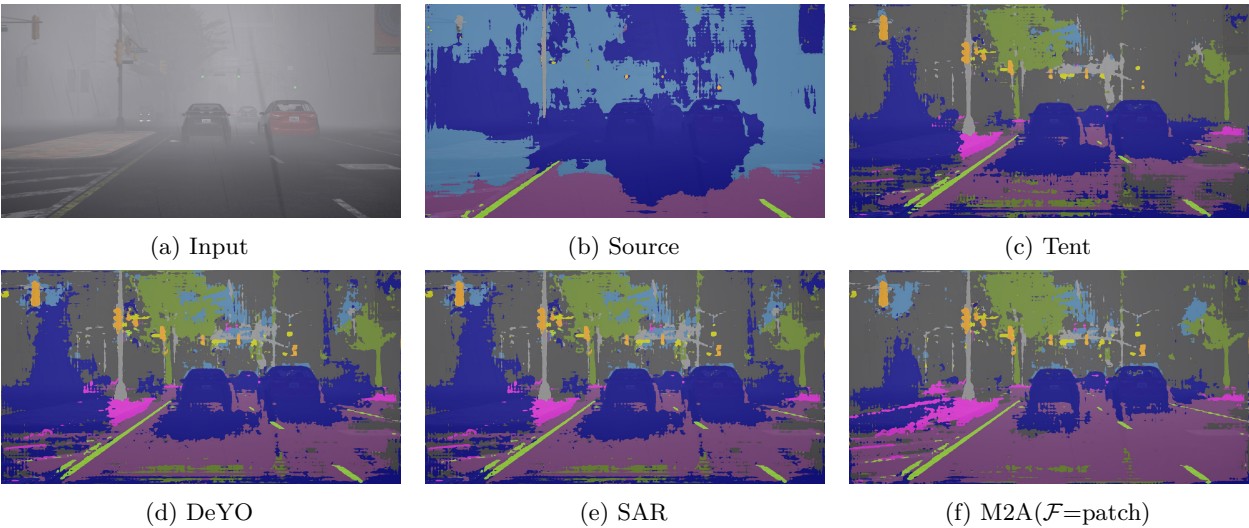

(a) Input      (b) Source      (c) Tent

(d) DeYO      (e) SAR      (f) M2A($\mathcal{F}$=patch)

Figure 7: Visualization of the segmentation results in CarlaTTA under CTTA. We use DeepLabV2 as the segmentation model.

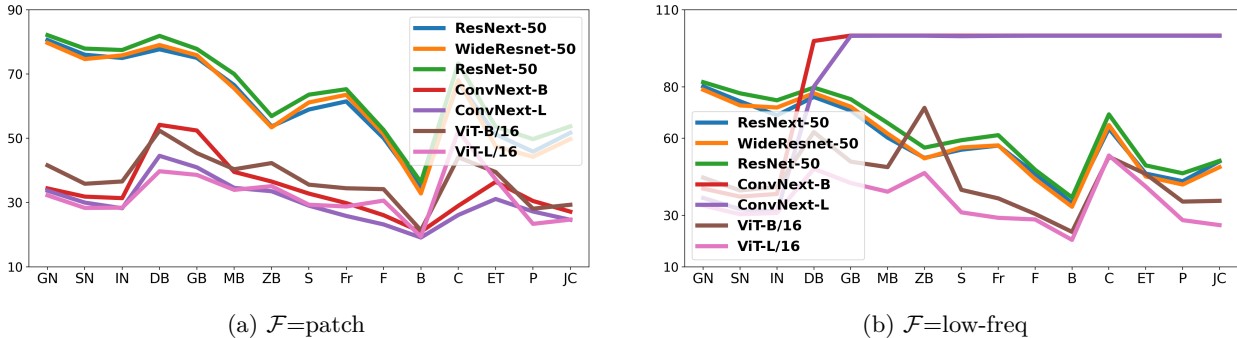

(a) $\mathcal{F}$=patch      (b) $\mathcal{F}$=low-freq

Figure 8: Per-corruption error profiles across seven architectures on ImageNet-C under CTTA, extending Table 5. (a) $\mathcal{F}$=patch: CNNs cluster at high error with a shared profile shape, while ViTs and ConvNeXts achieve substantially lower error; the architecture gap dominates the corruption axis. (b) $\mathcal{F}$=low-freq: ConvNeXt-B/L collapse to near-chance error from defocus blur onward, while ViT-L/16 remains competitive; traditional CNNs and ViT-B/16 occupy a middle band. Corruptions are ordered from Gaussian noise (GN) to JPEG compression (JC).

in commercial recirculating systems, making this setting a representative testbed for aquaculture-relevant vision models. A total of 100 fish, each weighing approximately 200 g, were recorded during feeding sessions. The dataset is split into 6 091 training, 760 validation, and 760 test images across four feeding-intensity classes: *none*, *weak*, *medium*, and *strong*.

**Source-model fine-tuning.** We fine-tuned ViT-B/16 and ViT-L/16 on the MRSFFIA training set, resizing all images to 384×384. Training used the Adam optimizer with learning rate $10^{-4}$, weight decay 0.0, early stopping with patience 10, and a maximum of 100 epochs; the checkpoint with the best validation accuracy was selected. The resulting source models achieve 90.2% (ViT-B/16) and 91.0% (ViT-L/16) accuracy on the clean test set.

**Corruption generation (MRSFFIA-C).** To construct the CTTA benchmark, we applied the 15 corruption types from Hendrycks & Dietterich (2018) (identical to those used in ImageNet-C and CIFAR-C) to the 760 test images at severities 1, 3, and 5, using the publicly available corruption codebase.

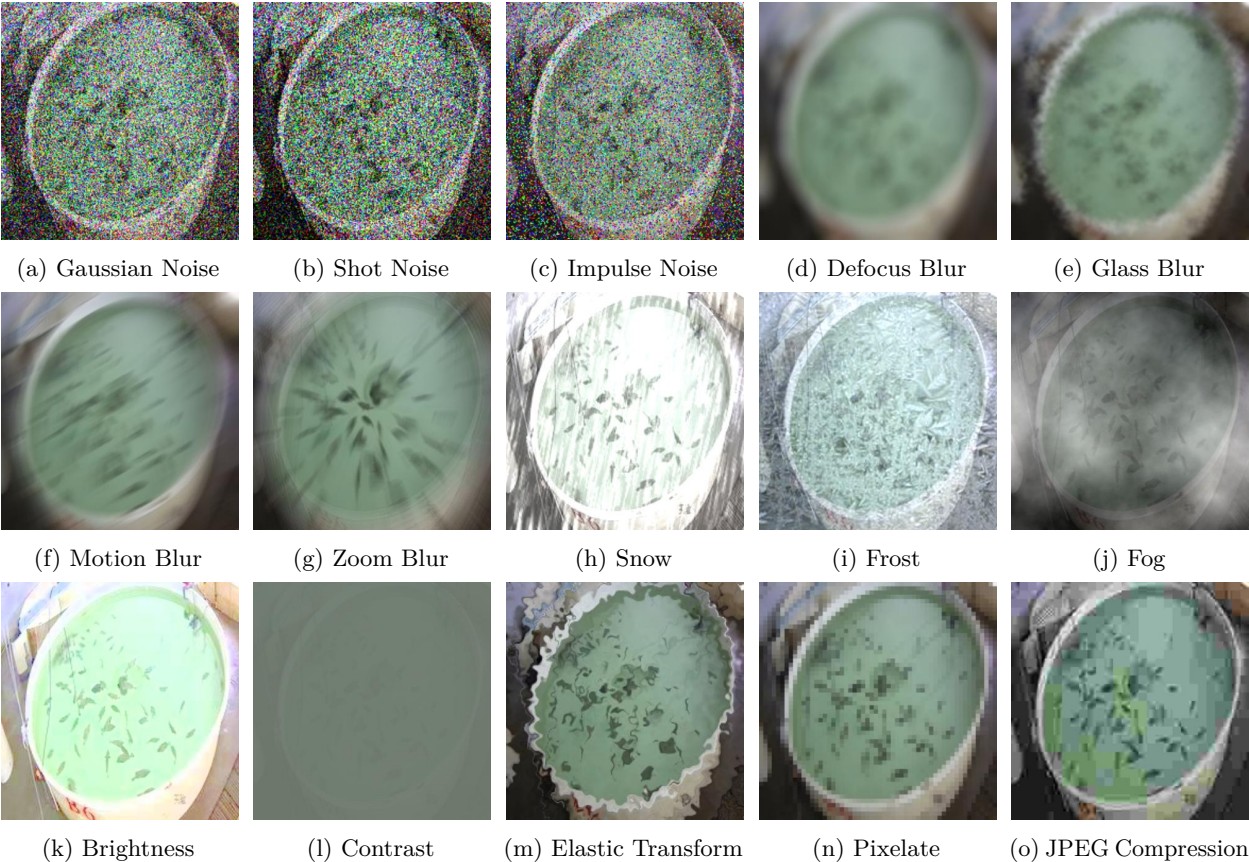

| | | | | |
|---|---|---|---|---|
| (a) Gaussian Noise | (b) Shot Noise | (c) Impulse Noise | (d) Defocus Blur | (e) Glass Blur |
| (f) Motion Blur | (g) Zoom Blur | (h) Snow | (i) Frost | (j) Fog |
| (k) Brightness | (l) Contrast | (m) Elastic Transform | (n) Pixelate | (o) JPEG Compression |

Figure 9: Samples of MRSFFIA-C dataset at severity level 5 (highest) with different corruption domains applied, following CIFAR-C and ImageNet-C corruptions.

### A.7.5 Fine-Grained Behavior Recognition

This subsection provides qualitative context for the MRSFFIA-C results (Table 7). Figure 9 shows sample frames across corruption types used to stress-test models intended for deployment in aquaculture facilities, exposing them to degradations such as turbidity, glare, motion blur, and compression artifacts that arise in tank-side cameras and remote video links. Discriminative cues in this task are collective and textural (fish density, movement patterns) rather than object-localized. At high severity, many corruptions obliterate these global features. Because the task lacks spatially localized class structure, frequency perturbations generate comparably informative views to patch masking, resulting in the narrower family gap reported and clarifying how CTTA robustness on MRSFFIA-C translates to more reliable, hands-off feeding monitoring in practice.

### A.7.6 Feature-Space Analysis

Figures 11 visualize t-SNE projections on CIFAR-10-C and MRSFFIA-C. On CIFAR-10-C, classes form tight clusters that persist even at high severity, providing the structural basis for patch masking's advantage. On MRSFFIA-C, classes are partially intermixed even at severity one, and cluster structure is largely destroyed at severity five. The narrowing of the family gap on fine-grained tasks corresponds to this lack of inherent spatial class separation in the feature space.

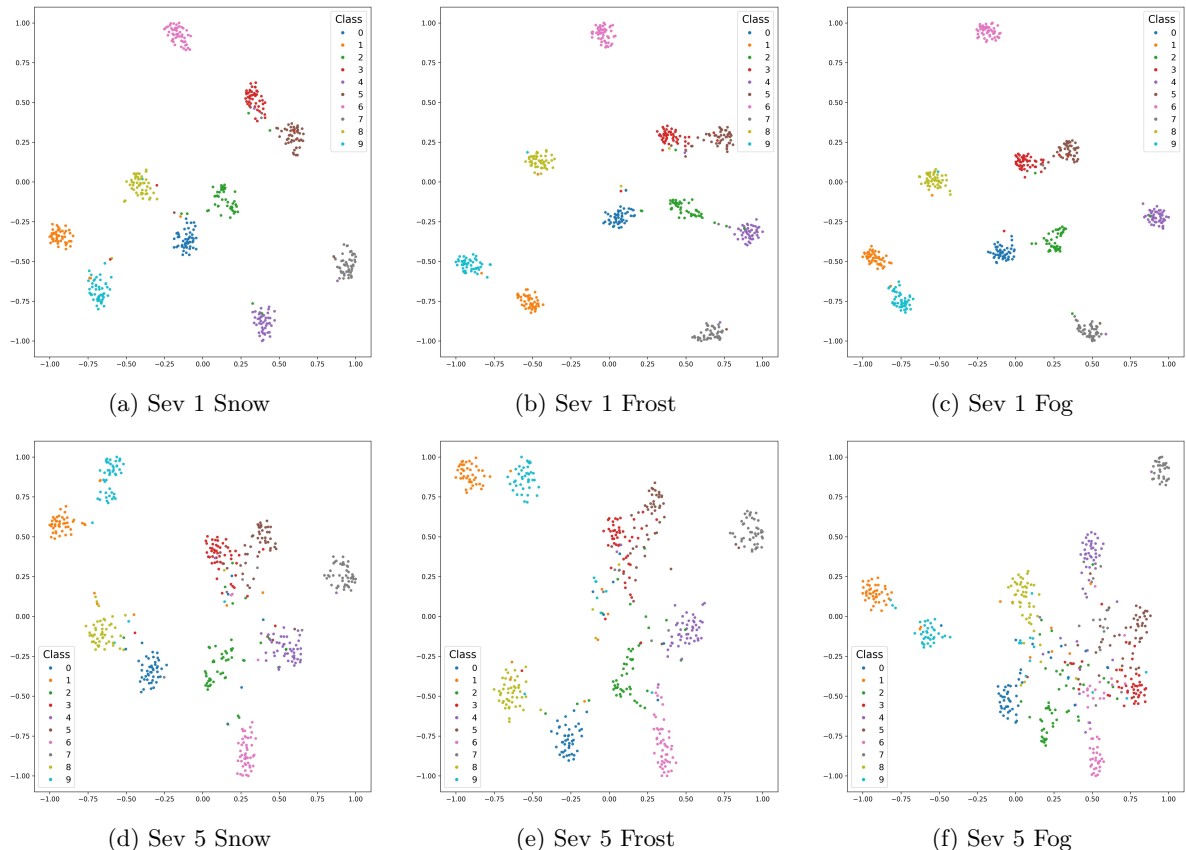

(a) Sev 1 Snow    (b) Sev 1 Frost    (c) Sev 1 Fog

(d) Sev 5 Snow    (e) Sev 5 Frost    (f) Sev 5 Fog

Figure 10: Feature visualizations of CIFAR10-C at severity level 1 and 5 obtained from ViT-B/16 (source) model.

## A.8 Mechanistic Analyses and Ablations

### A.8.1 Hyperparameter Robustness

This subsection provides loss diagnostics and hyperparameter sweeps for CIFAR-10-C (Figure 12, 13, 14). Figure 12 ablates the loss components on CIFAR-10-C and ImageNet-C: M2A (CL+EL) and a REM-style variant with CL+RL and random patch selection exhibit nearly identical error profiles, whereas removing CL causes catastrophic collapse in all cases, indicating that stability is driven by the combination of patch masking and consistency rather than by the particular heuristic used to score patches. Number of patches reveals a stability–collapse transition when granularity approaches the pixel level, and broad hyperparameter stability across lambda, mask views, and batch sizes suggests that adaptation effectiveness is an intrinsic property of the masking family rather than of a specific scoring heuristic.

### A.8.2 Masking Families

Figure 15 extends the family comparison to alternative CTTA training strategies by integrating Model Recovery, Sample Filtering, and their combination into M2A, following the ideas implemented in EATA (Niu et al., 2022) and SAR (Niu et al., 2023). These strategies reduce absolute error for several families and suppress some of the most severe corruption-specific spikes, most noticeably when Model Recovery and Sample Filtering are combined, but they do not overturn the overall family ordering: patch masking remains the most stable and lowest-error family overall, while pixel and frequency masking remain more corruption-sensitive. This suggests that alternative update heuristics can improve robustness, yet the masking family continues to be the dominant factor governing stability.

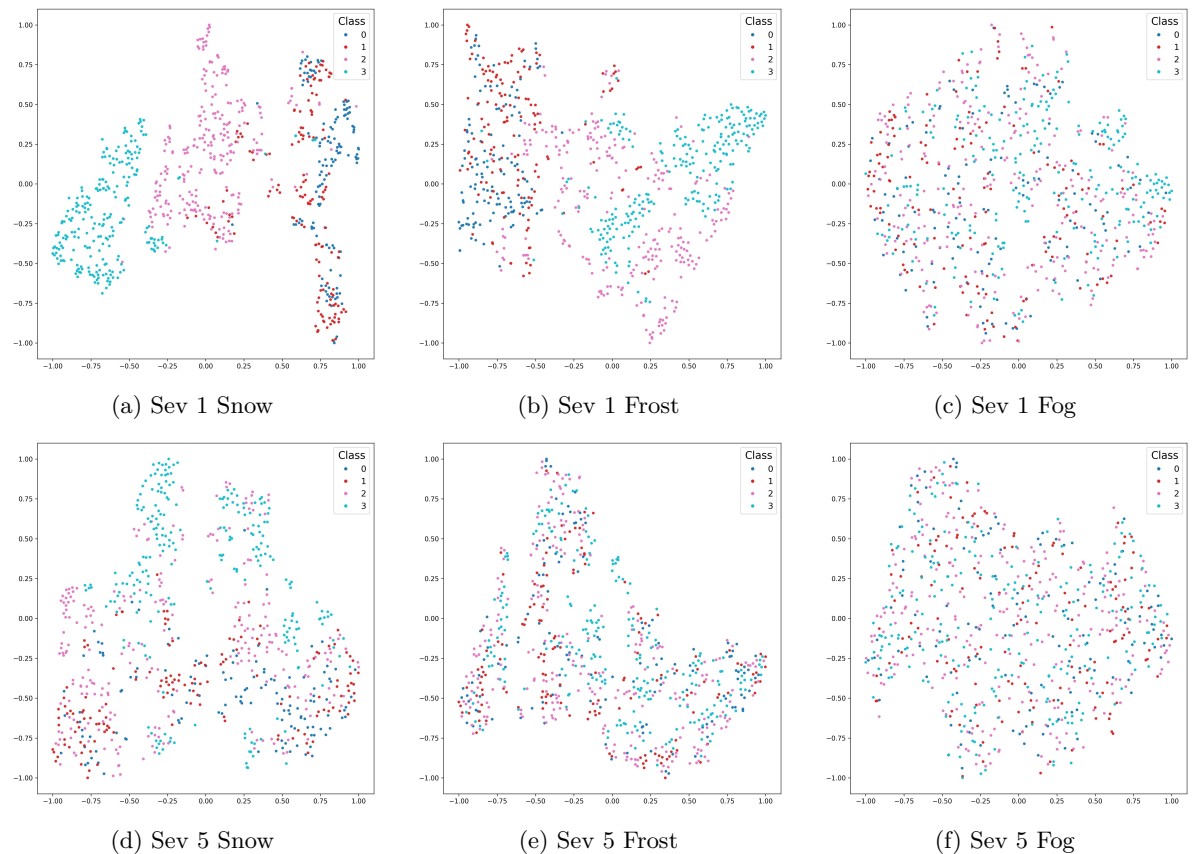

(a) Sev 1 Snow  (b) Sev 1 Frost  (c) Sev 1 Fog

(d) Sev 5 Snow  (e) Sev 5 Frost  (f) Sev 5 Fog

Figure 11: Feature visualizations of MRSFFIA-C at severity level 1 and 5 obtained from ViT-B/16 (source) model.

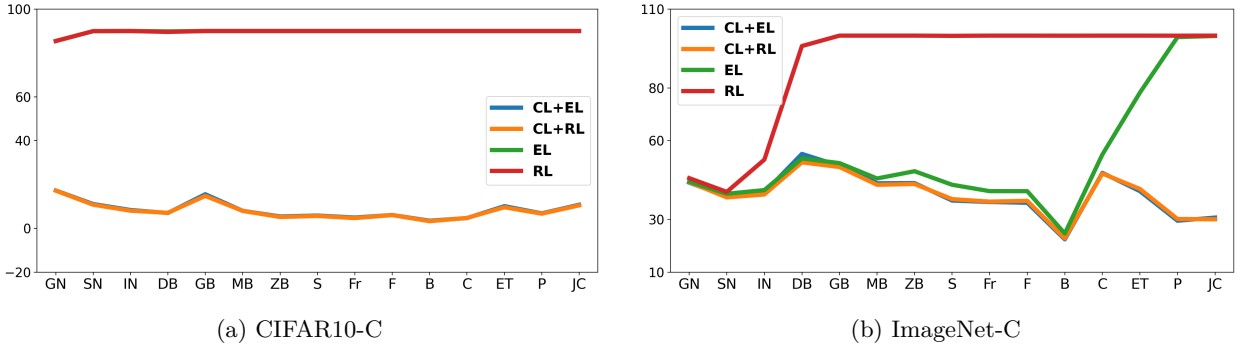

(a) CIFAR10-C

(b) ImageNet-C

Figure 12: Ablations on loss functions: Consistency Loss (CL), Entropy Loss (EL), and Ranking Loss (RL) on CIFAR10-C and ImageNet-C under CTTA with a ViT-B/16 backbone.

Furthermore, we provide a brief analysis of how masking families interact with different CTTA training schemes and help assess whether our conclusions generalize beyond a single objective. Figure 16 reports classification error rate (%) for different families under two loss combinations: CL+EL, which is the objective used by M2A, and CL+RL, which matches the loss design used in REM. To isolate the effect of the masking family rather than the scoring heuristic, we use the same family-comparison setting across both objectives. The key result is consistent in both subfigures: patch masking remains the most stable family, while the weaker families preserve their relative disadvantages. This suggests that the superiority of patch masking is not an artifact of the specific M2A loss, but persists under an alternative CTTA training scheme as well.

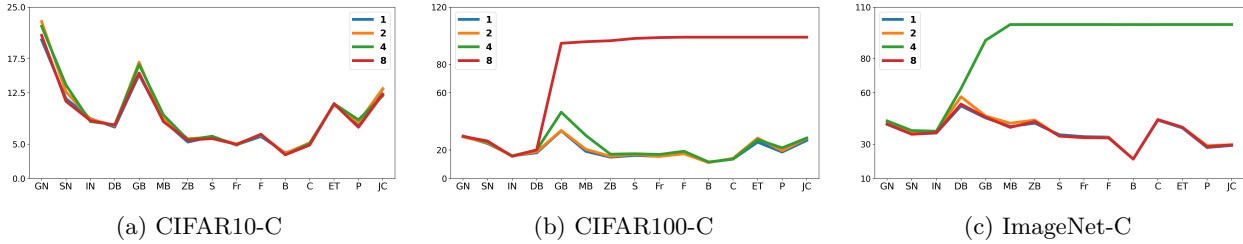

(a) CIFAR10-C        (b) CIFAR100-C        (c) ImageNet-C

Figure 13: M2A($\mathcal{F}$=patch) classification error under CTTA as the number of masked patches increases. Performance is stable at low counts but catastrophically collapses once patches exceed a dataset-dependent threshold (eight on CIFAR-100-C, four on ImageNet-C), consistent with fine-grained patches approaching pixel-level noise and destroying spatial coherence. CIFAR-10-C remains robust across all tested values.

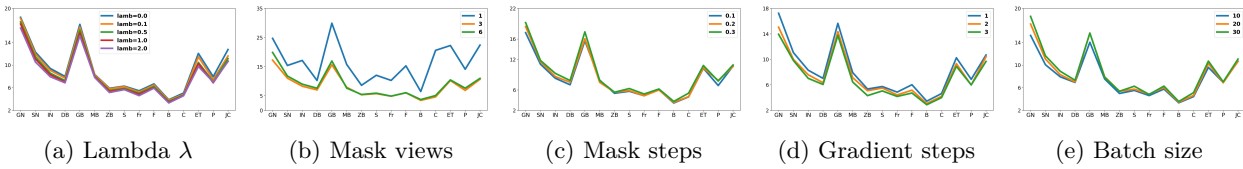

(a) Lambda $\lambda$  (b) Mask views  (c) Mask steps  (d) Gradient steps  (e) Batch size

Figure 14: M2A($\mathcal{F}$=patch) ablations on CIFAR10-C under CTTA. Each panel shows the effect of a single hyperparameter: (a) Lambda $\lambda$, (b) mask views, (c) mask steps, (d) gradient steps, and (e) batch size.

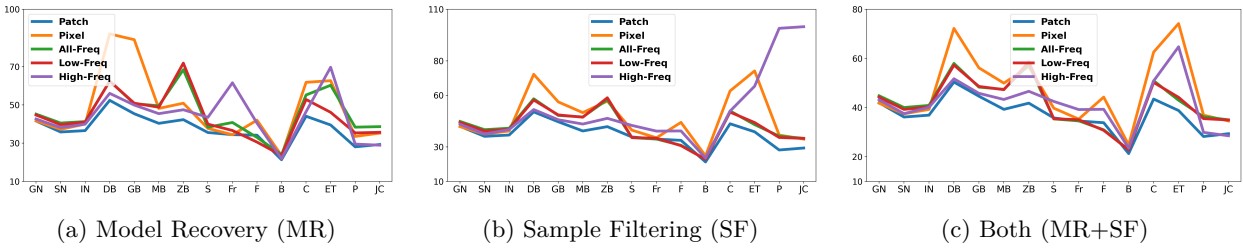

(a) Model Recovery (MR)    (b) Sample Filtering (SF)    (c) Both (MR+SF)

Figure 15: Masking Families under alternative CTTA training strategy using ImageNet-C, reported as classification error rate (%). Model Recovery and Sample Filtering are ideas from EATA (Niu et al., 2022) and SAR (Niu et al., 2023).

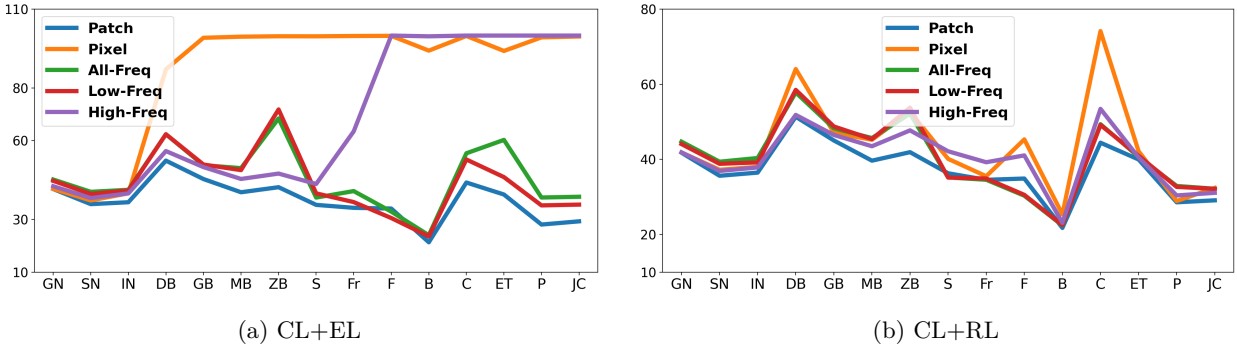

(a) CL+EL            (b) CL+RL

Figure 16: Masking families under different CTTA loss functions using ImageNet-C with ViT-B/16 backbone, reported as classification error rate (%). CL, EL, and RL denote Consistency Loss, Entropy Loss, and Ranking Loss, respectively.

Figure 17 complements this by also reporting classification error rate (%) while varying the number of test-time optimization steps, where the optimization step count is the number of gradient updates applied during CTTA for each incoming batch. Across optim_steps $\in \{1, 2, 3\}$, patch masking again shows consistently stable behavior, whereas the less robust families remain more sensitive as optimization becomes more aggressive. Thus, changing the update strategy affects the magnitude of adaptation, but does not overturn the family

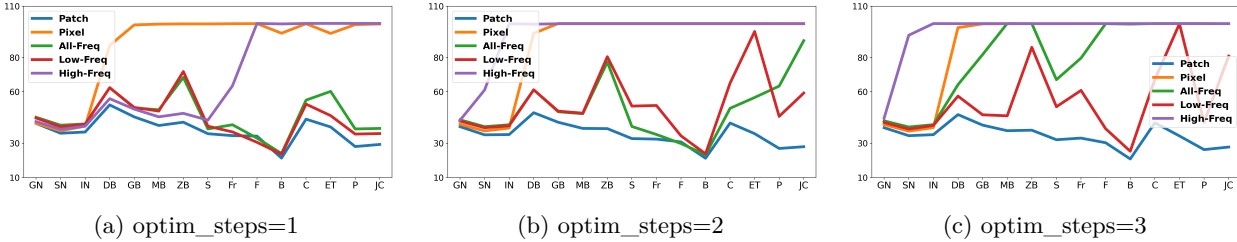

Figure 17: Masking families when the number of optimization steps is varied under CTTA using ImageNet-C with ViT-B/16 backbone, reported as classification error rate (%).

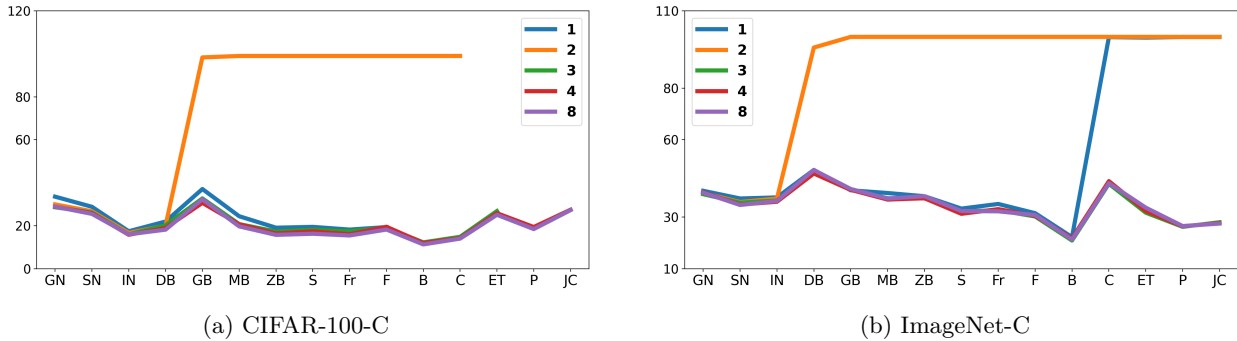

Figure 18: Effect of *very small* batch sizes on M2A($\mathcal{F}$=patch) under CTTA. Each curve sweeps batch sizes $\{1, 2, 3, 4, 8\}$ over the 15 corruptions. While moderate batches (3, 4, 8) stay in a similar performance range, batches of size 1–2 can trigger corruption-specific collapse, with error approaching the random-chance ceiling on some corruptions and datasets.

ordering: patch masking stays reliable across all tested optimization depths. Together, Figures 16 and 17 indicate that the observed family effect is generalizable across both alternative loss formulations and alternative update schedules.

### A.8.3 Small-Batch Behavior

Figure 18 extends the batch-size ablations to the *very small* regime on CIFAR-100-C and ImageNet-C, sweeping batch sizes $\{1, 2, 3, 4, 8\}$. While moderate batches (3, 4, 8) remain in a similar error range, batches of size 1–2 can trigger corruption-specific spikes, with error reaching the random-chance ceiling on some corruptions. This confirms that extremely small batches provide too little signal for the consistency loss, and that the stability holds once batch size is large enough for reliable gradient estimates.

### A.8.4 Domain Generalization Details

Table 11 reports the full CIFAR-10-C and ImageNet-C domain-generalization errors, including mean $\pm$ standard deviation over seeds, underlying the aggregated results in Table 2.

### A.8.5 Low-Frequency Masking Ablations

We repeat the hyperparameter sweeps with $\mathcal{F}$=low-freq (Figure 19). Unlike patch masking, no tuning configuration eliminates corruption-dependent spikes, particularly under blur-type corruptions. Additional gradient steps amplify these spikes, suggesting that optimization on degenerate views pushes the model further from discriminative representations. The instability is intrinsic to the family and irreducible via hyperparameter calibration.

Table 11: Domain generalization (forward transfer) under CTTA using ViT-B/16 backbone. Models adapt on 10 seen corruptions and are then evaluated without further adaptation on 5 unseen corruptions.

(a) CIFAR10-C

| Method | Directly test on unseen domains | | | | | Mean↓ | Gain |
|---|---|---|---|---|---|---|---|
| | B | C | ET | P | JC | | |
| Source | 5.3 | 49.7 | 23.6 | **24.7** | 23.1 | 25.2 | 0.0 |
| Tent | 5.1 | 34.1 | 22.1 | 27.7 | 22.4 | 22.2 | +3.0 |
| CoTTA | 20.4 | 83.5 | 52.1 | 35.3 | 44.0 | 47.0 | −21.8 |
| SAR | 5.3 | 49.2 | 23.5 | 24.8 | 23.1 | 25.1 | +0.1 |
| RoTTA | **3.9** | 8.0 | 14.3 | 32.3 | 21.4 | 15.9 | +9.3 |
| DeYO | 4.7 | 18.6 | 16.0 | 26.1 | **18.4** | 16.7 | +8.5 |
| ROID | 4.7 | 16.1 | 18.2 | 35.3 | 21.9 | 19.2 | +6.0 |
| REM | 5.5 | 13.7 | **13.8** | 24.9 | 22.4 | 16.0 | +9.2 |
| M2A($\mathcal{F}$=low-freq) | 6.4±0.1 | 12.0±0.5 | 18.9±0.8 | 36.3±1.5 | 25.2±0.6 | 19.8 | +5.4 |
| M2A($\mathcal{F}$=high-freq) | 89.9±0.2 | 90.0±0.0 | 90.0±0.0 | 90.0±0.0 | 90.0±0.0 | 90.0 | -64.8 |
| M2A($\mathcal{F}$=all-freq) | 6.5±0.1 | 11.7±0.4 | 18.8±0.8 | 34.5±2.7 | 24.7±0.8 | 19.2 | +6.0 |
| M2A($\mathcal{F}$=pixel) | 33.9±48.5 | 36.6±46.3 | 41.6±42.0 | 50.8±34.0 | 46.7±37.5 | 41.9 | -16.7 |
| M2A($\mathcal{F}$=patch) | 5.1±0.3 | **7.5±0.6** | **13.8±0.4** | 31.0±2.7 | 20.7±0.7 | **15.6** | +9.6 |

(b) ImageNet-C

| Method | Directly test on unseen domains | | | | | Mean↓ | Gain |
|---|---|---|---|---|---|---|---|
| | B | C | ET | P | JC | | |
| Source | 24.8 | 91.1 | 55.7 | 39.1 | 36.7 | 49.4 | 0.0 |
| Tent | 23.4 | 88.5 | 51.7 | 35.4 | 35.8 | 46.9 | +2.5 |
| CoTTA | 24.5 | 90.2 | 54.5 | 37.5 | 34.8 | 48.3 | +1.2 |
| SAR | 23.4 | 79.7 | 48.4 | **35.2** | 33.8 | 44.1 | +5.4 |
| RoTTA | 24.4 | 64.6 | 49.3 | 36.6 | **33.4** | 41.7 | +7.8 |
| DeYO | 23.6 | 74.8 | **48.3** | 37.0 | 35.3 | 43.8 | +5.7 |
| ROID | **22.3** | 75.6 | 49.7 | 37.4 | 34.8 | 43.9 | +5.5 |
| REM | 23.6 | 60.2 | 51.6 | 37.2 | 37.3 | 41.9 | +7.5 |
| M2A($\mathcal{F}$=low-freq) | 25.4±0.3 | 65.0±2.4 | 56.2±0.3 | 43.5±0.5 | 39.9±0.3 | 46.0 | +3.5 |
| M2A($\mathcal{F}$=high-freq) | 91.3±11.2 | 99.6±0.1 | 96.6±5.4 | 95.0±4.4 | 93.1±5.5 | 95.1 | -45.7 |
| M2A($\mathcal{F}$=all-freq) | 26.1±0.5 | 65.4±0.4 | 59.2±1.7 | 44.5±1.3 | 40.4±0.5 | 47.1 | +2.4 |
| M2A($\mathcal{F}$=pixel) | 98.6±0.7 | 99.8±0.0 | 99.8±0.1 | 99.4±0.3 | 98.9±0.6 | 99.3 | -49.8 |
| M2A($\mathcal{F}$=patch) | 22.8±0.2 | **56.3±1.2** | 51.5±1.7 | 38.4±1.1 | **37.2±0.5** | **41.3** | +8.2 |

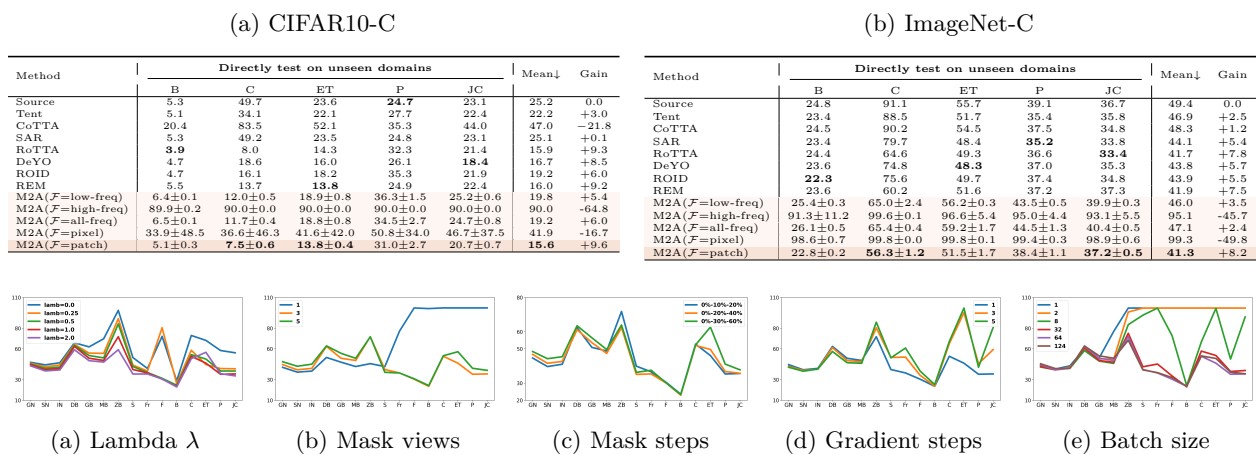

(a) Lambda $\lambda$   (b) Mask views   (c) Mask steps   (d) Gradient steps   (e) Batch size

Figure 19: M2A($\mathcal{F}$=low-freq) ablations on ImageNet-C under CTTA. Each panel shows the effect of a single hyperparameter: (a) Lambda $\lambda$, (b) mask views, (c) mask steps, (d) gradient steps, and (e) batch size. Compare with Figure 5 ($\mathcal{F}$=patch).

### A.8.6 Sample Trajectories

Figure 20 tracks least-confident samples across sequential domains under $\mathcal{F}$=patch. Initially misclassified samples often receive correct predictions as adaptation progresses through subsequent domains. This trajectory illustrates cumulative adaptation, where the model builds a progressively more robust representation that retroactively improves handling of difficult cases.

### A.9 Mask Selection Heuristics

We investigate why M2A can outperform other patch-masking approaches such as Continual-MAE and REM despite sharing the same broad masking family. Our view is that these alternatives often entangle the masking family with architecture-driven masking heuristics, whereas here we explicitly keep the masking family fixed to a contiguous patch and vary only the selection strategy $\mathcal{S}$. Random patch selection is the default M2A design described in Section 3; to make the comparison architecture-agnostic, we additionally implemented Center and Uncertainty patch selection in the same codebase while keeping the family $\mathcal{F}$, the loss, and the adaptation pipeline unchanged.

#### A.9.1 Center Selection

This forces binary spatial patch masking with a single contiguous square at the center of the image. For masked fraction $m_t$, the target area is $m_t HW$, which is converted into a square side length and placed at the image center:

$$s_t = \max\big(1, \min(\lfloor \sqrt{m_t HW} \rfloor, \min(H, W))\big), \qquad M_{\text{center}}^{(t)}(i,j) = \mathbf{1}\big[ \tfrac{H-s_t}{2} \le i < \tfrac{H+s_t}{2}, \ \tfrac{W-s_t}{2} \le j < \tfrac{W+s_t}{2} \big]. \tag{23}$$

This preserves the same contiguous patch family as the default method, but removes the randomness in the location of the masked region. In this sense, Center isolates whether a deterministic geometry alone can improve CTTA once the masking family is already fixed to a token-aligned contiguous block.

#### A.9.2 Uncertainty Selection

This also uses one contiguous square, but its location is selected by a local uncertainty score computed directly from the current input image. The image is converted to grayscale, normalized to $[0, 1]$, quantized

into 16 intensity bins, and a $9 \times 9$ local histogram is estimated by average pooling. The entropy map and the selected patch location are then

$$E_x(i,j) = -\sum_{b=1}^{16} p_b(i,j) \log\big(p_b(i,j) + 10^{-12}\big), \qquad (y_t, x_t) = \arg\max_{(y,x)} \text{AvgPool}_{s_t}\big(E_x\big)(y,x), \qquad (24)$$

and the final mask is the contiguous square of side $s_t$ anchored at $(y_t, x_t)$. Thus Uncertainty changes only the selection rule $\mathcal{S}$ by steering the mask toward locally heterogeneous regions, while keeping the masking family itself identical to contiguous patch masking.

**Empirical analysis.**   Figure 21 shows that these heuristic selection rules change performance only marginally on both ImageNet-C and CIFAR-10-C: the three curves remain close, with no consistent advantage for Uncertainty or Center over the default Random strategy. Our central view is that "getting $\mathcal{F}$ right" matters more than adding heuristic scoring: once the masking family is fixed, simple random selection combined with prediction-centric losses already captures most of the stability and performance we observe. This helps explain why M2A can remain competitive with other patch-masking methods: the dominant factor is not a sophisticated selection heuristic by itself, but the use of a structurally safe masking family within a stable CTTA objective.

### A.10   Limitations and Scope

**Selection strategy constraints.**   By fixing $\mathcal{S}$=random to isolate the family axis, the study does not explore the full factorial interaction between masking family, selection strategy, and masking schedule. It is possible that a corruption-aware or spectral-band-aware selection strategy could stabilize frequency masking by avoiding bands that overlap with the corruption's damage zone. Our results therefore characterize frequency masking under random selection; its upper-bound performance under optimized $\mathcal{S}$ remains an open question.

**Information-removal asymmetry.**   Equal masked ratios do not imply equal information removal. Zeroing a fraction of spatial patches removes localized content while leaving the rest intact, whereas zeroing frequency coefficients redistributes energy globally, altering every pixel. This asymmetric information loss means frequency masking may remove more task-relevant structure even at identical ratios. Quantifying this asymmetry—e.g., via mutual-information estimates—could reveal if frequency masking's instability stems partly from this spectral redistribution, and whether family-specific ratio calibration would narrow the performance gap predicted by the structural-preservation principle.

**Architecture specificity.**   The finding that spatial masking is superior to frequency masking is strongest on patch-tokenized architectures (ViTs). On CNNs the family gap largely vanishes (Table 5), and on global-cue tasks with large ViTs frequency masking becomes competitive (Table 7). The design guidance is therefore architecture- and task-conditional rather than universal.

**Mechanistic understanding.**   Finding 1 motivates a unified *structural-preservation principle* as a qualitative account for family-dependent stability. This account is intentionally lightweight and centers on two ingredients: spatial coherence and spectral overlap. The appendix formalization in Section A.3 should therefore be read as a conceptual restatement of that lens, with retained energy serving only as a minimal non-degeneracy term rather than as a separately validated empirical predictor.

### A.11   Extended Discussion

Under the controlled M2A protocol (fixed losses, random selection, shared masking schedule, single-step update), the main empirical findings can be read through the two qualitative ingredients of the structural-preservation hypothesis. In the family-comparison and lifelong-streaming results, patch masking is strongest because its contiguous support preserves spatial coherence across views and across time, so the consistency objective keeps receiving compatible signals even as the stream evolves. Pixel masking is weaker for the opposite reason: it fragments the masked set and therefore degrades coherence even when the masked ratio is

matched. Frequency masking is not uniformly bad, but its failure cases are organized by spectral overlap: blur-type corruptions make frequency masking collapse, while pixelate and related digital corruptions expose when frequency masking removes the very band that remains informative.

The same interpretation also explains the scoped exceptions. In the architecture study, the coherence advantage of patch masking is largest on patch-tokenized ViTs because masking aligned to the token grid directly determines which tokens remain usable, whereas on CNNs overlapping receptive fields dilute that advantage. In the aquaculture setting and on ViT-L/16, low-frequency masking becomes competitive because the task relies more on global cues and the resulting spectral overlap penalty is milder, so reduced spatial coherence is less damaging than in standard object-centric corruption benchmarks. Even then, the ablations show that no family is immune on the hardest cases: extreme severities can still produce residual spikes, which indicates that the hypothesis should be interpreted as a qualitative diagnostic for relative stability rather than as a claim of perfect robustness. Because all of the above patterns are established within the M2A framework, the broader suggestion that masking-family choice is an important design axis for CTTA in general should be read as a plausible implication of our study rather than as a conclusion that has been verified across all possible adaptation protocols.

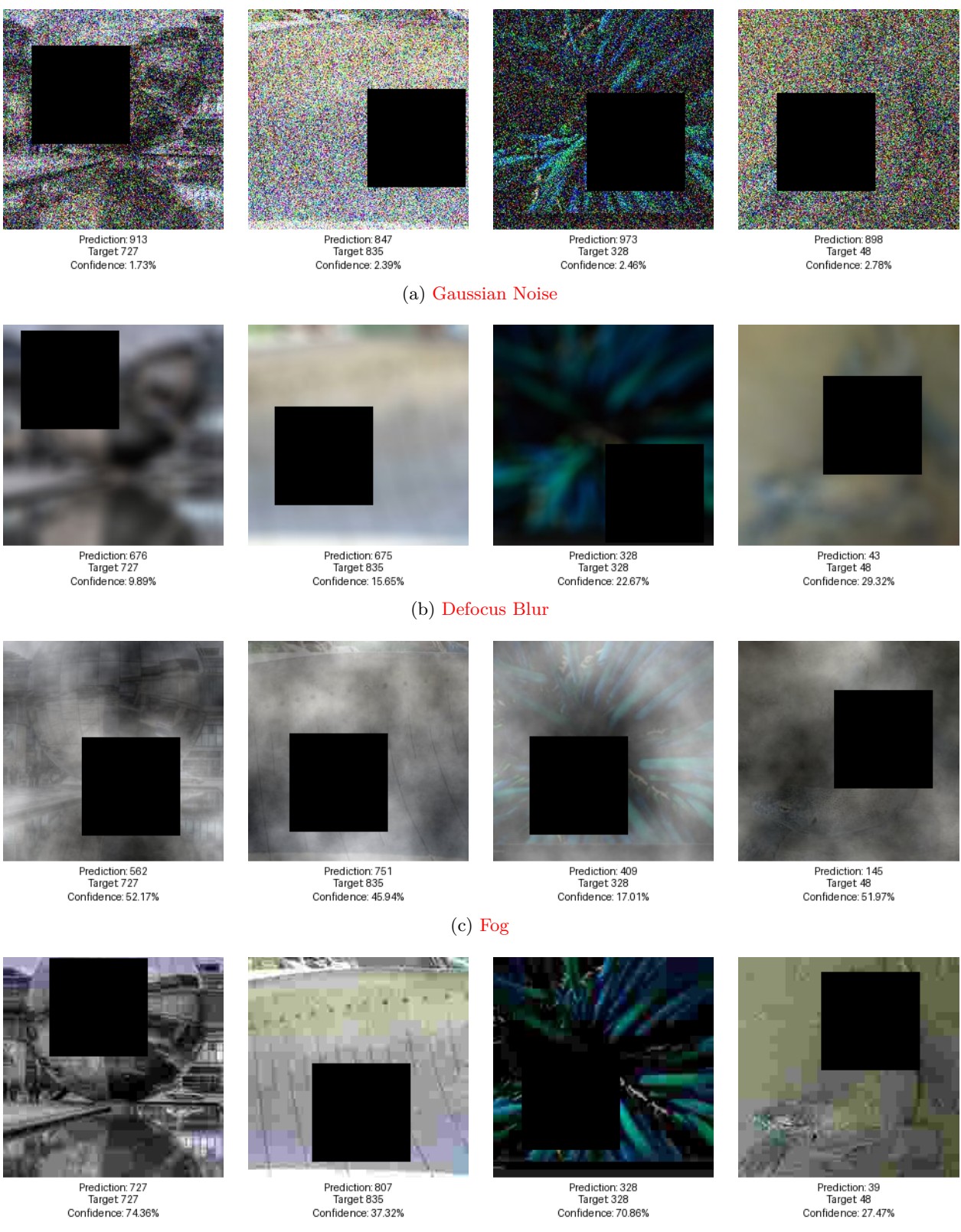

Figure 20: M2A(SM:Patch) predicted samples from ImageNet-C (severity 5) under CTTA. We save the images of the least-5 confident samples from the first domain (Gaussian Noise) and then track their progress and predictions across succeeding domains. Confidence is measured by the softmax probability of the predicted class. Starting from the rightmost column, the target classes are **planetarium**, **sundial**, **sea urchin**, and **giant lizard**.

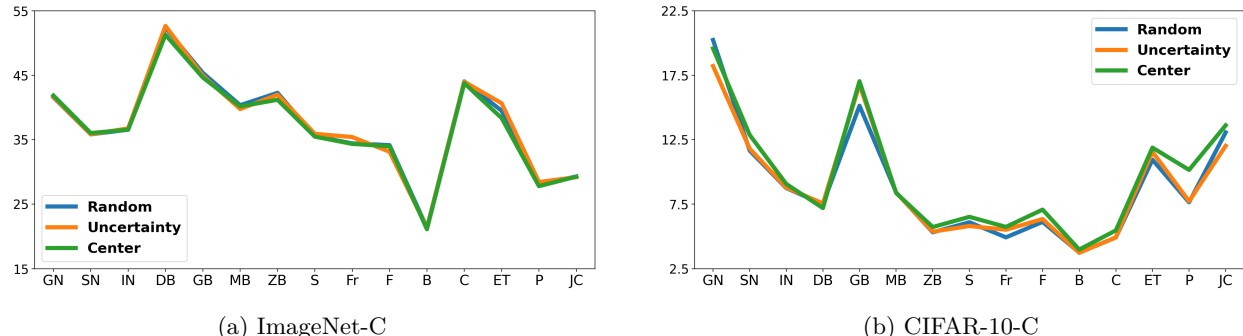

(a) ImageNet-C               (b) CIFAR-10-C

Figure 21: Patch masking under different masking-selection strategies, reported as classification error rate (%). We compare Random, Uncertainty, and Center patch selection. Random is the default M2A strategy discussed in Section 3.

