# OpenReview forum: "Family Matters: A Systematic Study of Spatial vs. Frequency Masking for Continual Test-Time Adaptation"
_TMLR — Accepted by TMLR_

### Review · Reviewer_46P9 · 2026-03-20

**Summary Of Contributions:**

This paper presents a systematic study of masking design in continual test-time adaptation (CTTA), arguing that prior work has overlooked the impact of the masking family by confounding it with selection strategies. To isolate this effect, the authors propose a controlled framework (M2A) that fixes the selection strategy and varies only the masking family, comparing spatial and frequency-based masking across multiple benchmarks and architectures. Their experiments show that spatial masking consistently yields stable and superior performance, while frequency masking often leads to catastrophic failure due to overlap with corruption spectra. The paper introduces a structural-preservation perspective to explain these results and further demonstrates that the optimal masking choice depends on architecture-task alignment, providing actionable design guidelines for CTTA systems.

**Audience:**

Yes

**Audience Explanation:**

This paper explores a design choice in continual test-time adaptation, which is relevant to researchers in distribution shift, robustness, and self-supervised learning. Its experimental insights and practical guidelines make it valuable to the audience.

**Broader Impact Concerns:**

No concerns.

**Claims And Evidence:**

Yes

**Claims Explanation:**

The paper demonstrates a clean experimental design by isolating the masking family, fixing the selection strategy and all other components, which enables a clear and controlled analysis of its impact. This design is complemented by an evaluation across multiple datasets and settings. The consistency of results across these diverse experiments strengthens the validity of the conclusions and provides convincing evidence that the masking family plays a critical role in CTTA performance.

**Requested Changes:**

* While the structural-preservation hypothesis is insightful, it would strengthen the paper to include a more formal theoretical analysis (such as a mathematical analysis) that explains when and why different masking families succeed or fail.
* It would be valuable to provide a brief analysis of how masking families interact with different CTTA variants (such as alternative update strategies or training schemes), to better understand whether the observed effects are generalizable.

---

> ### Author Response · Authors · 2026-04-21
> **Official Comment of Authors - Part 1**
>
> Dear Reviewer 46P9,
>
> Thank you for your thoughtful feedback, especially your suggestions to more formally clarify the structural-preservation hypothesis and to assess how the masking-family effect behaves across alternative CTTA variants. All revisions and changes to the manuscript are indicated in red text.
>
> **R3-1: “While the structural-preservation hypothesis is insightful, it would strengthen the paper to include a more formal theoretical analysis (such as a mathematical analysis) that explains when and why different masking families succeed or fail.”**
> - We thank the reviewer for this suggestion and have expanded **Appendix A.3 (Structural-Preservation Hypothesis)** with a more explicit conceptual formalization of the structural-preservation lens. Specifically, we introduce a schematic structural-preservation index (**see Eqs. (6)–(7)  in Appendix A.3**) as part of that formalization. As also noted in response to **Reviewer x47C (R1-1)**, this appendix discussion was revised not only to make the mathematical account clearer, but also to sharpen its epistemic framing: the manuscript now presents structural preservation as a supported qualitative interpretation for organizing when masking families are more or less stable, rather than as mechanistic proof, and we also clarify that $\mathcal{I}_{\mathrm{pres}}$, and $\tau$ are not estimated in the experiments.
> - Correspondingly, two statements in **Appendix A.3** are written as heuristic interpretive conditions: when the index is relatively large, the masked view is unlikely to substantially alter class-relevant label information, and for empirically stable masking families this condition should tend to hold for most inputs and mask draws along the CTTA stream. The formulation is intended as a qualitative bridge from the structural-preservation intuition to the observed family-level trends.
>
> **Eq. (6)**:
> - $\\mathcal{I}_{\\mathrm{pres}}(\\mathcal{F};x,t)$ is defined as the product of
> - $\\rho_{\\mathcal{F}}^{(t)}(x), \\quad
> \\mathcal{C}_{\\mathcal{F}}(M^{(t)}), \\quad
> (1-\\chi(\\mathcal{F},x)).$
>
> **Eq. (7)**:
> - $\\mathcal{I}_{\\mathrm{pres}}(\\mathcal{F};x,t) \\ge \\tau$ more likely stable
> - $\\mathcal{I}_{\\mathrm{pres}}(\\mathcal{F};x,t) < \\tau$ more likely unstable
>
> **Heuristic condition 1 (label-information preservation).** If $\mathcal{I}_{\mathrm{pres}}$ is relatively large, then the masked view is unlikely to substantially alter the class-relevant conditional label information present in $x$. In other words, structurally safe masking should preserve the dominant predictive content well enough that the class posterior is only mildly perturbed, rather than exactly invariant.
>
> **Heuristic condition 2 (preservation along the continual stream).** For masking families that are empirically stable, the preceding condition should tend to hold for most inputs and mask draws encountered along the CTTA stream at the masking levels used by M2A. Conversely, collapse-prone families should violate it more often on some domain-level pairs $(D_s,t)$, so that unstable views are encountered frequently enough to degrade adaptation over time. This distinction is intended only as a qualitative way to separate stable from unstable families, not as a proved high-probability guarantee.

---

> > ### Author Response · Authors · 2026-04-21
> > **Official Comment of Authors - Part 2**
> >
> > **R3-2: “It would be valuable to provide a brief analysis of how masking families interact with different CTTA variants (such as alternative update strategies or training schemes), to better understand whether the observed effects are generalizable.”**
> > - We thank the reviewer for this suggestion and have performed additional analysis in **Appendix A.8.2 (Masking Families)**, specifically **Figures 15, 16, and 17**. Figure 15 evaluates masking families under alternative CTTA training strategies by incorporating Model Recovery, Sample Filtering, and their combination into M2A, following  **EATA (Niu et al., 2022)** and **SAR (Niu et al., 2023)**. The main result is that these alternative update heuristics reduce absolute error for several families and suppress some corruption-specific spikes, especially when Model Recovery and Sample Filtering are combined, but they do not change the overall family ordering. In particular, patch masking remains the most stable and lowest-error family overall, while pixel and frequency masking remain more corruption-sensitive. Figures 16 and 17 provide supporting evidence that this effect is not tied to a single training setup: the same ordering persists under alternative loss formulations and under different numbers of test-time optimization steps. Taken together, the new appendix results suggest that the observed masking-family effect is generalizable across CTTA variants, while alternative training schemes mainly affect the magnitude of robustness rather than the qualitative ranking of masking families.
> >
> > **REFERENCES:**
> > - Niu, S., Wu, J., Zhang, Y., Chen, Y., Zheng, S. D., Zhao, P., & Tan, M. (2022). Efficient test-time model adaptation without forgetting. International Conference on Machine Learning.
> > - Niu, S., Wu, J., Zhang, Y., Wen, Z., Chen, Y., Zhao, P., & Tan, M. (2023). Towards stable test-time adaptation in dynamic wild world. International Conference on Learning Representations.

---

### Review · Reviewer_hn9X · 2026-03-24

**Summary Of Contributions:**

The paper delivers a systematic study of test-time training adaptation. It focuses on the masked image modeling paradigm and carefully ablates how different masking families affect the final performance. The paper makes two primary findings: (1) patch-wise masking is more effective than frequency-based masking; (2) the optimal network architecture depends on the specific dataset and task.

**Audience:**

No

**Audience Explanation:**

Overall, the conclusions of the paper do not add much to the community. There is no clear technical novelty claimed. It is unsurprising that the best architecture may vary across datasets. Although the paper reports good performance on several benchmarks and datasets, it remains unclear where the improvements originate, especially in comparison with recent work.

**Broader Impact Concerns:**

There is no broader impact section provided by the paper. I do not find it to be of critical concern.

**Claims And Evidence:**

No

**Claims Explanation:**

While the paper claims a systematic study of masking families, it only examines patch masking, pixel masking, and frequency masking. Other potentially effective strategies, such as grid-wise masking and block-wise masking, are not studied. The hyperparameter of masking ratio, which is crucial for self-supervised learning tasks, is not carefully explored. More broadly, other masking-related strategies, such as blurring, noising, and color jittering, are not considered. Therefore, the study is far from systematic.

The final configuration of the model is straightforward and simple, using patch-wise masking with random position selection. Notably, it achieves on-par or even better performance compared to several closely related methods, such as continual-MAE and REM. Given that these methods also rely on patch-wise masking, it is not clear why this approach performs better. Is it because random masking is stronger than uncertainty-based or attention-based selection? If so, the paper does not explicitly make this claim. If not, the paper fails to provide a discussion of where the performance gains come from. The only relevant mention is a brief sentence in the experimental section: “these baselines also differ from M2A in loss functions and auxiliary components, so this comparison provides suggestive context rather than a controlled quantification of S.”

**Requested Changes:**

1. The study should include broader scope of masking, and an analysis of the masking ratio.
2. The study should give details and analysis about how and why the method outperforms continual-MAE and REM. This would clarify the performance gains.

---

> ### Author Response · Authors · 2026-04-21
> **Official Comment of Authors - Part 1**
>
> Dear Reviewer hn9X,
>
> Thank you for your feedback and comments, particularly your questions about the scope of our findings, the systematicity of the masking study, the exploration of masking-related hyperparameters, and the paper’s broader technical contribution, and we address each of these points below. All revisions and changes to the manuscript are indicated in red text.
>
> **R2-1: “The paper makes two primary findings: (1) patch-wise masking is more effective than frequency-based masking; (2) the optimal network architecture depends on the specific dataset and task.”**
> - We appreciate this concise summary and would like to clarify Finding 1, because our claim is more nuanced than "patch-wise masking being universally better than frequency-based masking". Our result is conditional rather than universal: spatial masking is more effective on patch-tokenized architectures and tasks with spatially localized cues, while frequency masking can be competitive or even superior on global-cue tasks with high-capacity models. This is reflected in **Table 5**, where low-frequency masking achieves better results (compared to patch) on CNN-based architectures, and in **Table 7(b)**, where on MRSFFIA-C dataset with ViT-L/16, low-frequency masking outperforms patch masking and even obtains best result overall.
> - Accordingly, Finding 1 is not that patch masking always wins, but that the “masking family determines whether adaptation accumulates useful structure or compounds errors” depending on the architecture-task regime. In the revised manuscript, we now frame structural preservation consistently as a conceptual explanation for these family-level trends; this wording now appears explicitly in **Section 1 (Introduction)** and is carried through the later sections.
> - **The nuance is critical:** frequency masking fails under specific spectral overlaps, but can succeed when the corruption does not attack the masked band and the architecture can absorb global perturbations. Our structural‑preservation principle is intended as a qualitative account for when each family is appropriate.
>
> **R2-2: “... it only examines patch masking, pixel masking, and frequency masking. Other potentially effective strategies, such as grid-wise masking and block-wise masking, are not studied.”... “More broadly, other masking-related strategies, such as blurring, noising, and color jittering, are not considered. Therefore, the study is far from systematic.”**
> - In the revised manuscript, we implement and evaluate three mask sampling schemes—Block-Patch, Free-Patch, and Grid-Patch—on ImageNet-C (**Appendix A.5, Table 8**). Our “random spatial (patch) masking” in **Section 3.1** is exactly the Block-Patch sampler from **Appendix A.5**, where a single contiguous random patch block is zeroed out. To remove ambiguity, we now state this explicitly in Section 3 and repeat it in **Section 4**: all main results use contiguous Block-Patch, while Free-Patch is reserved for the scattered token-aligned appendix ablations.
> - Our notion of a masking family, following the MIM literature (Hondru et al.), is zero-out masking on input regions rather than generic augmentation. **Table 8** compares Free-Patch (random non-contiguous patches) and Grid-Patch (checkerboard masking) with Block-Patch; both alternatives underperform and show larger corruption-specific spikes, while Block-Patch achieves the lowest mean error across all masking curricula. This confirms that the block-wise random spatial masking used in our main experiments is the most stable and effective scheme in our study.
> - Blurring, noise addition, and color jitter are more naturally treated as standard corruptions or augmentation strategies, not part of the masking-family axis studied here. Because inputs are already corrupted (e.g., impulse noise, defocus blur), adding such transforms would confound the masking-family axis with the augmentation-policy axis.
> - To make this scope explicit, we added the following clarification in **Section 3**: “We restrict to zero-out masking families, following MIM literature (Hondru et al., 2024)...” As also emphasized in response to **Reviewer x47C (R1-3)**, this narrower definition is part of a broader claim-scope clarification: the paper’s conclusions are controlled findings within the M2A framework, while broader design guidance is presented as a plausible implication rather than a universal claim.
>
> **Table 8. Different mask sampling types. Entries report classification error rate (\%) per corruption type on ImageNet-C.:**
> | Sampling    | Mask Steps  | Mean ↓ |
> |-------------|-------------|-------:|
> | Free-Patch  | 0%-10%-20%  |  51.00 |
> |             | 0%-20%-40%  |  38.81 |
> |             | 0%-30%-60%  |  42.25 |
> | Grid-Patch  | 0%-10%-20%  |  51.37 |
> |             | 0%-20%-40%  |  78.51 |
> |             | 0%-30%-60%  |  43.72 |
> | Block-Patch | 0%-10%-20%  |  37.39 |
> |             | 0%-20%-40%  |  36.75 |
> |             | 0%-30%-60%  |  37.24 |

---

> ### Author Response · Authors · 2026-04-21
> **Official Comment of Authors - Part 2**
>
> **R2-3: “The hyperparameter of masking ratio, which is crucial for self-supervised learning tasks, is not carefully explored.”**
> - We agree that this aspect deserved a clearer treatment, and we have revised the manuscript to make the existing analysis more explicit. We explored the masking ratio systematically through the mask step α and the number of masked views. **Figure 5(c)** varies the masking schedule from 0–10–20% to 0–20–40% and 0–30–60%, and **Figure 5(b)** varies the number of views (1, 3, 5). We made this more explicit by adding a sentence in **Section 4.6.4 Ablation Study**.
> - We also note that **Figure 13 in the Appendix** provides a fine-grained mask ablation by varying mask granularity through the number of masked patches, which directly addresses the reviewer’s broader concern that mask-related hyperparameters were not explored carefully. It shows that performance is stable at low patch counts but can collapse once masking becomes too fragmented, so our study examines not only masking strength through ratio schedules and number of views, but also how the structure of the mask itself affects adaptation.
>
> **R2-4: “…it achieves on-par or even better performance compared to several closely related methods, such as continual-MAE and REM. Given that these methods also rely on patch-wise masking, it is not clear why this approach performs better.”**
> - We thank the reviewer for raising this point. This concern is closely related to the controlled-comparison issue raised by **Reviewer x47C (R1-2)**: although M2A, continual-MAE, and REM all use patch-wise masking at a high level, they also differ in losses, auxiliary components, and the overall adaptation recipe, so the cross-method table comparisons should be read as contextual rather than as isolating evidence for why one method outperforms another.
> - To narrow one plausible factor in a more controlled way, we added **Appendix A.9, Mask Selection Heuristics (Figure 21)**, where we fix the masking family to the same contiguous patch family and vary only the mask selection strategy $\mathcal{S}$, comparing Random, Uncertainty, and Center patch selection. The results show that these heuristic choices produce only small differences, with no consistent advantage over the default random selection. This suggests that M2A’s gains are not primarily due to a more elaborate patch-selection heuristic, but rather to using a structurally safe patch family within a stable prediction-centric CTTA objective.
> - Consistently, **Appendix A.8.1 (Hyperparameter Robustness), Figure 12** shows that M2A (CL+EL) and a REM-style (CL+RL) and random patch selection have nearly identical curves, which suggests that the remaining gap to Continual-MAE and REM is more likely due to their additional auxiliary components (e.g., HOG reconstruction or ranking hierarchy) and broader adaptation recipe than to loss or selection design alone. In other words, once the contiguous patch family is fixed, simple random patch selection already captures most of the stability and performance benefits. We therefore present **Appendix A.9 / Figure 21** as partial controlled evidence within the M2A framework, not as a complete explanation of all cross-method differences.
>
> **R2-5: “Overall, the conclusions of the paper do not add much to the community. There is no clear technical novelty claimed”**
> - We thank the reviewer for this feedback and respectfully clarify that we believe our work provides a practically relevant insight to the community. Our contribution here is not a new CTTA variant, but a controlled study that isolates a design axis which previous work has largely treated as fixed and shows that it materially affects stability. In particular, the catastrophic collapse of some masking families and its explanation through structural preservation, to our knowledge, non-obvious findings that have not been characterized in prior CTTA work. These results also provide practical guidance, since they suggest when a masking family is likely to be safe or risky under a given corruption profile.
> - We therefore see the contribution not as superficial novelty, but as a controlled empirical and conceptual study that prompted detailed methodological scrutiny from multiple angles from other reviewers. We also hope the assessment will consider the work’s overall scientific value, recognizing the importance of meaningful empirical clarification even in the absence of a new algorithm.
>
>
> **REFERENCES:**
> - Hondru, V., Croitoru, F., Minaee, S., Ionescu, R., & Sebe, N. (2024). Masked Image Modeling: A Survey. International Journal of Computer Vision, 133, 7154 - 7200.

---

> > ### Comment · Reviewer_hn9X · 2026-04-27
> >
> > I thank the authors for the informative rebuttal. After reading it, I have the following questions:
> >
> > 1. Thank you for clarifying the nuanced contributions. It is reasonable that the optimal masking strategy depends on both the network architecture and the dataset. It is not a significant contribution to me. In addition, prior work suggests that spatial masking can be less effective for CNNs due to information leakage (e.g., https://arxiv.org/pdf/2205.13943).
> > 2. After sweeping the mask ratio, does this change the conclusion regarding the optimal masking ratio across different scenarios? My understanding is that the initial submission used a fixed, predefined mask ratio.
> > 3. Thank you for elaborating on the differences from continual-MAE and REM. I understand that many implementation details differ, but I would like more clarity on what specifically makes M2A stronger, especially given that the baselines incorporate additional components such as HOG reconstruction and ranking hierarchy. While the current explanation is helpful, it does not yet provide a fully causal understanding.

---

> ### Author Response · Authors · 2026-05-03
> **Official Comment of Authors - Part 3**
>
> We thank the reviewer for the follow-up and for clarifying which aspects of our contribution and analysis remain of concern. We address each of these follow-up points below.
>
> **R2.1-1: “...It is reasonable that the optimal masking strategy depends on both the network architecture and the dataset. It is not a significant contribution to me…”**
> - We thank the reviewer for this insightful follow-up and for pointing out this line of prior work. We agree the key issue is not only whether the finding is correct, but why it matters. Our view is that the contribution is exposing a non-obvious failure mode of a default often left untested in masking-based CTTA: patch masking is commonly adopted across architectures, yet our results show that this default can weaken, break, or even reverse under corrupted online adaptation.
> - More concretely, the masking family is not just a minor implementation detail. In our study, the family gap shrinks on traditional CNNs and can even flip in boundary cases such as **Table 7(b)**, where on **MRSFFIA-C with ViT-L/16**, low-frequency masking outperforms patch masking. This is why we view the comparison to **Li et al. (A2MIM)** as complementary rather than contradictory: they study MIM pre-training on mostly **clean images**, whereas we study CTTA under heavily corrupted target streams.
> - In this CTTA setting, we do not claim that spatial masking is uniformly strong on CNNs. In **Section 4.6 (Table 5)**, patch masking is not the best choice on traditional CNNs such as ResNet-50, ResNeXt-50, and WideResNet-50, which is consistent with our structural-preservation account because overlapping receptive fields can dilute contiguous patch occlusion. At the same time, masking-based adaptation is not ineffective on CNNs in general: patch masking remains strongest on ConvNeXt-B/L, and in **Appendix A.7.2 (Table 10)**, M2A remains the best method overall on the CNN-based DeepLabV2 **segmentation** benchmark.
> - We therefore present the contribution not as a universal claim, but as a controlled empirical characterization of where the common patch-masking default remains reliable, where it weakens, and where it breaks.
>
> **R2.1-2: “After sweeping the mask ratio, does this change the conclusion regarding the optimal masking ratio across different scenarios?...”**
> - We thank the reviewer for this helpful follow-up. Based on the additional sweeps on **CIFAR10-C** and **Frequency Masking**, our main conclusion remains unchanged: they do not reveal a new scenario-specific optimal masking ratio. Instead, they show that for the stable masking family, a broad range of moderate easy-to-hard schedules behaves similarly, whereas for frequency masking, no comparable tuning removes the underlying instability.
> - Concretely, **Section 4.6.4, Figure 5(b-c) (ImageNet-C)** and **Appendix A.8.1, Figure 14(c)** show that for patch masking on **CIFAR-10-C**, the schedules 0-10-20%, 0-20-40%, and 0-30-60% yield very similar error profiles, as in **Figure 5(c)**, while **Figure 14(b)** shows that using multiple masked views is better than a single view, with only small differences once we move beyond one view. This indicates that, for the stable patch family, the masking family and a moderate easy-to-hard schedule matter more than finely tuning one exact percentage. **Appendix A.8.5** gives the complementary picture for low-frequency masking: **Figures 19(b-c)** show that changing the number of views or the masking schedule can modulate the size of some corruption-specific spikes, but does not remove them.
> - So the sweep reinforces the same design recommendation: patch masking is robust across a broad range of masking strengths, whereas low-frequency masking remains sensitive even after ratio tuning.

---

> ### Author Response · Authors · 2026-05-03
> **Official Comment of Authors - Part 4**
>
> **R2.1-3: “Thank you for elaborating on the differences from continual-MAE and REM...While the current explanation is helpful, it does not yet provide a fully causal understanding.”**
> - We thank the reviewer and agree at the outset that, without re-implementing **Continual-MAE** and **REM** under our exact setup, we cannot yet provide a fully causal explanation for the remaining gap. What our revision can do is narrow which factors seem less likely to explain it and state more clearly what remains unresolved.
> - First, **Appendix A.9 (Figure 21)** shows that once we fix the masking family to the same contiguous patch family, changing the **input-based** mask-location rule among Random, Center, and Uncertainty produces only small differences, so these heuristics do not appear to be the main driver. Second, **Figure 12** shows that M2A and a REM-style loss variant with random patch selection have nearly identical curves, suggesting that the ranking loss by itself is also not the main source of the gap.
> - What remains open is whether the gap to **Continual-MAE** and **REM** comes mainly from their auxiliary components, from their **model-dependent** token-selection heuristics under corruption, or from an interaction between these choices. The evidence we do have is that they use token-scored patch subsets rather than M2A’s single contiguous Block-Patch, and **Appendix A.5 (Table 8)** shows that this Block-Patch variant is the most stable patch sampler in our study. We therefore present the current takeaway as a falsifiable hypothesis rather than a causal claim, and fully isolating the remaining cross-method gap would require a dedicated controlled re-implementation beyond this revision.

---

### Review · Reviewer_x47C · 2026-04-11

**Summary Of Contributions:**

**Summary**

This paper studies an underexplored design axis in masking-based continual test-time adaptation (CTTA): the masking family itself. The paper introduces a controlled CTTA instantiation, M2A, which fixes the selection strategy, losses, masking schedule, and optimization protocol, while varying only the masking family across spatial and frequency-based alternatives.

The main finding is that the masking family has a major impact on CTTA stability and effectiveness. In the evaluated settings, patch masking is consistently the strongest and most stable choice, especially on patch-tokenized architectures and long continual streams, while frequency masking is more fragile and can collapse when the masked band overlaps with the corruption’s spectral signature. The paper further proposes a structural-preservation perspective to explain these observations, and supports its conclusions with experiments on corruption benchmarks, forward transfer and multiple architectures.

**Strength**

The paper addresses a clear and meaningful question by isolating the masking-family axis, which is usually confounded with mask selection strategy in prior masking-based CTTA methods.

The experimental design is reasonably well controlled within the M2A framework. By fixing the rest of the adaptation recipe and varying only the masking family, the paper provides credible evidence that this design axis has a strong impact on CTTA performance.

The empirical evaluation is broad and practically useful. The paper goes beyond standard benchmark comparisons and studies forward transfer, long-horizon adaptation, dynamic shifts, and architecture dependence, leading to more convincing and actionable design guidance.

**Weaknesses**

The structural-preservation explanation is plausible, but the evidence is still mostly correlational. The current experiments are consistent with this interpretation, but they do not fully establish it as the main mechanism behind the observed behavior.

The comparisons to prior masking-based methods are not fully controlled. These baselines differ from M2A not only in mask selection strategy, but also in losses, auxiliary components, and the overall adaptation recipe.

**Additional Comments:**

N/A

**Audience:**

Yes

**Audience Explanation:**

Researchers working on test-time adaptation, continual adaptation, and robust vision systems would likely be interested in these findings.

**Broader Impact Concerns:**

No major concerns.

**Claims And Evidence:**

Yes

**Claims Explanation:**

The empirical claim is reasonably well supported. Within the controlled M2A framework, the paper provides clear evidence that the choice of masking family has a substantial effect on CTTA performance and long-term stability. This is supported by experiments across standard corruption benchmarks, forward transfer, lifelong adaptation, dynamic shifts, and multiple architectures, and the overall empirical pattern is fairly consistent.

**Requested Changes:**

Better separate the paper’s strongest empirical claim from its broader interpretive claims. The results convincingly show that masking family matters within the proposed M2A framework, but the current writing sometimes makes the conclusions sound more general than the evidence fully supports.

Clarify the relation to prior frequency-based TTA methods such as SPA, especially why they are discussed in the paper but not included as direct baselines in the main experimental tables, and to what extent the difference is due to protocol mismatch versus additional stabilization components.

---

> ### Author Response · Authors · 2026-04-21
> **Official Comments of Authors - Part 1**
>
> Dear Reviewer x47C,
>
> Thank you for your constructive feedback, particularly your emphasis on clarifying the limits of our mechanistic interpretation, tightening the scope of our empirical claims, and distinguishing controlled evidence from broader contextual comparisons. All revisions and changes to the manuscript are indicated in red text.
>
> **R1-1: “The structural-preservation explanation is plausible, but the evidence is still mostly correlational. The current experiments are consistent with this interpretation, but they do not fully establish it as the main mechanism behind the observed behavior.”**
> - We thank the reviewer for this important clarification. The main empirical takeaway of the paper is that, within the M2A framework, masking family has a strong effect on CTTA stability. Structural preservation is our preferred qualitative interpretation of this pattern, not a mechanism that the current experiments definitively establish. We therefore agree that the evidence remains correlational, and our revision primarily narrows the claim. In **Section 3** and **Appendix A.3**, we now make this separation explicit by presenting the results as consistent with the structural-preservation account while acknowledging that architecture, task structure, and corruption-specific effects may also contribute. This change is intended to better align the manuscript’s claims with what the evidence currently supports.
>
> **R1-2: “The comparisons to prior masking-based methods are not fully controlled. These baselines differ from M2A not only in mask selection strategy, but also in losses, auxiliary components, and the overall adaptation recipe.”**
> - We thank the reviewer for this important point. We agree that the comparisons to prior masking-based methods are not fully controlled, since methods such as Continual-MAE and REM differ from M2A in mask selection strategy, losses, auxiliary components, and the overall adaptation recipe. We therefore do not interpret the cross-method table comparisons as isolating masking design alone, but only as contextual evidence. To add a more controlled analysis, we added **Appendix A.9 (Figure 21)**, where we fix the masking family to the same contiguous patch family within M2A and vary only the mask-selection strategy. This experiment does not fully explain the gap to Continual-MAE or REM, but it suggests that once the masking family is fixed, selection heuristics are a secondary factor, as Random, Uncertainty, and Center selection perform similarly. We believe this better clarifies both the value and the limits of the new ablation.

---

> ### Author Response · Authors · 2026-04-21
> **Official Comment of Authors - Part 2**
>
> **R1-3: “Better separate the paper’s strongest empirical claim from its broader interpretive claims. The results convincingly show that masking family matters within the proposed M2A framework, but the current writing sometimes makes the conclusions sound more general than the evidence fully supports.”**
> - We agree with the reviewer that the controlled empirical result and the broader design implication should be more clearly separated. In the revised **Section 5 (Conclusion and Future Work)**, we now open by explicitly scoping the main finding to the M2A protocol (fixed losses, random selection strategy, shared masking schedule, and single-step update) and state that the masking-family ordering is established under that specific framework. A new second paragraph in Section 5 further clarifies that whether the same ordering holds under substantially different losses, update rules, or adaptation protocols remains an open empirical question, and that the broader design guidance is presented as a plausible implication rather than a fully general conclusion. In **Appendix A.11 (Extended Discussion)**, we similarly preface the interpretation with the M2A protocol scope, and close by noting that the suggestion that masking-family choice matters for CTTA in general should be read as a plausible implication of the study, not a conclusion verified across all possible adaptation methods. We believe these revisions now cleanly separate what the experiments directly show from the higher-level interpretation we draw from them.
>
> **R1-4: “Clarify the relation to prior frequency-based TTA methods such as SPA, especially why they are discussed in the paper but not included as direct baselines in the main experimental tables, and to what extent the difference is due to protocol mismatch versus additional stabilization components.”**
> - We thank the reviewer for this comment. We now include **SPA (Niu et al., 2025)** in the benchmark tables (**Table 1**) for CIFAR-10-C, CIFAR-100-C, and ImageNet-C to provide broader empirical context for frequency-based adaptation methods. At the same time, we do not interpret this comparison as a controlled test of masking-family effects, because SPA differs from M2A in several coupled ways: it uses a weak-to-strong self-bootstrapping objective with confidence gating, injects Gaussian noise as an additional stabilization signal, and is primarily designed for single-pass fully TTA rather than continual TTA without resets. As a result, any performance gap between SPA and M2A cannot be cleanly attributed to protocol mismatch alone or to the additional stabilization components alone, since both vary together with the masking design. We therefore present SPA as a useful contextual baseline, while reserving masking-family conclusions for the more controlled within-framework analyses in the paper.
>
> **REFERENCES:**
> - Niu, S., Chen, G., Zhao, P., Wang, T., Wu, P., & Shen, Z. (2025). Self-Bootstrapping for Versatile Test-Time Adaptation. International Conference on Machine Learning (ICML).

---

### Decision · Action_Editor_NMeW · 2026-05-19

**Recommendation:** Accept with minor revision

**Additional Comments:**

The paper presents an extensive empirical study of masking strategies for continual test-time adaptation that focuses on an underexplored design axis. The experimental evaluation shows a consistent pattern that masking family has a significant effect on stability and performance under continuous domain shift. While two of the reviewers lean to accept, one other reviewer leans to reject, expressing concerns about the strength of the paper’s insights and the differences in experimental settings within the comparisons to previous masking-based methods. The AE finds these to be valid criticisms but nevertheless considers the empirical findings of this paper to likely have some value to CTTA researchers. A recommendation is made for acceptance with minor revisions.

The final manuscript needs to make clearer that different masking families change the input to different degrees, and these differences are not explicitly controlled across conditions in the experimental setup. This does not invalidate the empirical findings, but it does limit the extent to which the masking family can be interpreted as an independently isolated causal factor. This is currently discussed in the appendix and needs to also be briefly mentioned in the main text to ensure that readers clearly understand the scope of the claims.

**Audience:**

Yes

**Audience Explanation:**

There are members of the TMLR audience who would have interest in this paper because it highlights an underexplored design axis in continual test-time adaptation and provides a broad empirical comparison across masking families.

**Claims And Evidence:**

Yes

**Claims Explanation:**

The paper’s central claim is that the choice of masking family is the primary factor determining stability and performance in continual test-time adaptation, even more so than selection strategy. This claim is adequately supported and further validated by the additional empirical evidence provided during the discussion period. However, the perturbation strength may not be normalized across masking families, which limits the degree to which the experiments cleanly isolate the effect of the masking family itself. This limitation needs to be more explicitly acknowledged.

The claim regarding the structural-preservation explanation is fine after revision during the discussion period.